# Environment modulates protein heterogeneity through transcriptional and translational stop codon readthrough

Maria Luisa Romero Romero [1,2] ✉, Jonas Poehls [1,2], Anastasiia Kirilenko[1,2], Doris Richter[1,2], Tobias Jumel [1], Anna Shevchenko[1] & Agnes Toth-Petroczy [1,2,3] ✉

Stop codon readthrough events give rise to longer proteins, which may alter the protein's function, thereby generating short-lasting phenotypic variability from a single gene. In order to systematically assess the frequency and origin of stop codon readthrough events, we designed a library of reporters. We introduced premature stop codons into mScarlet, which enabled high-throughput quantification of protein synthesis termination errors in *E. coli* using fluorescent microscopy. We found that under stress conditions, stop codon readthrough may occur at rates as high as 80%, depending on the nucleotide context, suggesting that evolution frequently samples stop codon readthrough events. The analysis of selected reporters by mass spectrometry and RNA-seq showed that not only translation but also transcription errors contribute to stop codon readthrough. The RNA polymerase was more likely to misincorporate a nucleotide at premature stop codons. Proteome-wide detection of stop codon readthrough by mass spectrometry revealed that temperature regulated the expression of cryptic sequences generated by stop codon readthrough in *E. coli*. Overall, our findings suggest that the environment affects the accuracy of protein production, which increases protein heterogeneity when the organisms need to adapt to new conditions.

Protein synthesis termination can achieve high fidelity, yet it is not perfect. Within all life forms on Earth, the end of the translation process is signaled by stop codons and catalyzed by release factors[1–3]. This process has evolved to rewrite genomic information into canonical-size proteins accurately[4]. However, stop codon readthrough (SCR)[5] may occur either by a transcription error, when the RNA polymerase misincorporates a nucleotide and eliminates the stop codon, or by a translation error in which the ribosome misincorporates a tRNA at the stop codon (also called nonsense suppression[6,7]). Thus, SCR covers transcription and translation events that deviate from the genetic code without identifying a specific mechanism. These errors result in protein variants with extended

C termini[8–10], generating a heterogeneous protein-length population from a single gene.

Proteome diversification arising from SCR can potentially generate phenotypic variability, shaping the cell's fate[11]. For instance, errors in protein synthesis termination can be adaptive and functional[8,10,12–17] or non-adaptive[18], occasionally leading to fitness decrease[19]. Further, slippery sequences upstream the stop codon that cause the ribosome to slip and thereby lead to stop codon readthrough are often functional[14,20,21]. Transcription and translation errors may generate short-lasting phenotypic variability on a physiological time scale, faster than genomic mutations. Thus, SCR events may facilitate rapid adaptation to sudden environmental changes. These

[1]Max Planck Institute of Molecular Cell Biology and Genetics, 01307 Dresden, Germany. [2]Center for Systems Biology Dresden, 01307 Dresden, Germany. [3]Cluster of Excellence Physics of Life, TU Dresden, 01062 Dresden, Germany. ✉e-mail: romeroro@mpi-cbg.de; toth-petroczy@mpi-cbg.de

tremendous evolutionary implications reveal the need to study the rules that dictate SCR error rates under diverse living conditions.

Several studies have highlighted the relevance of SCR under diverse environmental conditions. Both carbon starvation[22] and excess glucose promote the readthrough of TGA in *E. coli* by lowering the pH[23]. However, these studies provided information for a given genetic context, detecting a maximum of 14% TGA readthrough for cells grown in LB supplemented with lactose[23]. Further, low growth temperatures dramatically increased ribosomal frameshift rates in *Bacillus subtilis*[24]. Nevertheless, the impact of temperature on SCR remains unknown.

Here, we systematically examined SCR events of all stop codons in a variety of genetic contexts under different temperatures and nutrient conditions. We specifically explored: i) How frequent errors occur in protein synthesis termination. ii) Whether non-optimum environmental conditions modulate the chances for evolution to encounter these events. iii) Whether they originate from translation or transcription errors. We designed a library of reporters that allowed for high-throughput quantification of protein synthesis termination errors in *E. coli* using fluorescent microscopy. We targeted 43 arbitrary positions along the mScarlet sequence, and, in each position, we mutated the wild-type codon to each of the three stop codons. Thus, only upon errors resulting in skipping protein synthesis termination would a full-length mScarlet be synthesized.

We confirmed that protein termination accuracy depends on the identity and genetic context of the stop codon. We then proposed a set of simple rules to predict hotspots in the protein sequence that are error-prone for protein synthesis termination. We further showed that environmental stress conditions such as low temperature and nutrient depletion increase the SCR rate of all stop codons. Accordingly, the opal stop codon TGA, present in 29% of the *E. coli* proteome, fails to terminate the protein synthesis in certain positions at a rate of up to 80% under stress conditions at a given nucleotide context. RNA-seq and mass spectrometry experiments of selected reporters revealed that protein synthesis mis-termination is not only due to ribosomal readthrough, but RNA polymerase errors also contribute to SCR. We found that the RNA polymerase is more likely to mis-incorporate a nucleotide at premature stop codons. Finally, mass spectrometry analysis of the K12 MG1655 *E. coli* proteome provided evidence of cryptic sequences revealed by SCR, validating our method for predicting SCR in a more natural context. Overall, our findings suggest a cross-talk between the environment and the flux of biological information that increases protein heterogeneity when organisms need to adapt to new conditions.

## Results

### Visualizing and quantifying stop codon readthrough events in *E. coli*

To monitor error rates in protein synthesis termination, we designed a fluorescence reporter by which stop codon readthrough can be visualized and quantified in *E. coli* inspired by the pioneering work of R.F. Rosenberger and G.Foskett[25] and the more recent study of Meyerovich et al.[24]. The strategy relies on introducing a premature stop codon into an mScarlet allele. Thus, only upon SCR, full-length and, therefore, functional mScarlet will be synthesized (Fig. 1A).

We first introduced a TGA stop codon in the randomly selected position 95 of an mScarlet flanked by two tags, a Strep-tag at the N-terminus and a His-tag at the C-terminus. On the one hand, these tags provided a way to purify expressed fragments. On the other hand, the His-tag served as an orthologous method to detect SCR events by Western blot. To tightly regulate the expression of the reporter, we used a low copy number vector and an inducible promoter.

We transformed a wild-type K12 MG1655 *E. coli* strain with the Gly95-TGA reporter. Then, we grew the transformed cells until stationary phase under optimal conditions, testing the expression of the Gly95-TGA reporter with titrated concentrations of the inducer. We

used cells transformed with the empty vector as negative control (NC) and cells expressing wild-type mScarlet as positive control (PC; Fig. 1B, C). While no signal was detected in the negative control, an elevated fluorescent signal with increasing inducer concentration was detected in the cells expressing the Gly95-TGA reporter and in the positive control. These results suggest that functional mScarlet was expressed in the cells carrying the Gly95-TGA reporter, indicating SCR. Assuming that the fluorescence properties of the SCR variants were not altered, we were able to quantify the error rate by calculating the relative fluorescence signal as the percentage of the median compared with the PC median. We found that SCR was not a rare event; at a 400 µg/L AHT concentration, protein synthesis termination failed for 2.1% of the expressed Gly95-TGA reporters (Fig. 1C).

Since we compared cells with orders of magnitude difference in fluorescence intensity, we confirmed that the fluorescence measurements were within the dynamic range of the instrument, and the relationship between mScarlet concentration and fluorescent intensity was linear (Supplementary Fig. 1). Therefore, relative fluorescence is a valid approximation for the relative protein abundance, allowing us to quantify the stop codon readthrough error rate.

### Stop codon readthrough events are frequent in *E. coli*

As demonstrated in previous studies[24,26–29], we showed proof-of-principle that SCR occurred at the opal stop codon, TGA, and could be visualized and quantified with a fluorescent reporter. Next, we extended the strategy to study SCR event frequency and whether and how the identity of the stop codon and the sequence context regulated the error rate of protein synthesis termination. To address these questions in a high-throughput fashion, we designed a library of reporters for stop codon readthrough. We randomly targeted 43 codons along the mScarlet sequence and mutated them to each of the three stop codons. We confirmed by Sanger sequencing that the final library consisted of 117 reporters: 38 with TAA, 39 with TAG, and 40 with TGA stop codons. We individually transformed *E. coli* with the reporters and grew them, inducing the reporter expression under normal conditions (37 °C in LB medium) in a 384-well plate. Next, we automatically imaged the cells in the stationary phase to determine fluorescence (Fig. 1D, see methods for detailed description).

Several reporters (16 out of 117) displayed high fluorescence output, meaning full-length mScarlet was expressed (Fig. 1E). This is a rather unexpected result considering that the premature stop codons were introduced randomly along the mScarlet sequence, without considering the genome context that may promote SCR. For instance, the fluorescence distribution of the cells carrying a reporter with a premature stop codon at position 105 displayed a relative median fluorescence of 0.7 to 6%, depending on the stop codon identity (Fig. 1E).

The stop codon identity seemed pivotal in regulating the SCR propensity, with TGA being the most error-prone stop codon. 14 out of the 40 reporters carrying a premature TGA displayed median fluorescence above the threshold defined as the median plus two standard deviations of the fluorescence signal of the NC. Further, the reporters with a stop codon at position 105 suggested that its identity affected the likelihood of SCR events. The highest relative fluorescence values were measured for TGA and followed the trend TGA > TAG > TAA, corroborating previous observations in bacteria[30], yeast[31], and mammals[16,32]. Surprisingly, the least efficient stop codon, TGA, is present in 29% of the *E. coli* proteome. However, TAG, which is less prone to be read-through, is only present in 8% of the *E. coli* proteome[33]. We surmise, that the highest error rate detected at TGA could be due to the inefficient RF2 (prfB allele), which recognizes TGA, present in the wild-type K-12 MG1655 *E. coli*[34,35].

In summary, our results suggest that SCR events are more frequent in *E. coli* than previously thought.

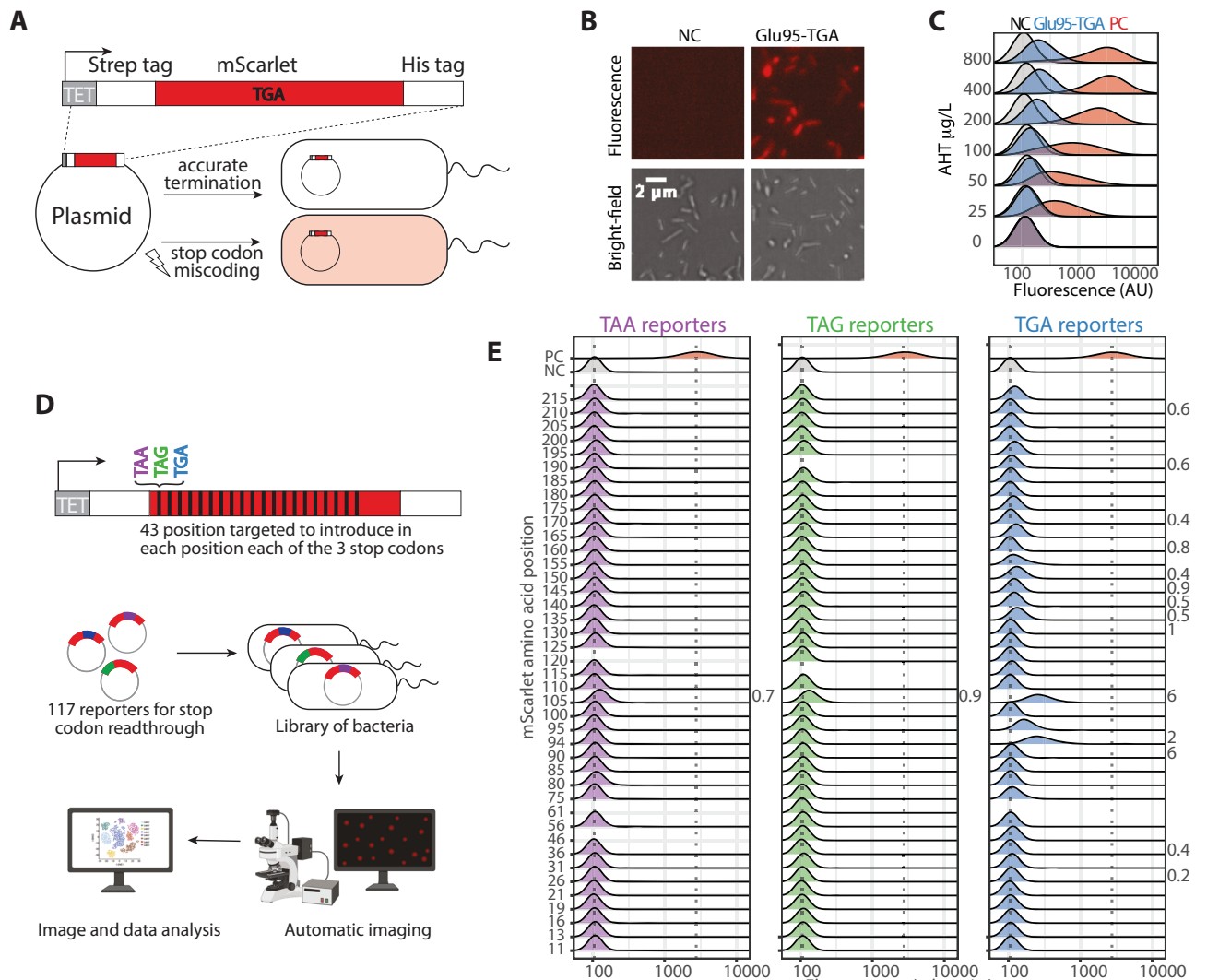

**Fig. 1 | Visualization and quantification of stop codon readthrough events in *E. coli* using fluorescence reporters. A** To study stop codon readthrough (SCR), we designed a fluorescence reporter introducing a premature stop codon into an mScarlet allele. **B** We detected full-length mScarlet and, therefore, SCR events in cells transformed with the reporter. **C** Titrating the expression of the reporter increases the detection of SCR events. **D** We designed a library of reporters for SCR mutating to TAA (purple), TAG (green), and TGA (blue), 41 randomly selected codons along the mScarlet sequence. We studied the library of reporters in a high-throughput fashion. **E** Fluorescence distributions displayed by the *E. coli* cells, transformed with each library's reporters and grown at 37 °C in rich media. The propensity of SCR, calculated as the percent of the median fluorescence compared with the positive control (PC, wild-type mScarlet), is shown for the distributions with a median fluorescence higher than the negative control. There are hotspots in the protein sequence prone to SCR. The identity of the stop codon affected the likelihood of stop codon readthrough with the trend TGA > TAG > TAA. Each fluorescence distribution was derived from one biological replicate. Source data are provided as a Source data file. Figure 1A and D were created with BioRender.com released under a Creative Commons Attribution-NonCommercial-NoDerivs 4.0 International license.

## Non-optimal growth temperatures and nutrient scarcity promote stop codon readthrough

SCR events can generate short-lasting phenotypic variability faster than genomic mutations. Thus, SCR events may facilitate rapid evolution due to sudden environmental changes. However, up to now, little attention has been paid to the effect of environmental conditions on SCR. It is known that the readthrough of stop codons in *E. coli* depends on the growth media[22,23]. However, how temperature affects SCR remains unknown. To examine this, we first screened our previously described library of reporters expressed in wild-type *E. coli* grown at different temperatures (Fig. 2A and Supplementary Fig. 2).

A closer inspection of the temperature-effect scan revealed hotspots in the sequence prone to SCR. We define hotspot positions, where the premature stop codon is read-through in more than 50% of the tested conditions, for example, positions 105 and 135 (Fig. 2A).

These hotspots seem to depend on the sequence context of the premature stop codon since, regardless of the stop codon identity, they are likely to prevent correct protein synthesis termination at different temperatures.

Within all temperatures, TAA is least likely to be read-through while TGA is the most likely, in agreement with previous observations[32,36,37]. At each temperature, more TGA reporters display stop codon readthrough than TAG and TAA (Fig. 2A, B, and Supplementary Fig. 2). Additionally, the stop codon termination error rates follow the same trend at the hotspot positions: TGA > TAG > TAA (Fig. 2C).

The SCR rates vary considerably as a function of the temperature. Interestingly, at lower temperatures, more reporters display stop codon readthrough events (Fig. 2A, Supplementary Figs. 2 and 3AB). At the hotspot positions, the protein synthesis termination seems more

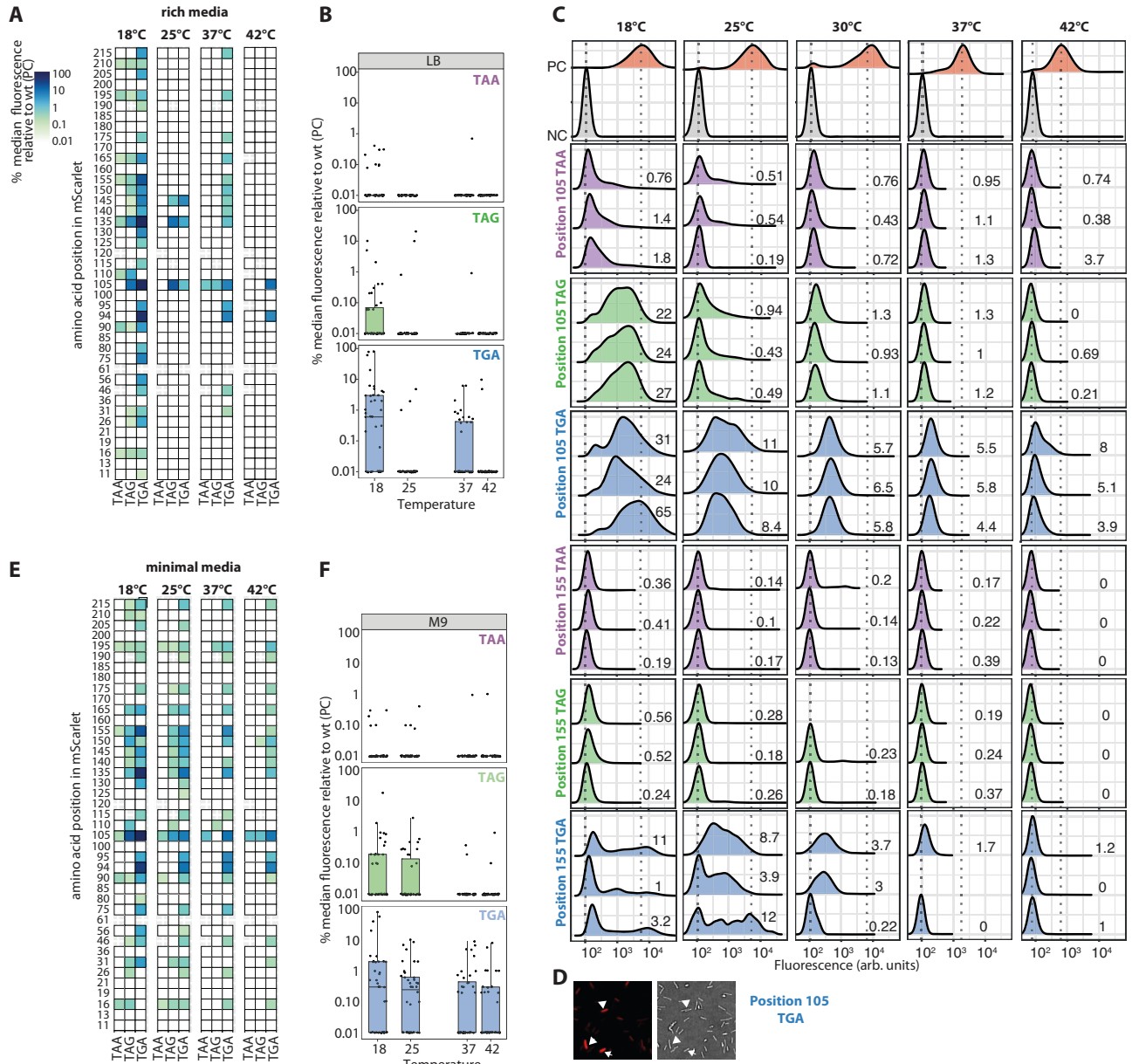

**Fig. 2 | Non-optimal growth temperatures and nutrient scarcity promoting stop codon readthrough. A** In cells grown in rich media, the stop codon readthrough (SCR) levels varied considerably as a function of temperature. At 18 °C, more reporters displayed SCR events and at a higher rate. The plotted variable is the median percentage of each reporter's fluorescence distribution, normalized by the wild-type (PC). **B** When cells were grown in rich media, TAA was the most accurate codon terminating the protein synthesis, and TGA was the least accurate at all tested temperatures. The plotted variable is the median percentage of each reporter's fluorescence distribution relative to the wild-type (PC). The top subpanel describes the reporters with TAA ($N$ = [38, 38, 38, 35], mean = [0.05, 0.01, 0.03, 0.01]), the middle with TAG ($N$ = [39, 39, 39, 39], mean = [0.49, 0.80, 0.03, 0.01], and the bottom with TGA ($N$ = [40, 40, 40, 40], mean = [7.23, 0.21, 0.51, 0.38]) respectively at 18 °C, 25 °C, 37 °C and 42 °C. **C** Fluorescence distributions of *E. coli* cultures transformed with the reporters that carried each of the stop codons at positions 105 and 155, grown at 18 °C, 25 °C, 30 °C, 37 °C, and 42 °C in LB. Three biological replicates are shown per reporter. The vertical lines represent the NC and PC median fluorescence (wild-type mScarlet). **D** Fluorescence and bright field

images of an *E. coli* culture transformed with the reporter that introduced a TGA at position 105 of the mScarlet. While some cells were prone to SCR (marked with arrows), others did not exhibit it. **E** Nutrient scarcity promoted SCR. In cells grown in minimal media, SCR levels increased while lowering the growth temperature. The plotted variable is the median percentage of each reporter's fluorescence distribution, normalized by the wild-type (PC). **F** When cells were grown in a minimal medium, TAA was the most accurate codon terminating the protein synthesis, and TGA was the least for all tested temperatures. The plotted variable is the median percentage of each reporter's fluorescence distribution relative to the wild-type (PC). The top subpanel describes the reporters with TAA ($N$ = [38, 38, 38, 38], mean = [0.04, 0.04, 0.04, 0.04]), the middle with TAG ($N$ = [39, 39, 39, 39], mean = [0.72, 0.17, 0.02, 0.04], and the bottom with TGA ($N$ = [40, 40, 40, 40], mean = [5.58, 1.11, 0.89, 0.52]) respectively at 18 °C, 25 °C, 37 °C and 42 °C. The boxs span the interquartile range (25th to 75th percentile) with the median marked by a horizontal line. Whiskers extend 1.5 times the interquartile range from the quartiles. Source data are provided as a Source data file.

accurate at the optimal growth temperature, 37 °C (Supplementary Fig. 2). However, at low temperature (18 °C), when TGA is at position 135, the SCR is surprisingly frequent, at a rate of ~80%. The experiment at 25 °C seems to be an outlier on the observed temperature trend as it presents fewer reporters with SCR than the one at 37 °C (Fig. 2A, B, and Supplementary Fig. 2). However, most reporters that exhibited SCR events at 37 °C did so with a low rate (below 1%, Supplementary Fig. 2).

To better address the temperature effect and study the reproducibility among biological replicates, we focused on the reporters with TAA, TAG, and TGA at positions 105 and 155. We analyzed three biological replicates for each reporter at 18 °C, 25 °C, 30 °C, 37 °C, and 42 °C (Fig. 2C). These experiments, along with their statistical analysis (see Methods section for further details), revealed two findings: i) significant cell-to-cell heterogeneity within a clonal population (Fig. 2C, D and Supplementary Table 1), and ii) higher levels of SCR with decreasing temperature, particularly evident at 18 °C, when a stop codon was inserted at positions 105 and 155 (Supplementary Fig. 4 and Supplementary Table 2). Previous studies have suggested that such heterogeneity may facilitate adaptation to changing environments[26]. The observed heterogeneity poses challenges for quantifying SCR. Throughout this work, we utilized relative median fluorescence as a summary statistic to quantify SCR error rates. Nevertheless, these numeric values do not fully represent the distributions' complexity, as they are neither normal nor symmetric.

To test whether the observed temperature effect is an artifact due to the unfolding of mScarlet at high temperatures, we assayed the stability of mScarlet, showing that mScarlet has an apparent melting temperature ($T_m$) above 80 °C (Supplementary Fig. 3C).

To examine how nutrient depletion modulates protein synthesis termination accuracy, we screened our library of reporters in wild-type *E. coli* grown in a minimal medium, M9, at different temperatures (Fig. 2D and Supplementary Fig. 2). The most surprising result was the frequent occurrence of SCR events observed in low-nutrient conditions. Indeed, the number of hotspots prone to SCR increased to 10 (position 16, 90, 105, 135, 140, 150, 155, 165, and 195; Fig. 2D). Besides the higher error rate, the stop codon identity (Fig. 2E, Supplementary Figs. 2, 3A, B) appeared to play a similar role when growing *E. coli* in minimal media and in rich media. I.e., the SCR level followed the trend TGA > TAG > TAA.

To assess the generality of the temperature effect, we statistically analyzed all reporters under all experimental conditions (see Methods section). We assessed the median of the fluorescence relative to the wild-type, excluding those reporters that showed no SCR at the studied temperatures. The analyses revealed evidence of a non-linear temperature-driven effect on SCR: 18 °C > 25 °C - 37 °C > 42 °C (Supplementary Fig. 5, Supplementary Table 3).

To further explore the key nutrients essential for keeping an accurate protein synthesis termination, we first supplemented the minimal media with a higher carbon source concentration (1.6% glycerol, Supplementary Fig. 6A) and, secondly, with a higher casamino acid concentration (0.4% casamino acid, Supplementary Fig. 6B). In both cases, the protein synthesis termination accuracy increased. Accordingly, when cells were grown in non-supplemented minimal media, 18 reporters with TGA displayed a median fluorescence above threshold (defined as the median plus two standard deviations of the NC's fluorescence signal). However, when supplementing with higher glycerol and casamino acid concentrations, only 6 and 8 reporters, respectively, presented a median fluorescence above the threshold. This suggests that both nutrients, the carbon source and the casamino acids, are essential to prevent SCR events.

Global inspection of these results suggests that environmental stress conditions, such as non-optimal growth temperature or nutrient scarcity, promote stop codon readthrough and, thus, protein heterogeneity. A similar temperature effect has been previously reported for ribosomal frameshift errors[24], suggesting an increase in different types of protein synthesis errors at low temperatures. A known general response to environmental stress conditions in *E. coli* comprises the downregulation of genes related to transcription and ribosomal biogenesis[38]. This general stress response may end up in an increase in protein synthesis errors, which can modulate protein heterogeneity. Since sudden environmental changes can modulate protein heterogeneity in a rapid and short-lasting manner, we hypothesize that *E. coli* might make use of SCR to quickly adapt to sudden and short-lasting stress conditions.

## C-terminal His-tag detection as an orthologous method to quantify stop codon readthrough events

We have linked the fluorescence signal of mScarlet with SCR events. However, the readthrough of a stop codon could disrupt the mScarlet structure or decrease its stability, resulting in full-length yet non-functional (i.e., non-fluorescent or dark) mScarlet or in mScarlet with potentially enhanced fluorescence. We used an orthologous method to detect SCR events and evaluated the limitations of our fluorescent reporters' library. The reporters include a His-tag at the C-terminal (Fig. 1A, D), and thus, only upon SCR His-tag is synthesized. We performed western blotting using anti-Histag antibodies as an orthologous method to detect and quantify SCR events.

We expressed reporters in wild-type *E. coli* at 18 °C in LB media and calculated the relative expression of His-tag compared to the positive control using Western blot (Supplementary Figs. 7 and 8). We subsequently compared the results with the corresponding relative fluorescence values obtained by microscopy. Most reporters consistently show similar SCR rate measured by fluorescence and by western blot assays. Only a few reporters presented a high His-tag expression but did not exhibit fluorescence ("dark reporters") (Supplementary Table 7).

We further checked that the highest SCR error rates detected (Trp-94-TGA, Ala-105-TGA, and Pro-135-TGA) were not artifacts resulting from an enhanced mScarlet fluorescence variant. Since tryptophan is usually misincorporated by SCR at TGA sites (see results section "Stop codon readthrough events related to amino acid misincorporations" and references[39,40]), we studied the fluorescence of the variants where tryptophan was introduced in positions 94 (same sequence as the wild-type mScarlet), 105, and 135 (Supplementary Fig. 9). While the Ala-105-Trp mutant had a comparable fluorescence to the wild-type mScarlet, the Pro-135-Trp mutant displayed one order of magnitude reduced fluorescence (Supplementary Fig. 9). We confirmed a 79% SCR rate for the reporter that introduced a TGA at position 135 by His-tag detection (Supplementary Table 7). We hypothesize that the higher fluorescence observed previously for this reporter relates to a different amino acid misincorporation.

Overall, the C-terminal His-tag analysis helped us identify and tackle a limitation in our reporters' design. Most reporters estimate similar SCR rates with the fluorescence measurement as with the His-tag detection, proving that our fluorescent reporter library is a valid method for studying SCR events. Notably, we have considered the Western blot His-tag detection assay as a qualitative analysis. Consequently, for downstream analyses, we relied on fluorescence measurements to quantify SCR events.

## High G and low T content downstream of the stop codon increase the likelihood of stop codon readthrough

Our results show hotspots in the sequence prone to SCR events. Since we targeted 43 positions in the mScarlet sequence, we can systematically compare the genetic context effect on SCR. We analyzed the nucleotide occurrences surrounding the premature stop codon and the amino acid position to determine whether they correlate with protein synthesis termination errors.

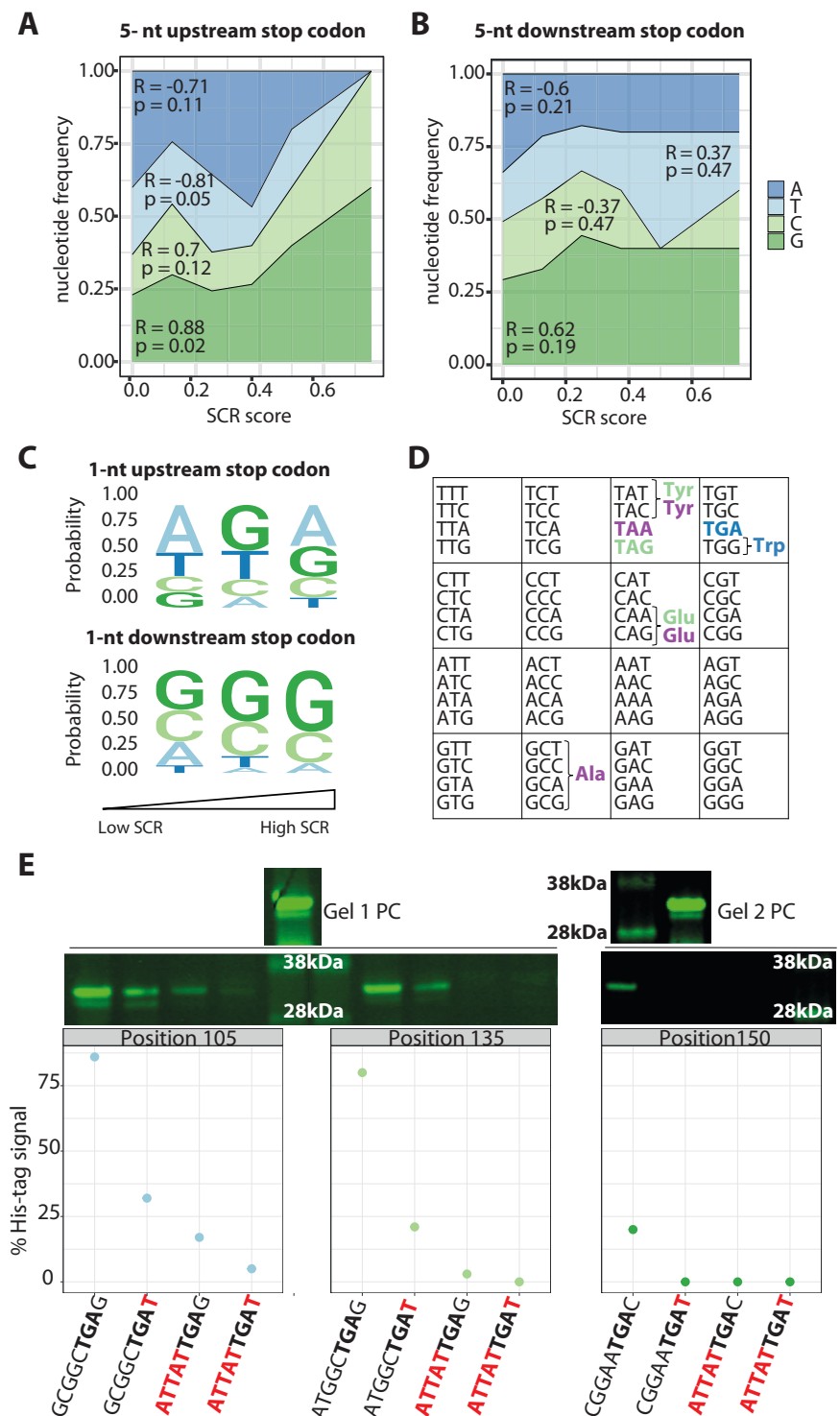

We derived a score to estimate the likelihood of an SCR event at a given position. Then, we binned the scored values and correlated them with the occurrence of each nucleotide in a 5-nt window upstream and downstream of the premature stop codon. The results show a significant positive correlation for G and negative for T content with SCR likelihood upstream of the premature stop codon (Fig. 3A). However, a non-significant correlation was found in a 5-nt window downstream of the premature stop codon (Fig. 3B) (see Methods section for further details). This is consistent with previous work reporting that the two amino acids at the C-terminus of the nascent peptide, and therefore the six nucleotides upstream of the stop codon, modulate termination error frequency in *E. coli*[41–43].

Numerous reports have shown that the nucleotide immediately following the stop codon influences SCR in eukaryotes[32,44] and prokaryotes[45,46]. Hence, we also studied the effect of the identity of the base following the stop codon on SCR. For this purpose, we used the previously described score for SCR likelihood. Sequence-logos of the base following the stop codon position suggest that the presence of T after the stop codon significantly increases the protein synthesis termination efficiency (Fig. 3C). Contrarily, G in the fourth position significantly increases the likelihood of SCR (Fig. 3C). This result agrees partially with a pioneer work showing how T and C favor termination efficiency[46]. However, it differs from previous observations in mammalian cells where the nucleotide at the +4 position increased SCR

**Fig. 3 | The nucleotide sequence downstream the premature stop codon's effect on the probability of stop codon readthrough.** **A** The effect of the identity of the nucleotides in a 5-amino acid window upstream of the stop codon on stop codon readthrough (SCR) probability. Statistical analysis was performed using Pearson correlation and two-sided testing. **B** High G content in a 5-amino acid window downstream of the stop codon does not increase the probability of SCR, while T decreases it. Statistical analysis was performed using Pearson correlation and two-sided testing. **C** Reporters were binned into three categories based on the SCR rates, and nucleotide frequencies were calculated 1-nt upstream and downstream of the stop codons. The identity of the base immediately before the stop codon did not impact SCR probability. The identity of the base immediately after the stop codon impacted SCR probability. T increases, and G decreases the protein synthesis termination efficiency. **D** Stop codon readthrough events occurred primarily due to amino acid misincorporations identified by mass spectrometry. TGA, in blue, was almost always replaced by tryptophan. TAA, in purple, was mainly replaced by alanine, glutamine, and tryptophan, while TGA, in green, was replaced by glutamine and tyrosine. Misincorporation of several other amino acids at the stop codon was clearly a minor process (relative abundance <10%). These MS experiments were performed with the reporters after purification with His-tag affinity resin. Thus, we could not quantify the likelihood of SCR events. However, the MS analyses did provide information on the relative abundance of the amino acids' misincorporation, presented in Supplementary Table 4. **E** The downstream mutation to T and the upstream mutation to ATTAT reduced the likelihood of SCR events for all three tested positions. To quantify the SCR likelihood, we measured the His-tag expression with western blot in *E. coli* cells transformed with these mutants, grown at 18 °C. We calculated the percentage of His-tag expression compared with the internal positive control (PC, wild-type mScarlet). Source data are provided as a Source data file.

frequency in the order C > U > A > G[32]. The identity of the base preceding the stop codon does not seem to influence the likelihood of SCR (Fig. 3D, see Methods section).

To experimentally confirm the previous findings, we designed a set of mutants and measured their SCR likelihood. Since these mutants changed the mScarlet's primary structure in up to four amino acids, which will likely affect its functionality, we measured the SCR events, detecting the His-tag expression with western blot. We focused on those reporters with the highest SCR score: those with a premature stop codon in positions 105, 135, and 155. For these reporters, we mutated the downstream region towards T because, according to our analysis, T reduces the likelihood of SCR events when placed after the stop codon. We mutated the upstream region towards ATTAT because T reduces the likelihood of SCR events when placed in a 5-nt window before the stop codon. This sequence is the richest in T content among reporters that did not exhibit SCR events in any of the studied conditions (Supplementary Table 6). We predicted these mutants to increase the protein synthesis termination accuracy. We transformed *E. coli* cells with these mutants and grew them at 18 °C. We observed that both the downstream mutations towards T and the upstream mutation towards ATTAT reduced the likelihood of SCR events for all three tested positions. Further, mutating the upstream and downstream regions simultaneously had an additive effect (Fig. 3E). Thus, we confirmed our prediction experimentally.

Finally, to study whether the distance to the C-terminus modulates the propensity of SCR events, we analyzed the correlation between the SCR score and the mScarlet amino acid position. We found no significant correlation (Supplementary Fig. 10, R = 0.15, p = 0.362).

Overall, these analyses indicate that the adjacent region to the premature stop codon (5-nt upstream and 1-nt downstream) influences the SCR rate. While G increases the SCR propensity, T has the opposite effect. The higher stability of the secondary structures of the mRNA with high GC content may hamper the fidelity of translation. To test this hypothesis, we predicted the minimum free energies (MFE) of the possible secondary structures in a 100-nt window downstream and upstream of the inserted stop codon. We analyzed the correlation between the SCR score and the minimum free energy of the predicted structures and found no significant correlation (Supplementary Fig. 11A, B, and C, see Methods for more details).

As expected, decreasing the temperature stabilized the predicted RNA secondary structures, which may explain the higher SCR errors at lower temperatures (Supplementary Fig. 11A, B, and C). Yet, the higher stability of the secondary structures cannot explain the presence of hotspots for SCR errors in the mScarlet sequence. Since, in the context of ribosomal frameshifting, it has been argued that local thermodynamic stability has a greater effect than the structure's overall stability[47], we focused next on the local thermodynamic stability of the secondary structures surrounding the premature stop codon and its correlation with the SCR scores (Supplementary Fig. 11 D and E,

see methods for more details). We found no correlation (Supplementary Fig. 11 D and E).

Overall, the predicted mRNA secondary structures could not explain the presence of hotspots for SCR events.

## Stop codon readthrough events related to amino acid misincorporations

Next, we investigated the events occurring at the position of the stop codon for the cases where SCR was detected. We chose nine reporters for which SCR events have been detected in the previous experiments, and the premature stop codon was located within tryptic peptide detectable by mass spectrometry. The selected reporters were expressed in *E. coli* grown on nutritionally rich medium (LB) at 18 °C; the expressed products were purified using C-terminal His-tag by Ni-NTA chromatography and analyzed by GeLC-MS/MS. Wild-type mScarlet sample was processed similarly and used as a control.

Protein synthesis termination error may relate to the misincorporation of an amino acid at the stop codon position, skipping the stop codon[39], or a frameshift. Our data showed that amino acid misincorporation occurred for all nine selected reporters producing a mixture of several products, and the misincorporated amino acids were not a random selection (Fig. 3D, Supplementary Fig. 12, and Supplementary Table 4). TGA stop codon was reproducibly replaced by tryptophan and cysteine, where tryptophan was favored according to quantitative estimates. Misincorporation of several other amino acids at the TGA position was clearly a minor process (Supplementary Table 4). On the other hand, the TAG stop codon followed another pattern. It was repeatedly replaced by tyrosine, glutamine, and lysine, with all three amino acids having comparable misincorporation rates. TAA stop codon exhibited, in the only sample studied, replacement by tyrosine, glutamine, lysine, and alanine. No misincorporation events were detected in the control sample.

Overall, our results agree with observations in *S. cerevisiae*, where TAG and TAA were found to be replaced by glutamine, lysine, and tyrosine, and TGA by tryptophan, cysteine, and arginine[40]. In mammals, tryptophan, cysteine, arginine, and serine can be incorporated at the TGA stop codon position[39]. Most detected amino acid misincorporations may be explained by a single nucleotide mismatch between the tRNA anticodon and the stop codon. A similar strategy is reported for stop-to-sense reassignment in some organisms, where, e.g., tRNA^Glu cognates TAG and TAA[48], and tRNA^Trp binds to TGA[49].

We detected only a single mass spectrum matched to a peptide with a stop codon deletion (the sample with TGA at position 190), suggesting that skipping the stop codon is a non-significant contributor to SCR events. No cases of extended deletion (stop codon together with 1-2 neighboring amino acids) were detected. We also did not observe alternative-frame peptides, as expected, since we purified the proteins using a C-terminal tag in the frame. Thus, our approach does not allow us to rule out the existence of frameshifts.

Altogether, our results indicate that SCR events are due to amino acid misincorporations, and the pattern of misincorporated amino acids depends on the stop codon identity.

## Low RNA polymerase accuracy at premature stop codons

We explored whether the amino acid misincorporations identified by LC-MS/MS were due to transcription or translation errors. We studied the mRNA sequence of 12 reporters, including the nine reporters previously studied by mass spectrometry and the mScarlet wild-type, expressed in wild-type *E. coli* grown at 18 °C in LB media (Data S2). We confirmed the sample sequences by DNA sequencing (Data S3).

The most surprising aspect of the RNA-seq results is the higher probability of mismatches at the premature stop codon sites (Supplementary Fig. 13A). Some of these mismatches do not result in an amino acid exchange in the protein sequence due to the degeneration of the genetic code, i.e., several codons encode for the same amino acid or stop signal. Nevertheless, it appears that the mismatches at the premature stop codon sites are unusually high (Supplementary Fig. 13B). The observed increase of mismatches may be the result of the selective degradation of mRNA containing premature stop codons[50,51], e.g., by Rho-mediated transcription termination of premature stop codon-carrying mRNAs. However, the prevalence of synonymous mismatches leading to another stop codon (Fig. 4E) suggests a higher RNA polymerase error rate at these stop codons. We hypothesized that the reason for the RNA polymerase errors could be the nucleotide context around the premature stop codon. However, the probability of a mismatch in a given position was significantly higher for nucleotides encoding for a premature stop codon than those encoding for an amino acid or the canonical stop codon (Fig. 4A). The same trend could be observed for non-synonymous mismatches only (Fig. 4B). To further study the effect of the sequence context, we analyzed the probability of a mismatch based on the adjacent nucleotides (Fig. 4CD). The analysis revealed that the identity of the adjacent nucleotides did not affect the accuracy of the RNA polymerase. Interestingly, according to this analysis, the RNA polymerase is more prone to mismatch at a stop codon when it is premature or it is in the ribosome's open reading frame, pointing to the coupling of translation and transcription as a putative reason for the higher inaccuracy of the RNA polymerase.

We subsequently moved to study whether the nucleotide misincorporations by the RNA polymerase errors match the protein sequences identified by MS. We aimed to clarify the source of the SCR: Is it transcriptional or translational? We analyzed the amino acids encoded by the mRNA sequences at the premature stop codon in each reporter (Fig. 4E). As expected, the wild-type stop codon was found in most of the reporters' mRNA sequences. Due to the degeneration of the genetic code, i.e., several codons code for the same amino acid or stop signal, RNA polymerase errors often result in a different stop codon only different in one nucleotide (e.g., TAA to TAG) (Fig. 4BD). However, occasionally, RNA polymerase errors result in an amino acid insertion at the premature stop codon site. The amino acid detected by mass spectrometry was often found already encoded in the mRNA sequences due to nucleotide misincorporations, although at much smaller proportions and among many other amino acids. Overall, the RNA-seq and mass spectrometry results suggest that mainly translation errors contribute to SCR events. This is in agreement with previous studies that revealed higher translational than transcriptional error rates[14,24,52]. Transcription errors contribute less to SCR events. Yet, they diversify the resulting protein sequence independent of the stop codon identity.

## Proteome-wide detection of stop codon readthrough events revealed the conditional expression of non-coding sequences in *E. coli*

To test whether and to what extent SCR occurs in a natural context, we analyzed the proteome of wild-type *E. coli* by mass spectrometry, searching for evidence of SCR events. We matched the acquired peptide fragmentation spectra to a customized database comprising canonical *E. coli* protein sequences and predicted products of stop codon readthrough events. Since growth temperature was proven to be essential for SCR, as described above, the experiments were conducted for cells grown at 37 °C and at 18 °C.

In total, we identified 16 peptides from non-coding regions, mapping to 15 different proteins (Supplementary Table 5). These peptides do not exist in canonical protein sequences and can be generated only if the corresponding stop codon is read-through. Of the 16 peptides, 12 covered the stop codon position, enabling the identification of the misincorporated amino acid.

While we found that TAA can be miscoded by many amino acids, TGA was preferentially miscoded by tryptophan (5 out of 10 cases, Fig. 5A, Supplementary Table 5). For the ribosomal protein rpsG, in addition to tryptophan, cysteine was identified as misincorporated at the TGA site (Supplementary Table 5 and Supplementary Fig. 14). Since tryptophane is one of the rarest amino acids in the *E. coli* proteome (comprising 2% of the proteome), it seems overrepresented among the SCR peptides. Notably, tryptophan misincorporation at the TGA position may be common in other organisms[40,48].

Of the 15 genes where SCR was detected, 9 contained TGA and 6 TAA, representing 0.77% of TGA- and 0.23% of TAA-containing *E. coli* genes, respectively. We did not detect SCR events for genes with a TAG stop codon, probably due to its low representation in *E. coli* (8%, Fig. 5A). This is in line with the observation from the reporter library of TGA being the most error-prone of the three codons and a previous ribosome profiling study on *E. coli* that observed enrichment for TGA and depletion of TAA in the SCR events detected[35].

Subsequently, we looked for additional stop codons in a 15-nt window downstream of the canonical *E. coli* stop codons. The probability of an additional stop codon correlates with the protein synthesis termination accuracy. For example, it is more likely to find an additional stop codon after TGA, the most error-prone stop codon, followed by TAG, and finally TAA (Fig. 5B). This trend is maintained when extending the search to 30-nt and 60-nt windows (Fig. 5B), suggesting differences in selection pressure to fix an additional stop codon, depending on the accuracy of the stop codon. However, this selection pressure could be affected by the fact that the wild-type K-12 MG1665 *E. coli* strain used here carries a defective RF2, which is less efficient in recognizing the TGA stop codon. We, therefore, repeated the analyses using the BL21 *E. coli* strain, which expresses a more efficient RF2 variant. Both *E. coli* strains showed the same results (Fig. 5B and Supplementary Fig. 15). Regardless of the efficiency of the RF2, it is more likely to find an additional stop codon after TGA, the most error-prone stop codon, followed by TAG and, lastly, TAA.

We detected more cases of SCR in *E. coli* samples grown at l8 °C than at 37 °C, in agreement with the fluorescence reporter's study (a peptide was considered present in a sample if it was identified in at least three out of six replicates). Five genes were exclusively detected at 18 °C. In comparison, only two genes were exclusively detected at 37 °C (Fig. 5C, Supplementary Tables 5 and 14). Overall, this reinforces the conclusion drawn from the library experiments that lower temperature increases SCR events (Fig. 5C).

Lastly, we wanted to analyze how the genome context modulates the likelihood of proteome-wide SCR. The previous experiments with the fluorescence reporters indicated that the identity of the base following the stop codon affected SCR likelihood (Fig. 3BC). While G seemed to increase the likelihood of SCR events, T appeared to decrease it. We subsequently explored whether the identified 15 genes with SCR follow the same trend. We calculated the enrichment of each nucleotide at the position immediately after the stop codon in the 15 identified genes compared with the rest of the genome. We confirmed that the identity of the nucleotide downstream of the stop

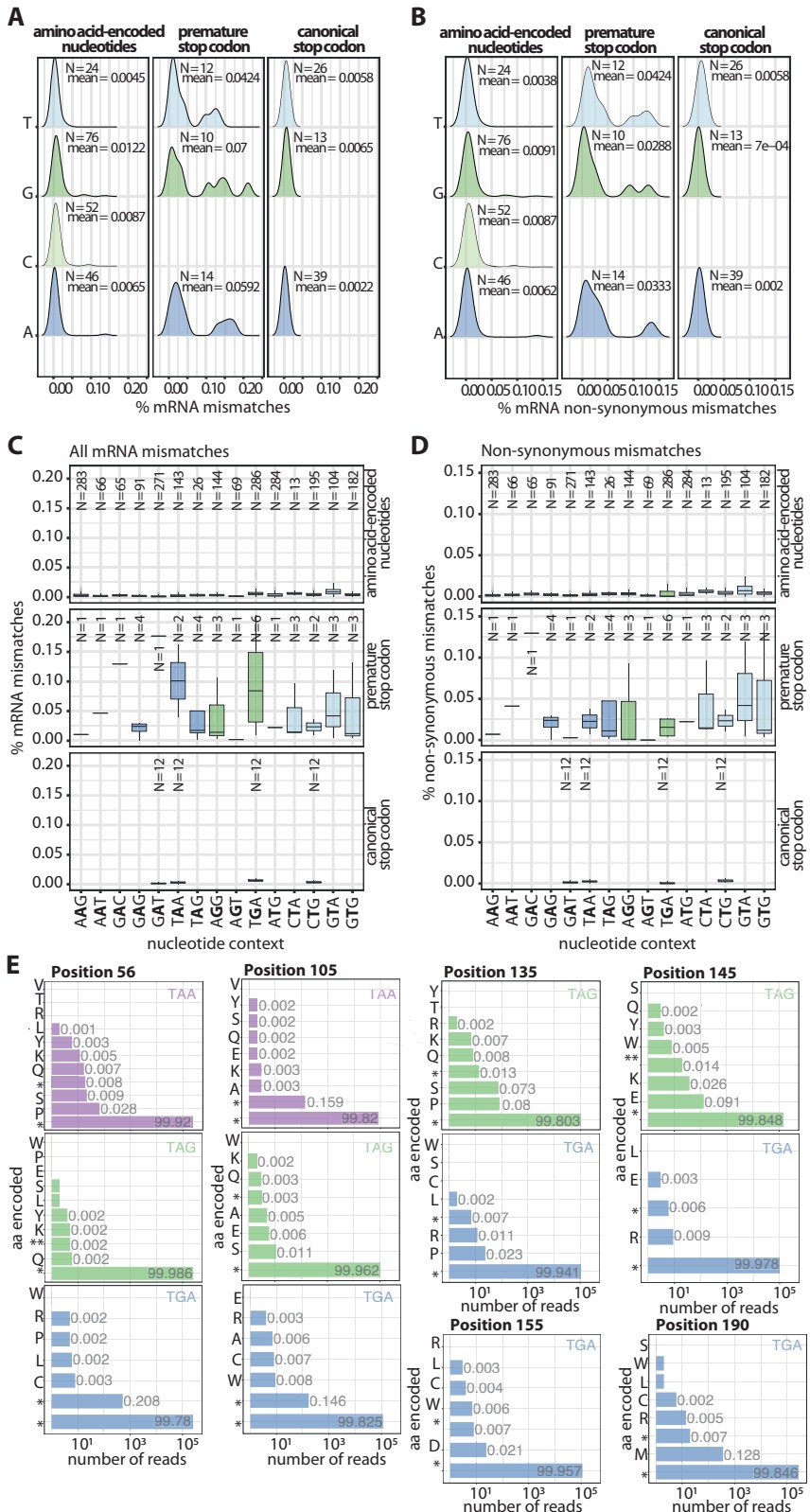

codon modulated the likelihood of SCR. While G decreased the protein synthesis termination accuracy, T had the opposite effect (Fig. 5D).

Overall, there was an agreement between the conclusions drawn from reporters and from the proteome-wide study. Furthermore, the proteome-wide mass spectrometry analysis revealed the expression of cryptic sequences, i.e., peptides from noncoding regions that could be generated only by SCR events, only expressed under certain environmental conditions.

## Discussion

This study demonstrates that evolution frequently samples stop codon readthrough (SCR) events in *E. coli*. Furthermore, we report that internal factors, such as stop codon identity and genetic context, and

**Fig. 4 | Low RNA polymerase accuracy at premature stop codons. A** The likelihood of the RNA polymerase mismatching was higher at premature stop codons. Density plots of %mRNA mismatches (nucleotide substitutions) observed in RNA-seq experiments at the four nucleotides (T, G, C, A) grouped by amino-acid encoding (left), premature stop codon (middle), or canonical stop codon (right) encoding nucleotides. **B** When focusing solely on non-synonymous mismatches (those leading to amino acid changes), RNA polymerase was more likely to incorporate a mismatched nucleotide at a stop codon when in the frame. The same analysis is presented as in panel A, but includes only non-synonymous mismatches. **C** The identity of the adjacent nucleotides did not affect the probability of RNA mismatch at a given position. The probability of RNA mismatch in a given nucleotide was higher when it encoded a premature stop codon than for canonical stop codon or out-of-frame stop codon sites. The plot shows the percentage of RNA

mismatches of a given nucleotide (represented in bold on the x-axis), depending on the adjacent nucleotides. **D** The identity of the adjacent nucleotides did not affect the probability of non-synonymous RNA mismatch at a given position. The same analyses are presented as in panel **C**, but only for RNA mismatches that lead to amino acid changes (non-synonymous mismatches). **E** RNA polymerase errors at premature stop codons resulted in a broad range of misincorporated amino acids. A bar plot showing the amino acid encoded by the mRNA sequence at the premature stop codon versus the number of reads. The percentage of encoded amino acids is shown next to each bar. The boxes span the interquartile range (25th to 75th percentile) with the median marked by a horizontal line. Whiskers extend 1.5 times the interquartile range from the quartiles. Source data is provided as a Source data file.

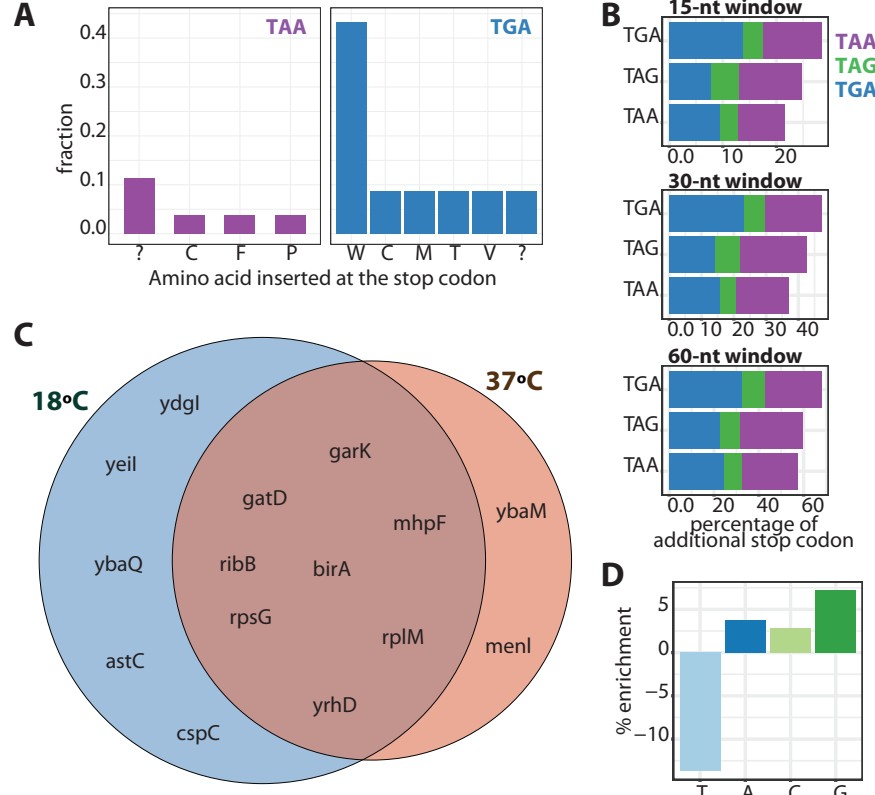

**Fig. 5 | Proteome-wide mass spectrometry detecting stop codon readthrough (SCR) events in 16 peptides mapping to 15 proteins in *E. coli*. A** We detected SCR in 0.23% of TAA *E. coli* proteins and 0.77% of TGA *E. coli* proteins. While we found that TAA could be miscoded by numerous amino acids with similar frequencies, TGA was most often miscoded by Trp. **B** The occurrence of an additional stop codon in a 15, 30, and 60-nt window downstream of the stop codon correlated with the protein synthesis accuracy of the stop codon in the *E. coli* genome of the K12

MG1655 strain. TGA, as the least accurate stop codon, had the highest frequency of an additional stop codon in the 3' regions of genes. **C** We detected more cases of SCR in *E. coli* samples grown at 18 °C than at 37 °C. Five genes were exclusively detected at 18 °C, and only two genes were exclusively detected at 37 °C. **D** The identity of the nucleotide downstream of the stop codon modulated the SCR probability. While G decreased the protein synthesis termination accuracy, T increased it. Source data are provided as a Source data file.

external factors, such as growth temperature or nutrients, modulate SCR event rates (Fig. 2, Supplementary Figs. 1–4). We have studied the impact of SCR events on the proteome (Fig. 5, Supplementary Table 5). Transcriptional errors at stop codons introduce a vast exploration of the mutational space since RNA polymerase often miscodes premature stop codons towards different codons independently of the stop codon identity (Fig. 4E). Translational errors introduce a comparatively minor exploration of the mutational space, depending on the stop codon identity (Fig. 3D, Supplementary Table 4), yet, they have higher rates, being the main contributor to SCR events.

Despite the significant number of functional SCR reported[8,10,12–16], most errors in protein synthesis termination are considered non-adaptive[18]. However, our findings show an error rate of up to 6%

when bacteria are grown under normal conditions (Fig. 1E) (previous error rates recorded by fluorescence reporters for non-programmed SCR in *E. coli* grown under normal conditions were 2%[23,26] and 0.4% in *B. subtilis*[24]). Even lower SCR levels have often been linked to functionality[12]. For example, it has been reported how a 1% ribosomal readthrough level at a short conserved stop codon context is used in animals and fungi to generate peroxisomal isoforms of metabolic enzymes[15]. Further, our results indicate that the selection pressure to prevent SCR is not enough to reduce the usage of the most error-prone stop codon or increase release factors' efficiency. Interestingly, wild-type *E. coli* K12 strains have an RF2 variant with reduced ability to terminate translation[35]. Instead, our data suggest selection pressure to fixate an additional stop codon downstream of the most error-prone stop codon (Fig. 5B).

Further, our data show that non-optimal growth temperatures and nutrient scarcity dramatically increase SCR events. As observed in some reporters, SCR occurs at a rate up to 80% (Supplementary Fig. 2) (the previous highest reported error rate for a non-programmed stop codon was 14% TGA readthrough for cells grown in LB supplemented with lactose[23]). The effect of nutrient scarcity in SCR may be due to the lower carbon source concentration, which has been previously linked to modulating the RF2 activity[23]. We hypothesize that the higher stability of the secondary structures of nucleotides at low temperatures could explain the effect of temperature on SCR. Strong mRNA secondary structures hamper the unfolding of the mRNA, potentially affecting the accuracy of protein synthesis[42]. The same arguments may support why the identity of the nucleotide at the adjacent regions affects the protein synthesis accuracy since stem-loops containing GC base pairs have been shown to decrease ribosomal accuracy[42]. Although we detect no significant correlation between predicted mRNA secondary structures and SCR likelihood (Supplementary Fig. 11), that may result from the prediction's limitations[53]. At least in the context of ribosomal frameshifting, the effect of mRNA structure on ribosome movement appears to depend not only on thermodynamic stability but also on the exact distance between structure and ribosome[47,54], the size of the structure[55], and the number of possible conformations[56,57]. These more complex relationships may not be captured by prediction methods.

On the other hand, it has been reported that the amino acid identity at the C-terminal of the nascent peptide modulates termination accuracy[41–43]. Proline and glycine in the −1 and −2 positions upstream of the stop codon increase termination error frequency[42]. Interestingly, codons encoding for proline and glycine are enriched in C and G. Thus, the observed correlation between GC content upstream of the stop codon and the fidelity of termination may be a consequence of the amino acid effect on the termination.

Based on these results, we propose avoiding TGA and optimizing the adjacent region of the stop codon when designing a vector for protein expression in *E. coli* to minimize SCR events. The most common strategy to purify proteins involves *E. coli*, often grown at low temperatures, which may promote the production of non-desired protein forms[58].

Ribosome profiling to explore stop codon readthrough proteome-wide in *Drosophila melanogaster* proposed that readthrough adds plasticity to the proteome during evolution[14]. Similarly, we hypothesize that bacteria could use SCR errors as a mechanism that allows the rapid diversification of its proteome to adapt to sudden environmental changes. Further, ribosome profiling of *E. coli* identified >50 genes having possible translation past the stop codon[35]. We also detected cryptic sequences among the *E. coli* proteome only expressed by SCR under cold shock. Further validation through quantitative mass spectrometry will be invaluable in confirming the identity of these 16 peptides. Exploring their role in generating protein diversity is an interesting open research question. Importantly, the proteome-wide analysis validates the proposed rules to predict endogenous SCR in vivo. Further studies are required to elucidate the phenotypes and potential functional role of these cryptic sequences.

Interestingly, we observed that genes within multi-genes operons are enriched in TGA, the most error-prone stop codon, while depleted in TAA, the most accurate one (Supplementary Fig. 16A). Further, genes within multi-gene operons are enriched in G and depleted in T in the position right after the TGA stop codon (Supplementary Fig. 16B). Since we observe that the presence of T increases while G decreases the protein synthesis termination accuracy, protein expression termination errors are predicted to be enhanced among genes within multi-gene operons. Further, 50% of the genes detected by the proteome-wide mass spectrometry analysis that suffered SCR events are within multi-gene operons (representing 43% of the *E. coli* proteome).

We investigated the source of error behind these high SCR levels. We discern that, although ribosomal errors are the main contributors to SCR, RNA polymerase errors contribute on a minor scale yet introduce a greater proteome diversification. Intriguingly, our study shows that RNA polymerase misincorporates nucleotides non-randomly, i.e., it mainly misincorporates at premature and in-frame stop codons. Future work will be required to identify the molecular mechanisms causing this bias in RNA polymerase error rates.

Our work highlights that both transcription and translation errors contribute to protein diversity. We show that SCR is more frequent than previously thought[12,15,23,24,26], thereby providing an evolutionary mechanism enabling cells to respond rapidly to the environment by increasing protein heterogeneity.

## Methods

### Gene Libraries

The reporter gene library was ordered from Twist Bioscience. The gene library was cloned in a pASK vector (Purchased from Addgene #65020. This plasmid is a modified version of the vector pASK-IBA3 plus, with the following changes: the substitution of the ampicillin-resistance gene with a chloramphenicol-resistance gene and the replacement of the pBR322 replicon with a p15A replicon). We used a tetracycline-controlled promoter as it allows tight regulation of protein expression upon anhydrotetracycline titration[59]. We chose p15A as the replication origin since it provides low copy number of vectors in the cell[60].

Below is the DNA and protein sequence of the wild-type mScarlet. The positions mutated to TAA, TAG, and TGA are highlighted in bold. The N-terminal strep-tag and the C-terminal His-tag are marked in blue and red, respectively. Downstream of the His-tag, we introduced two stop codons.

**DNA sequence.**

```
ATGGCGAGCGCGTGGAGCCACCCGCAGTTCGAAAAAATGGTGAGCAAAGGCGAAGCGGTGATTAAAGA
ATTTATGCGCTTTAAAGTGCACATGGAAGGCAGCATGAACGGCCATGAATTTGAAATTGAAGGCGAAG
GCGAAGGCCGCCCGTATGAAGGCACGCAGACGGCGAAACTGAAAGTGACGAAAGGCGGCCCGCTGCCG
TTTAGCTGGGATATTCTGAGCCCGCAGTTTATGTATGGCAGCCGCGCGTTTACGAAACATCCGGCCGGA
TATTCCGGATTATTATAAACAGAGCTTTCCGGAAGGCTTTAAATGGGAACGCGTGATGAACTTTGAAG
ATGGCGGCGCGGTGACGGTGACGCAGGATACCAGCCTGGAAGATGGTACCCTGATTTATAAAGTGAAA
CTGCGCGGCACGAACTTTCCGCCGGATGGCCCGGTGATGCAGAAAAAAACGATGGGCTGGGAAGCGAG
CACGGAACGCCTGTATCCGGAAGATGGCGTGCTGAAAGGCGATATTAAAATGGCGCTGCGCCTGAAAG
ATGGCGGTCGCTATCTGTACGCGGATTTTAAAACGACGTATAAAGCGAAAAAACCGGTGCAGATGCCGGGC
GCGTATAACGTGGATCGCAAACTGGATATTACGAGCCATAACGAAGATTAACGGTGGTGGAACAGTA
TGAACGCAGCGAAGGCCGCCATAGCACGGGCGGCATGGATGAACTGTATAAACTCGAGCACCACCATC
ACCATCACCATCACTGATAA
```

**Protein sequence.**

```
MASAWSHPQFEKMVSKGEAVIKEFMRFKVHMEGSMNGHEFEIEGEGEGRPYEGTQTAKLKVTKGGPLP
FSWDILSPQFMYGSRAFTKHPADIPDYYKQSFPEGFKWERVMNFEDGGAVTVTQDTSLEDGTLIYKVK
LRGTNFPPDGPVMQKKTMGWEASTERLYPEDGVLKGDIKMALRLKDGGRYLADFKTTYKAKKPVQMPG
AYNVDRKLDITSHNEDYTVVEQYERSEGRHSTGGMDELYKLEHHHHHHH**
```

### *E. coli* strain and media

Plasmids encoding the reporters for SCR events were electro-transformed in the wild-type K-12 MG1655 *E. coli* strain. Transformants were grown on LB-agar plates and inoculated into 384-well plates containing LB media supplemented with 15 μg/mL of chloramphenicol. To store the transformants at −80 °C (glycerol stock), they were grown at 37 °C to saturation, and glycerol was added until a final concentration of 20%.

To investigate the effect of temperature on SCR, *E. coli* cultures were grown into 384-well plates without shaking under saturated humidity conditions at 18, 25, 37, and 42 °C to saturation. To address the nutrient depletion effect on SCR, LB and M9 media were tested.

To titrate the expression of the Gly95-TGA reporter, 0, 25, 50, 100, 200, 400, and 800 μg/L of anhydrotetracycline was added to the media. For the library study, to induce the expression, 400 μg/L of anhydrotetracycline was added to the media. Cells were grown under light protection to avoid the photodegradation of the anhydrotetracycline.

**M9 standard media**. M9 media was supplemented with 0.4% glycerol, 0.2% casamino acids, 1 mM thiamine hydrochloride, 2 mM MgSO$_4$, and 0.1 mM CaCl$_2$.

**M9 supplemented with higher carbon source concentration**. M9 media was supplemented with 1.6% glycerol, 0.2% casamino acids, 1 mM thiamine hydrochloride, 2 mM MgSO$_4$, and 0.1 mM CaCl$_2$.

**M9 supplemented with higher casamino acid concentration**. M9 media was supplemented with 0.4% glycerol, 0.4% casamino acids, 1 mM thiamine hydrochloride, 2 mM MgSO$_4$, and 0.1 mM CaCl$_2$.

### Microscopy Screenings and Sample Preparation
A liquid handling robot (Beckman Coulter Biomek FXp with Thermo Cytomat 6002) was used to perform plate-to-plate transfers of cells. The cells were inoculated from a glycerol stock into 384-well plates (Eppendorf microplate 384/V #0030621301) containing 100 μL of LB-media supplemented with 15 μg/mL of chloramphenicol. Pre-culture plates were then grown until cells reached saturation (~24 h) at 37 °C. Then, 2 μL of culture from the saturated cultures were used to inoculate the 384-well plates (Eppendorf microplate 384/V #0030621301) containing 100 μL of the appropriate media (LB or M9) supplemented with 15 μg/mL of chloramphenicol and 400 μg/L of anhydrotetracycline. The cells were grown at different temperatures (18, 25, 37, and 42 °C) at constantly controlled temperatures until reaching saturation (24-48 h) under light protection. Then, cells from the saturated cultures were transferred into Greiner 384-well glass-bottom optical imaging plates (#781092) previously coated with poly-L-lysine containing 50 μL of PBS. We implemented different dilution steps in response to variations in growth rates (Supplementary Table 8 and Supplementary Data 3): 1:2500 for 37 °C and 25 °C, 1:1000 at 42 °C, and 1:500 at 18 °C for LB media, and 1:1250 for 37 °C and 25 °C, 1:500 at 42 °C, and 1:100 18 °C at M9 media. To coat the Greiner 384-well glass-bottom optical imaging plates, 50 μL of 0.01% (w/v) of poly-L-lysine (SIGMA, #P4832) was added and incubated for at least 1 h. Then, the plates were washed with water and dried overnight.

All confocal imaging was performed on an automated spinning disc confocal microscope (CellVoyager CV7000, Yokogawa), using a 60×1.2NA objective. For excitation, a 561 nm laser was used, and fluorescence was detected through a 600/37 bandpass emission filter. We recorded 2560×2560 16-bit images at binning 2 for brightfield and for mScarlet fluorescent protein in epifluorescence mode on SCROS cameras. A laser-based hardware autofocus was used to acquire 5 x to 9 images per well.

### Fluorescence data analysis
Image analysis was performed with Fiji[61], and downstream data analysis and visualization were performed using R v4.1.2.

The cells were identified and segmented, and their fluorescent signal (mean and standard deviation, mode, minimum, and maximum), as well as additional cell properties (area, x- and y-coordinates), were determined in Fiji using custom macros[61].

### Statistical analysis
**Friedman and sign test**. We employed the Friedman and sign tests to evaluate the statistical differences among the biological replicas depicted in Fig. 2C. These tests are non-parametric and suitable for non-normally distributed data. The Friedman test is ideal for comparing more than two samples, while the sign test is suitable for comparing two samples. To satisfy the requirements of these tests, which demand equal-sized samples, we determined the maximum number of cells available across replicas. Replicas with fewer than 200 cells were excluded from the analysis (Supplementary Table 1).

**Wilcoxon test**. To determine whether one distribution significantly exceeded another, we performed the Wilcoxon test. This non-parametric method is suitable for analyzing non-normally distributed data, serving as an alternative to the *t*-test. Similar to the Friedman and sign tests, the Wilcoxon test necessitates equal-sized replicates. We evaluated the maximum number of cells available across replicas and excluded those with fewer than 200 cells. Then, we pooled the replicas together as a dataset, ensuring equal representation from each replica to prevent bias in the results.

### Western blot
Single-colony *E. coli* LB-cultures were grown at 37 °C until reaching an absorbance of 0.6 at 600 nm. Next, they were grown at 18 °C for 2 h. Protein expression was induced by adding anhydrotetracycline to a final concentration of 400 μg/L, and cells were grown overnight at 18 °C. Aliquots of ~1 mg of total proteins (A$_{600nm}$ = 1 ≈ 0.3 mg/mL) were centrifuged. Cell pellets were resuspended in 400 μL of ice-cold disruption buffer (PBS containing 10% Glycerol, 1 mM MgSO$_4$, Benzonase 0,05U/m, Roche complete cocktail EDTA free 1 tablet/10 mL) and 300 mg glass beads (0,1 mm Scientific industry SI-BG01) were added. Cells were disrupted in FastPrep-24™ (MP Biomedicals) at low temperatures and centrifuged. Supernatants were processed by SDS–polyacrylamide gel electrophoresis (4–20% Tris-Glycine-Gel, Anamed #TG 42015).

All gels contained the lysate of cells carrying the mScarlet (PC), an empty vector (NC), and a protein marker. The mScarlet and marker bands were visualized with a Typhoon 9500 at 532 nm directly from the gel. The rest of the proteins were transferred to nitrocellulose membranes (Whatman BA85), performing semi-dry blotting (Transfer buffer: 20 mM Tris-Base, 160 mM Glycine, 0, 1% SDS, 20% Methanol). From the membranes, total protein quantities were assayed using Fast Green FCF (Sigma, #F7252) staining solution[62] and imaged using LICOR Odyssey 700 nm. For immunodetection, the membranes were blocked with Blocking buffer (5% milk, 0, 1% Tween20 in PBS) followed by incubation with 1:5000 Qiagen mouse anti-penta His-tag (# 34660) and 1:10.000 Licor IRDYE 800 goat anti-mouse (#926-32350). Signals were detected with LICOR Odyssey and analyzed with Fiji[61].

The mScarlet sample (PC) shows in-gel fluorescence in several bands with a molecular weight of around 26 kDa, probably due to degradation. The area selected to analyze the His-tag expression was where the PC presented fluorescence (Supplementary Fig. 7).

### Analysis of the genetic context effect
**Stop codon readthrough likelihood score (SCR score)**. Those reporters displaying a median fluorescence below a threshold (defined as the median plus two standard deviations of the fluorescence signal of the NC) received a score of 0. The reporter with the highest fluorescence signal was assigned the highest score, with a value of 1.0. All other reporters were ranked between 0 and 1.0 according to their median fluorescence values. We repeated this ranking strategy for all stop codons at all tested conditions. Then, we calculate the average score among conditions and the three stop codons (24 values per position) to have a unique score per position and, therefore, per genome context.

**Correlation between nucleotide content and stop codon readthrough likelihood score**. The stop codon readthrough likelihood

scores were binned in 8 equal-width bins. The number of samples per bin are: $N = 12$ (bin = 0), $N = 12$ (0<bin<0.125), $N = 9$ (0.125<bin<0.250), $N = 2$ (0.250<bin<0.375), $N = 1$ (0.375<bin<0.5), $N = 0$ (0.5<bin<0.625), $N = 1$ (0.625<bin<0.750).

**Correlation between the nucleotide identity adjacent to the stop codon and stop codon readthrough likelihood score.** The stop codon readthrough likelihood scores were binned into three categories: i) accurate protein synthesis termination (score = 0, $N = 12$), ii) medium tendency to SCR (0 <score <mean of all scores, $N = 16$), and iii) high tendency to SCR (score > mean of all scores, $N = 9$).

**Protein purification**
Wild-type mScarlet for the stability assay, for the calibration curve, and the samples to study by mass spectrometry were expressed in *E. coli* and purified with His-tag affinity chromatography. Briefly, genes encoding these proteins with C-terminal 8x His-tag were cloned into a pASK vector under a tetracycline-controlled promoter. Vectors were transformed into K-12 MG1655 *E. coli* cells. From this point, two procedures were used to grow the cells and induce protein expression: i) To purify the samples for mass spectrometry analyses, cells were grown on LB-medium supplemented with 15 μg/mL of chloramphenicol at 37 °C until reaching an absorbance of 0.2 at 600 nm, then at 18 °C until reaching absorbance of 0.6 at 600 nm. Protein expression was induced by anhydrotetracycline (final concentration of 400 μg/L), and cells were grown overnight at 18 °C (-12 h). ii) To purify the wild-type mScarlet for the stability assay and calibration curve, cells were grown on LB-medium supplemented with 15 μg/mL of chloramphenicol at 37 °C until reaching an absorbance of 0.6 at 600 nm. Then, protein expression was induced by the addition of anhydrotetracycline (final concentration of 400 μg/L), and cells were grown for 4 h at 37 °C.

Cells were harvested by centrifugation and resuspended in lysis buffer comprising 20 mM sodium phosphate, 500 mM NaCl, and 20 mM imidazole, pH 7.4, supplemented with Roche complete cocktail EDTA free 1 tablet/10 mL and benzonase 0,05U/m. Cells were lysed using an LM20 microfluidizer (Microfluidics), and the lysates were clarified by centrifugation for 1 h at 4 °C at 12000 rpm and loaded on a His GraviTrap™ column (GE Healthcare). After washing with 10 mL of washing buffer (40 mM sodium phosphate, 500 mM NaCl, and 20 mM imidazole, pH 7.4), proteins were eluted with 20 mM sodium phosphate, 500 mM NaCl, and 500 mM imidazole at pH 7.4[63].

The purity of all the samples was assessed by SDS–polyacrylamide gel electrophoresis (Life Technologies GmbH), and protein concentration was determined by absorbance at 280 nm (using the extinction coefficient of the wild-type mScarlet for all the samples, 39880 M⁻¹cm⁻¹). For the mass spectrometry analyses, we separated 50 μg of each sample by SDS-PAGE. For the thermostability assay and the calibration curve, mScarlet was purified and dialyzed against PBS.

**Calibration curve for fluorescence measurements**
Triplicates of purified mScarlet were fluorescently imaged at 0, 3, 4, 7, 10, 15, 23, 35, 52, 78, 117, 175, 262, 393, 590, 885, 1327, 1991, and 2986 nM in PBS. Saturation of the fluorescence signal above 70000 AU defines the upper limit of the dynamic range (Supplementary Fig. 1). A linear relationship between mScarlet concentration and fluorescence arbitrary units was observed between 40 and 800 nM.

**mScarlet thermostability assay**
Aliquots of purified mScarlet at 60 nM in PBS were incubated at a range of temperatures (from 30.5 °C to 98.2 °C) for 30 min. Samples were then fluorescently imaged. The procedure was performed in triplicates.

Unfolded mScarlet is not functional and, therefore, does not exhibit fluorescence. On the contrary, mScarlet that remains folded exhibits fluorescence. The thermostability curve (Fig. S3C) indicated that mScarlet remained functional, i.e., fluorescent, until 70 °C.

**mRNA secondary structure prediction**
We used RNAfold2[53] to calculate the optimal RNA secondary structure that has the minimum free energy at the experimentally tested temperatures. We predicted the minimum free energies of the possible secondary structures in a 100-nt window downstream and upstream of the inserted stop codon. Based on Chen et al.[64], we defined a 6-nt spacer downstream of the stop codon, which is typically occupied by the ribosome. No spacer was considered upstream of the stop codon (Supplementary Figs. 11A, 9B, C).

We then focused on the local thermodynamic stability of the secondary structures surrounding the premature stop codon. We studied the number of any base pairs, G-C pairs, and the longest stretch of consecutive pairs in a set of windows (5, 10, 20, and 50-nt) upstream (Supplementary Fig. 11D) and downstream (Supplementary Fig. 11E) of the premature stop codon of the most stable RNA structure previously predicted (100-nt window upstream and 100-nt downstream of the premature stop codon plus the 6-nt spacer).

**Mass spectrometry analysis of reporters**
Gel regions corresponding to the molecular weight of the reporters were excised and analyzed by LC-MS/MS. Briefly, samples were in-gel digested with trypsin (sequencing grade, Promega, Mannheim), the resulting peptides extracted by two changes of 5% formic acid (FA) and acetonitrile, and dried down in a vacuum centrifuge. Peptide pellets were dissolved in 100 μL of 5% FA and 5 μL aliquot of peptide mixture and taken for MS analysis.

LC-MS/MS analysis was performed on a nanoUPLC Vanquish system interfaced online to an Orbitrap HF hybrid mass spectrometer (both Thermo Fischer Scientific, Bremen). The nano-LC system was equipped with Acclam PepMap™ 100 75 μm x 2 cm trapping column and 50 cm μPAC analytical column (Thermo Fischer Scientific, Bremen). Peptides were separated using a 75 min linear gradient, solvent A−0.1% aqueous FA, solvent B − 0.1% FA in acetonitrile. Samples were first analyzed using data-dependent acquisition (DDA) and then by targeted acquisition with an inclusion list guided by the results of RNA-seq analysis. DDA analysis was performed using the Top20 method; precursor m/z range was 350−1600; mass resolution (FWHM)−120 000 and 15 000 for MS and MS/MS spectra, respectively; dynamic exclusion time was set to 15 s. The lock mass function was set to recalibrate MS1 scans using the background ion (Si(CH3) 2 O)6 at m/z 445.1200. Targeted analysis was performed in profile mode; a full mass spectrum at the mass resolution of 240 000 (AGC target 3×10⁶, 150 ms maximum injection time, m/z 350−1700) was followed by PRM scans at a mass resolution of 120 000 (AGC target 1×10⁵, 200 ms maximum injection time, isolation window 3 Th) triggered by a scheduled inclusion list. To avoid carryover, 3-5 blank runs were performed after each sample analysis, the last blank was recorded and also searched against a customized database.

Spectra were matched by MASCOT software (v. 2.2.04, Matrix Science, UK) against a customized database comprising *E. coli* protein sequences extracted from a UniProt database (version October 2022) and a set of modified mScarlet sequences. mScarlet sequences included three-frame translated nucleotide sequences with stop codon insertions, sequences with deletion of 1-2 amino acids surrounding the position of the stop codon that were denoted as "X" (equivalent to any amino acid). Database search was performed with 5ppm and 0.025 Da mass tolerance for precursor and fragment ions, respectively; enzyme specificity−trypsin; one miscleavage allowed; variable modifications− methionine oxidation, N/Q deamidation, cysteine sulfonic acid, cysteine propionamide, peptide N-terminal acetylation. The results

were then evaluated by Scaffold software (v.4.11.1, Proteome Software, Portland) and also manually inspected. Identification of modified peptides was accepted if it passed the 95% peptide probability threshold and if the matched fragmentation spectra (minimal number of PMS: 2) comprised fragment ions, unequivocally confirming the misincorporated amino acid (Supplementary Fig. 12). The ratio of peptide forms comprising misincorporated amino acids was estimated based on extracted ions chromatograms (XIC) for each form generated in the Xcalibur software (Thermo Fischer Scientific) and normalized to the sum intensity of all forms of the peptide. Peptides with misincorporated Lys were excluded from the calculations.

## RNA-seq

Single-colony *E. coli* LB-cultures were grown at 37 °C until an absorbance of 0.4 at 600 nm. Then, they were grown at 18 °C until an absorbance of 0.6 (~2 h), protein expression was induced by the addition of anhydrotetracycline to a final concentration of 400 μg/L, and cells were grown overnight at 18 °C. Cells were diluted to an absorbance of 1.0 at 600 nm, and 100 μL of the cell suspension was mixed with 200 μL of RNAprotect bacteria reagent (Qiagen #76506). Cells were harvested by centrifugation, and pellets were frozen. Then, the total RNA of the samples was extracted according to manufacturing specifications (RNeasy Protect Bacteria Mini Kit Qiagen kit).

To detect rare transcriptional error events, we optimized i) the reverse transcription reaction, ii) the cDNA amplification reaction, and iii) the PacHifi sequencing step. Such optimization enabled us to achieve >99.99% accuracy. Briefly:

i) 300 ng of total RNA per sample was mixed with 50 ng of random hexamers and dNTPs and hybridized at 65 °C for 5 minutes. cDNA was reversely transcribed using the Thermo Fisher Superscript IV transcriptase according to the manufacturer's instructions (reporter median error frequency of $5.01*10^{-5}$[65]).

ii) Then, mScarlet was specifically amplified from the resulting cDNA with forward primer 5' AGTTATTTTACCACTCCCTATCAGT 3' and reverse primer 5' AGTAGCGGTAAACGGCAGAC 3', resulting in an amplified PCR fragment of 948 bps. The NEB Q5® High-Fidelity DNA polymerase was used according to the manufacturer's instructions. This DNA polymerase is one of the highest fidelity polymerases incorporating, according to the manufacturer, 1 error in 28.000.000 base pairs). PCR conditions were: an initial denaturation step for 30 sec at 98 °C, 30 cycles of 10-sec denaturation at 98 °C followed by annealing at 67 °C for 30 sec, and extension at 72 °C for 30 sec. A final extension step was performed for 2 min at 72 °C. Primer and dNTPs have been removed with 1x volume AMPure bead purification. Amplified mScarlet fragments were finally quantified with the Thermo Fisher Qubit high sensitivity DNA quantification system. 1 ng of the amplified mScarlet fragments were analyzed on the Agilent Fragment Analyzer system using the NGS high sensitivity kit.

iii) Pacbio SMRTbell® libraries have been generated following the PacBio® Barcoded overhand adapters for multiplexing amplicons (Express template kit 2.0). For each multiplexed library, eight samples have been pooled equimolarly. Briefly, 50 ng of each amplified mScarlet fragment was damage repaired, followed by end repair and A-tailing according to the instructions. Pacbio barcoded overhang adapters (BAK8A and BAK8B) were ligated to the PCR fragments and equimolarly pooled prior to two final AMPure bead purification steps (1x volume). The final quality control of the resulting library was performed on the Agilent Fragment Analyzer with the large fragment kit. We generated, in total, two different 8plex PacBio HiFi libraries. The v4 PacBio sequencing primer, in combination with the SEQUEL II binding kit 2.1, was used to sequence both eightplex PacBio SMRTbell® libraries. 80 pM and 120 pM of each library were loaded by diffusion loading, pre-extension time was 0.3 hours, and run time 10 hours on the SEQUEL II, making use of the SEQUEL II sequencing 2.0 chemistry. Circular consensus reads have been called with the PacBio SMRT link

ccs calling tool and demultiplexed with lima, the PacBio demultiplexer, and primer removal tool (https://lima.how/). During the Pacbio HiFi library sequencing, we obtained reads up to 100 kB. Since our cDNA fragments are short (~1 kB), most of them have been read multiple times during the circular sequencing steps (quantified as the number of passes). Specifically, only ~15% of all reads had less than 10 full passes, and the maximum number of passes was 60. The standard PacBio SMRT link pipeline for circular consensus sequencing, by default, considers only reads with an error rate of less than 1%. Additionally, based on Wenger et al., 10 passes correspond to 99.9% base accuracy, and 20 passes correspond to 99.99% base accuracy, in agreement with our empirical error rate observed (Supplementary Fig. 15, mean = 0.005% mismatch per base).

The long PacBio RNA-seq reads were mapped to the reference with BWA v0.7.17-r1198. BAM files representing the mapped reads were further processed with Samtools mpileup v1.15.1 using htslib 1.15.1 to generate a textual description of the mapped reads, including information about positions and read mutations, insertions, deletions, and indels found (all BAM files are provided as Data S2). The pileups were further processed with R to extract information on frequencies of mutated trimers along the mScarlet gene. For extracting information on synonymous and non-synonymous mutations, PySam v0.16.0 with Python 3.7.12 was used with the BAM/SAM output from BWA. All plots were generated with base plotting of R v4.1.2.

## Analysis of *E. coli* proteome by mass spectrometry

**Cell culturing and sample preparation.** *E. coli* (strain K-12 MG1655) was grown overnight at 37 °C on LB medium, then split into two parts, one incubated at 37 °C, the other at 18 °C until reaching OD600 of ca. 0.6. The experiment was performed in three biological replicates, and samples were prepared and measured in a block-randomized fashion[66] with two technical repeats. Cells were pelleted by centrifugation at 4500 g for 10 minutes, washed with PBS, and resuspended in 1 ml of lysis buffer consisting of 8 M Urea, 0.1 M ammonium bicarbonate, 0.1 M NaCl, and 1x Roche cOmplete™ Protease Inhibitor Cocktail (Roche Diagnostics Deutschland GmbH, Germany). Then, 2 micro spatula spoons of 0.5 mm stainless steel beads (Next Advance Inc., USA) were added, cells were lysed in a TissueLyser II (QIAGEN GmbH, Germany) for 2x 5 min at 30 Hz at 4 °C, and the debris removed by centrifugation for 10 min at 13,000 g. Protein concentration in the supernatant was measured by Pierce BCA Protein Assay (Thermo Scientific, USA), and aliquots of 100 μg of proteins were taken for LC-MS/MS analysis. After reduction and alkylation, proteins were precipitated with isopropanol[67] and digested overnight at 37 °C with a 1:50 enzyme: protein ratio by Trypsin/Lys-C Mix (Promega GmbH, Germany). The resulting peptides were desalted on a MicroSpin column (The Nest Group, Inc., USA) and dried down in a vacuum concentrator. Prior to mass spectrometric analysis, the samples were reconstituted in 0.2% aqueous formic acid; peptide concentration was determined by measuring absorption at 280 nm and 260 nm using a Nanodrop 1000 ND-1000 spectrophotometer (Thermo Fisher Scientific Inc, USA) and the Warburg-Christian method[68] and adjusted to a final concentration of 0.12 μg/μL; 5 μL were then taken for analyses.

**LC-MS/MS analysis.** LC-MS/MS analysis was carried out on the mass spectrometric equipment described in the 'Mass spectrometry analysis of reporters' section. Peptides were separated using 120 min 2-sloped gradient, solvent A − 0.1% aqueous FA, solvent B − 0.1% FA in acetonitrile: 80 min 0 to 17.5 % ACN, 40 min 17.5 % to 35% ACN at a flow rate of 0.5 μl/min. The gradient was followed by a 7 min wash with 95% ACN. To avoid carryover, 2 blank runs were performed after each sample analysis. Further settings: spray voltage − 2.5 kV, capillary temperature − 280 °C, S-lens RF value − 50. Spectra were acquired by Data Independent Acquisition (DIA) in a staggered fashion[69,70]; full-scan mass spectrum with a mass range of 395-971 m/z and resolution of 60,000 (AGC

3e6, 40 ms maximum injection time, fixed first mass 100) was followed by 32 MS2 scans in centroid mode at a 30,000 mass resolution with an isolation window of 18 m/z covering a 400-966 m/z mass range (55 ms maximum injection time, AGC 1e6, normalized collision energy 24).

**Database search and validation of candidate peptides.** Acquired spectra were matched against a customized database using DIA-NN software suit v1.8[71]. A customized database, including sequences resulting from potential stop codon readthrough (SCR) events, was created based on the *E. coli* K12 MG1655 reference genome and the corresponding genome annotation in the NCBI database (accession GCF_000005845.2, version from 18/05/21). For each gene, the 'downstream sequence' was determined by finding the next in-frame ORF after that stop codon in the genome (using the reverse complement for genes on the negative strand). The minimum length of the downstream sequence was 60 nucleotides, and the maximum was either 300 nucleotides or the distance to the next in-frame ORF. Genes were then in silico translated 20 times from canonical start to the end of the downstream sequence or until the second in-frame stop codon, each time translating the canonical (the first) stop codon with a different amino acid, and added to the fasta file. Thus, for each gene, the sequence list contained 20 sequences, each encompassing the canonical and genomic downstream sequence, with one of the 20 amino acids in place of the canonical stop. This file, as well as a file containing common contaminant sequences[72], were used as input for DIA-NN to create the spectral library. Database preparation and data analysis were performed with Python 3 and Jupyter Notebook, using pandas[73], numpy[74], and Biopython[75].

Staggered DIA raw files were converted to.mzML format and demultiplexed using Proteowizard MSConvert v3.0.2[76] and analyzed with DIA-NN v1.8. The predicted spectral library was generated from sequences in a customized database under the following settings: enzyme - Trypsin/P, one missed cleavage allowed; C(carbamidomethyl) as fixed modification; M(oxidation) and N-terminal M-excision as variable modifications, allowing up to one variable modification per precursor. Precursor and protein group matrices were filtered at 1% FDR. Other settings were: –double-search –smart-profiling –no-ifs-removal –no-quant-files –report-lib-info –il-eq.

The precursor-by-sample-matrix output generated by DIA-NN was processed further to identify SCR events. First, all precursors mapped to a contaminant protein or any sequence in the Swissprot database (version from 22/12/21) were discarded. The remaining precursors mapped to the downstream sequence of an *E. coli* gene either fully (complete sequence is after the stop codon) or partially (overlapping the stop codon) and were treated as candidate SCR precursors. Singly charged precursors containing lysine or arginine were removed, as well as precursors not extending for at least 2 amino acids over the stop codon. Further, only candidate SCR precursors reported in 3 out of 6 replicates for at least one temperature value were retained. Fragmentation spectra of candidate SCR precursors were then manually validated in Skyline 'daily', version 22.2.1.351[77]. For this, the spectral library generated by DIA-NN based on observed fragment intensities (with 'full-profiling' library generation setting) and the peak boundaries from the DIA-NN main output table were imported into Skyline, along with sequences of SCR precursors remaining after filtering. Candidates were evaluated by comparison of the empirical library spectra generated by DIA-NN against Prosit-predicted spectra[78] and by assessing chromatographic coelution of fragments. Peptide identification was accepted if i) peptide sequence matched at least three fragments (y3 and higher for tryptic peptides, b3 and higher or a combination of b/y-ions for others); ii) the fragments co-eluted; iii) two fragments predicted by Prosit to be the most intense (excluding y1, y2, b1, and b2) had to be matched. In addition, indexed retention times predicted by Prosit were compared to measured (indexed) retention times, and outliers eluting much earlier or later than

predicted were removed from the analyses (Supplementary Fig. 18). Peptides meeting all the filters described were further considered for analyses of differential abundance, sequence context, and stop codon usage. To identify statistically significant differences in abundance between the 37 °C and 18 °C conditions, a Student's two-sided *t*-test with Benjamini-Hochberg adjustment[79] was carried out on the log2 peptide intensities of the two groups. In cases where multiple precursors represented the same peptide sequence, the respective intensities were summed.

## Statistical analysis and data visualization
Statistical analysis was performed by R v4.1.2. For all box plot representations, the thick black line indicates the median, the box indicates the 25th and 75th percentiles, and the whiskers indicate 1.5 times the interquartile range. Figures were created in Adobe Illustrator and Fig. 1A and D were created with Biorender (biorender.com) and released under a Creative Commons Attribution-NonCommercial-NoDerivs 4.0 International license.

## Reporting summary
Further information on research design is available in the Nature Portfolio Reporting Summary linked to this article.

## Data availability
The mass spectrometry data have been deposited in the MassIVE repository under accession number MSV000091065 (https://doi.org/10.25345/c5tm72991) and to the ProteomeXchange repository under accession number PXD039448. RNA-seq data have been deposited in NCBI's Gene Expression Omnibus and are accessible through GEO Series accession number GSE226936 and are provided as Supplementary Data 1 (BAM files) and their DNA-seq chromatograms as Supplementary Data 2 (bl1 files). The raw fluorescent data obtained from microscopy images and the Western blot gels are provided as Source data files. Source data are provided with this paper.

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

## Acknowledgements

We thank Andrej Shevchenko for his support and advice with mass spectrometry. We thank Lena Hersemann, Noreen Walker, and Andre Gohr from the Scientific Computing Facility of the MPI-CBG for helping with RNA-seq data and image analyses; Marc Bickle and Martin Stöter from the Technology Development Studio for guidance and assisting with the automatic microscope; Barbara Borgonovo, Eric Geertsma and Aliona Bogdanova from the Protein Biochemistry Facility of the MPI-CBG for helping with the western blot experiment; Julia Jarrells from the Cell Technology Facility of the MPI-CBG for the RNA extraction preparation and Sylke Winkler and Nicola Gscheidel from the Sequencing and Genotyping Facility of the MPI-CBG for the RNA-seq experiments. We thank Michele Marass and Miri Trainic for their help with scientific writing and editing.

## Author contributions

M.L.R.R. and A.T.P. designed the project. M.L.R.R., A.K., D.R. performed experiments. J.P., T.J. and A.S. performed mass spectrometry experiments and analyses. M.L.R.R. analysed the data and prepared all the figures. M.L.R.R., A.T.P., J.P., A.S. interpreted the data. M.L.R.R. and A.T.P. wrote the manuscript with the help of all co-authors. A.T.P. acquired funding.

## Funding

## Competing interests

The authors declare no competing interests.
