## [Peer Review File · Nature Communications]

Environment modulates protein heterogeneity through transcriptional and translational stop codon readthroughEditorial Note: Parts of this Peer Review File have been redacted as indicated to remove third-party material where no permission to publish could be obtained.

REVIEWER COMMENTS

Reviewer #1 (Remarks to the Author):

The paper by Romero et al describes a systematic study on the stop-codon readthrough efficiency in *E. coli*. Using mScarlet as reporter, they analyze stop codon miscoding efficiency (SCM) at stop codons randomly introduced at the internal positions of the mScarlet gene. Their readout is fluorescence, which can only appear when the stop codon is misread as sense codon. They identify the UGA codon as the leakiest stop codon and show that SCM is modulated by the nucleotide 3' from the stop codon. Furthermore, using mass spectrometry they show that UGA is mostly misread by Trp-tRNA, whereas UAG can be misread by other tRNAs. These experiments are mostly fine, however, there is nothing new here, all this is known and has been shown in numerous publications for bacteria, yeast and mammals.

Two further experiments are more interesting. First, the authors investigate how growth conditions (temperature and nutrition) affect SCM efficiency and show that readthrough is much higher at lower temperature and at conditions of nutrient scarcity. The effect of starvation is not surprising, as nutrient scarcity is known to globally increase the misreading level. The temperature effect is interesting, however, the data seems to be inconsistent. Specifically, Fig 2B shows a relatively high misreading efficiency for the UGA codon at 18°C, almost none at 25° and then high again at 37° (Fig 2B). The authors do not comment on this surprising effect, but this seems to be a problem with data reproducibility. Why is there an empty space between the 25° and 37° bars, are these different sets of experiments? Also the results of the validation experiments shown in Figs 2E and F are not convincing. The misreading efficiency expressed as % of His-tag signal varies from 0 to 80%, whereas most of the fluorescence signals are low, below 20%, except for 3 outliers (94, 105 and 135) It is absolutely unclear how the authors manage to obtain a correlation coefficient of 0.7 based on these data. Then, they delete points that have a particularly large deviation using a 30% difference criterion (i.e., they exclude points that deviate by more than 30% in the two types of readout - why?). 30% is entirely an arbitrary value; at 15% they would have to exclude even more points and then the whole validation exercise becomes meaningless. Instead, they should have excluded all of the points where the signal is too close to 0 in both types of signals and then calculate the correlation coefficient. In my opinion, there is a very poor correlation between fluorescence and His-tag reported, which raises serious questions about the validity of the experimental approach.

Another interesting set of data is mass spec identification of the read-through products. There are very few, even at 18°C, and the SCM efficiency is very low (up to 1.35%; Fig. 5). This is consistent with the literature data, where the spontaneous readthrough efficiency has been estimated in the range of 0.2 to 2%, but strongly contradicts the authors' notion that SCM can reach 80% at 18° (Fig. 2).

Minor point:

l. 280: the authors state that they did not observe alternative-frame peptides, which is expected, because they use a 0-frame C-terminal tag for purification. It would be important to state that these results do not exclude that alternative-frame peptides do exist, as it is easy to overlook this.

Reviewer #2 (Remarks to the Author):

Review of Romero et al., submitted to Nature Communications (2023)

Introduction:

In this manuscript, Romero and colleagues have employed a reporter assay to investigate stop codon readthrough at many positions spread throughout a protein coding sequence. They utilize this

approach to measure the dependence of readthrough propensity on sequence position, stop codon identity, and several environmental conditions. Their analyses reveal some local sequence features correlated with extent of readthrough. They follow up with mass spectrometry and RNA-seq measurements to determine the relative contributions of transcriptional and translational errors to the observed miscoding in a subset of reporter constructs. Finally, the authors perform proteome-wide mass spectrometry measurements and compare the readthrough trends for about 30 *E. coli* proteins to their results from the reporter assay.

Overall, the major contribution of this work is the medium-throughput (by today's standard) reporter-based measurements of sequence position and sequence context effects on stop codon readthrough. The systematic nature of these measurements allows the authors to measure effects that would be difficult to uncover otherwise, including surprisingly high levels of miscoding (e.g., 80%) that can arise at specific sequence positions and under environmental conditions as simple as growth in minimal media. However, there are unfortunately several serious issues with this manuscript. These range from the authors' portrayal of the novelty of their approach and results, to the presentation of the data, to the analyses performed and conclusions drawn therefrom. Below, we outline several major critiques, followed by several more minor points.

Major points:

- 1.) In the Introduction (e.g., paragraph beginning at line 49), the authors reference prior work of Ballesteros et al. (EMBO J 2001), Meyerovich et al. (PNAS 2010), and Zhang et al. (PNAS 2020) as a basis to outline the novelty of their contributions herein. However, the referencing of prior work is misleading and incomplete:
 - o Zhang et al. 2020 also measured read-through at all three stop codons as well as multiple frameshift-prone sequences. Meyerovich et al 2010 similarly looked at TAA and TGA stop codons in their studies in another bacterial species, *B. subtilis*. Therefore, the measurement of readthrough effects at multiple stop codons in a given context (line 52) cannot be portrayed as a point of novelty in the present work.
 - o Similarly, the authors cite Rosenberger and Foskett (Mol Gen Genet 1981) as the inspiration for their reporter assay (lines 83-84). This is surely a classic work, but the assay used is almost identical to that presented in Meyerovich et al. 2010. This more recent example should be referenced when the assay is introduced.
 - o The authors also fail to acknowledge a body of prior work that has used ribosome profiling to explore stop codon readthrough proteome-wide. This includes Dunn et al. (eLife 2013) and Baggett et al. (PLoS Genet 2017). The latter is cited by the authors, but only in a cursory fashion in the Discussion, related to the mutated RF2 in *E. coli* strain K-12, and not related to the conclusions of this work on stop-codon readthrough proteome-wide in *E. coli*.
- 2.) In the paragraph beginning at line 132 and throughout the paper, the authors comment on the highest error rates occurring at TGA stop codons (a known phenomenon). However, the K-12 strain of *E. coli* used for all experiments presented here has a mutation in RF2, the release factor that recognizes TGA, known to have reduced hydrolysis activity (Uno et al., Biochimie 1996) and increased readthrough (Baggett et al., PLoS Genet 2017). This fact should be noted as soon as results with highest error rates at TGA codons are reported in the manuscript. This is especially important considering that the most striking examples measured and cited, e.g., an error rate of ~80% at position 135 at 18 °C, occur specifically at TGA stop codons.
- 3.) In Figures 2, 3, 4, and 5, there are numerous serious issues in the presentation of the data impeding understanding of the results and their significance. A few examples of this systemic issue:
 - o Throughout the manuscript, the figure captions are very minimal and are missing key details in how experiments were conducted, what is being plotted in each panel, or even which experiments / results the plots pertain to. The captions are simply not descriptive enough and would need to be re-written to address this issue.
 - o Fig 2A: the variable plotted in the heat map is never defined. Furthermore, the title of the legend is "log 10", but the axis is labeled in linear units (i.e., 100,10,1... vs. 2,1,0...).

- o Figs 2B, 2D: It is fundamentally unclear what is being depicted in these panels.
- o Fig 2E: the presentation of the data is confusing. What is the meaning of the numbers? What is the meaning of the vertical lines from these numbers to the two lines? What are the two (dashed vs solid) lines?
- o It is not clear what exactly is being shown in Fig 3D. Is this a purely qualitative table of findings? What are the underlying quantitative data from the MS experiments?
- o Figs 4C, 4D: these lack a caption entirely. The manuscript text also fails to clarify what is shown here. Therefore, the analysis and results represented here are entirely unclear.

4.) The authors briefly postulate that GC content downstream of the stop codon is correlated with readthrough frequency due to increased stability of secondary structures in this nearby region. This naturally raises the question of what kind of secondary structures might be relevant and why they might lead to more misreading of the stop codon. With only 43 sites in total, it would be reasonable to calculate predicted secondary structures in the vicinity of each tested premature stop codon (and potentially all sites in mScarlet). The MFE of such structures, and other features such as whether they are predicted to encapsulate the stop codon itself, might correlate with observed SCM. The authors should computationally predict secondary structures for each tested site and determine to what extent they might explain their findings. Indeed, the authors cite Chen et al. (NSMB 2013) on the point that downstream stem-loops containing GC pairs decrease ribosomal accuracy. Presentation of the corresponding secondary structures for the reporter constructs used here would help to assess if this mechanism is likely driving the results in this case and strengthen the paper.

5.) There are some important issues with the analysis related to Figure 4 which limit how these results should be interpreted:

- o The apparently surprising finding that higher transcriptional errors are observed specifically at premature stop codons (PTCs) is actually strongly influenced by Rho-mediated premature transcription termination in the context of PTCs. This causes increased degradation of mRNAs in which the PTC is not corrected by an error, and an in-cell "selection" for the mRNAs containing such transcriptional errors. Therefore, the authors cannot conclude that transcriptional error rates are elevated at PTCs or invoke an unknown mechanism related to ribosome-RNAP coupling. Instead, the elevated observation of errors at the stop codons should be explained as a consequence of Rho-mediated transcription termination.
- o The authors report error rates at the RNA level on the order of 0.01% to 0.15%. However, it is unclear how (or whether) such low error rates can be detected by the PacBio sequencing technology employed, which has recently reported error rates on the order of 0.2% with (Wenger et al., Nature Biotech 2019). The authors should discuss their choice of RNA sequencing technology, the expected sequencing error rates, and clearly state how they relate to the measured transcriptional error rates reported.

6.) The results of the bioinformatic analysis for additional downstream stop codons in Fig. 5B are clearly influenced by the defective RF2 variant found in the K-12 strain of *E. coli*, as noted above. This should be noted in the text and the authors should perform the same analysis with a B strain of *E. coli* (with an efficient RF2) to see if the frequency of additional stop codons after a "TGA" is decreased as would be expected.

7.) For Figure 5C and the associated text, it is initially strange to see that some proteins have more and similar read-through at 37 °C vs. 18 °C, in contrast to the reporter measurements. The authors should comment on this. A large contribution to this effect may be that there is (apparently) no normalization of each protein's expression levels, which likely vary with temperature, which confounds interpretations of the relative amounts of readthrough at different temperatures. The authors should note this as well, or normalize for this effect and explain how this is done.

Minor points:

- 1.) In the discussion of the results of Fig. 2E-F, the authors make a major point of achieving a near

"ideal" slope of 1 for their % fluorescence (main SCM readout) vs. % His-tag signal plots by excluding 4 "dark reporters" in Fig 2E. However, this overemphasizes the linearity of the results. The vast majority of points cluster near the origin of the plot, so few points are contributing much to the linear trend being fit. Essentially only 3 points drive the "near-ideal" linear fit (near ~40-80, ~50-100 in x,y), and this requires the removal of 4 data points. Thus, it is inappropriate to emphasize the slope of this fit; instead, what the authors have clearly shown is a positive relationship between the two variables, but not necessarily a clean linear one, based on the data at hand. (Happily, this change does not fundamentally alter the authors' results or their interpretation.)

2.) Meyerovich et al. (PNAS 2010) measured temperature-dependence of frameshifting to rescue GFP reporter fluorescence, and the conclusions reached were similar to those in the present work for stop codon miscoding (lower temperature was correlated with more frameshifting / readthrough). This suggests a common mechanism may be at play. The authors should discuss the similarity of these results and potential implications.

3.) "Minimal media" is misspelled in Figure 2.

4.) In discussing the extent to which transcriptional errors, as opposed to misreading at the translational level, might drive SCM (section starting at line 284), the authors should cite previous work such as Dunn et al. (eLife 2013) which previously showed by ribosome profiling that readthrough events translatome-wide are primarily not driven by sequence changes at the mRNA level. They should also cite the literature on transcriptional error rates and translational error rates such as Li and Lynch, eLife 2020, and Meyerovich et al., PNAS 2010 which suggest a roughly order-of-magnitude difference in these frequencies.

5.) The similarity in the trend of SCM between low-temperature and low-nutrient conditions is noted but not discussed substantially. However, this suggests some underlying mechanism common to these two cases. The authors might discuss this.

6.) SCM "hotspots" are temperature-dependent in nontrivial ways, e.g., several reasonably highly-miscoded sites in mScarlet at 37 oC are no longer miscoded at 25 oC, and then re-emerge as potential "hotspots" at 18 oC (Fig. 2A). The authors should comment on this more complex temperature-dependence and its possible origins. Could this be evidence of some elevation of SCM at 37 oC compared to 25 oC, for example, in contrast to the overall trend the authors emphasize?

Reviewer #3 (Remarks to the Author):

This study by Romero Romero et al looks at changes in stop codon miscoding events across different environmental conditions. The manuscript does a good job of showing that temperature and nutrition can impact miscoding events of a premature codon. This work also shows that while both transcriptional errors and ribosomal readthrough can lead to miscoding events, ribosomal readthrough is the main contributor to miscoding. Furthermore, they find that there is a correlation between the nucleotides after the stop codon and miscoding events. Overall, while this manuscript has made a lot of interesting observations, I believe it could be improved if they understood more of the miscoding events they are seeing.

While it is stated that "stop codon miscoding may occur with a rate as high as 80%, ... , suggesting that evolution frequently samples stop codon miscoding events." It is unclear how evolutionary relevant the high rate of miscoding they see for a premature stop codon is to endogenous stop codon miscoding. For one they see fewer miscoding events the closer they get to the end of the ORF. They never see >10% miscoding after amino acid position 155. Similarly, while they find significant

transcriptional misincorporation for premature stop codons, there is an almost unmeasurable amount of transcriptional misincorporation at canonical stop codons. This shows that the cell treats premature stop codons very differently than canonical stop codons.

It seems the manuscript could be strengthened if there was a better understanding of why there is such a difference between a premature stop codon and the canonical stop codon. Is it related to distance? If you added a GFP on to the end of Scarlet does that increase the miscoding at the canonical Scarlet stop codon? If this was true, this could mean that stop codons in multi-gene operons could increase readthrough during environmental stress and the gene products could become fused? Could this increase protein function efficiency?

The paper could also be improved if they moved beyond correlation for the effect of the nucleotides downstream of the stop codon and directly showed that mutating a G to T or T to G decreases or increases readthrough from their reporters.

There are a number of small fixes to the manuscript, especially in the Figures and Figure legends: Figure 2B and 2D subpanels should be labeled better.

Line 346 - "after TGA, the most error-prone stop codon, then after TGA, and then after TAA" The second mention of TGA should presumably be TAG.

Line 347 - "trend is maintained if we extend the search to 20-nt and 30-nt windows (Fig 5B)" but Fig 5B shows 15-nt, 30-nt, and 60-nt, not 15, 20, 30 as mentioned in the text.

Figure 5B - It is unclear what the blue, green, and purple colors mean

Figure 5C - The figure legend mentions 9 genes exclusive to 18C and 3 genes detected at 37C (I assume that should be exclusive to 37C), yet the figure only shows 5 exclusive to 18C and 2 to 37C.

Line 824 - The figure legend for Figure 4 is missing for Fig 4C,D. There should also be more description in the figure one what is being presented in Fig 4C,D.

Line 832 0 "TAG, as the last accurate stop codon" this should say TGA.

Response Letter to Reviewers

Reviewer #1 (Remarks to the Author):

1- The paper by Romero et al describes a systematic study on the stop-codon readthrough efficiency in E. coli. Using mScarlet as reporter, they analyze stop codon miscoding efficiency (SCM) at stop codons randomly introduced at the internal positions of the mScarlet gene. Their readout is fluorescence, which can only appear when the stop codon is misread as sense codon. They identify the UGA codon as the leakiest stop codon and show that SCM is modulated by the nucleotide 3' from the stop codon. Furthermore, using mass spectrometry they show that UGA is mostly misread by Trp-tRNA, whereas UAG can be misread by other tRNAs. These experiments are mostly fine, however, there is nothing new here, all this is known and has been shown in numerous publications for bacteria, yeast and mammals.

We agree that these observations have been made in the past, and we were pleased to see that our experiments recapitulated them thereby validating our experimental approach.

2- Two further experiments are more interesting. First, the authors investigate how growth conditions (temperature and nutrition) affect SCM efficiency and show that readthrough is much higher at lower temperature and at conditions of nutrient scarcity. The effect of starvation is not surprising, as nutrient scarcity is known to globally increase the misreading level. The temperature effect is interesting, however, the data seems to be inconsistent. Specifically, Fig 2B shows a relatively high misreading efficiency for the UGA codon at 18°C, almost none at 25° and then high again at 37° (Fig 2B). The authors do not comment on this surprising effect, but this seems to be a problem with data reproducibility. Why is there an empty space between the 25° and 37° bars, are these different sets of experiments?

We are pleased that the reviewer found the temperature effect interesting. We studied the entire library of reporters at 18°C (minimum growth temperature for *E. coli*), 25°C, 37°C (optimum growth temperature), and 42°C (maximum growth temperature). The reason why we chose 25°C at the intermediate temperature between 18°C and 37°C responded to environmental causes. Since the institute has a permanent 25°C room, we did not need to set up an incubator, and we saved energy. Each of the experiments at different temperatures was done individually because the experimental setup was too long to do it simultaneously, and, at each temperature, *E. coli* required different growth times to reach saturation.

As the reviewer points out, the experiment done with rich media at 25°C seems to reveal less number of reporters with SCM than the experiment done with rich media at 37°C. Thus, the likelihood of SCM followed the temperature trend: 18°C >> 25°C < 37°C >> 42°C. On the other hand, the experiments done with minimal media followed the temperature trend: 18°C > 25°C > 37°C > 42°C. Thus analysing the data globally, we concluded that low temperatures increased the likelihood of SCM events. That said, we agree with the reviewer that the inherent irreproducibility linked to *E. coli* cultures grown during different time scales and comparing the fluorescence output of experiments on different days must be better addressed.

We focused on a subset of reporters to address the reproducibility of the temperature effect. We selected the reporters with stop codons at positions 105 and 155 and studied three biological replicas of each reporter containing TAA, TAG, and TGA. In total, together with the positive and negative control, we grew 24 single colonies individually in rich media at different temperatures until saturation. This time, in order to better understand the temperature trend, we included an experiment at 30°C. Thus, we performed the experiment at 18°C, 25°C, 30°C, 37°C, and 42°C.

Then, we studied their fluorescence output and plotted the distribution of the mean fluorescence per cells (Fig. S4).

As temperature decreases, we confirm higher fluorescence output, i.e., higher SCM levels. Interestingly, we also observed high heterogeneity among cells from a clonal population at low temperatures. That is, at low temperatures, some cells are very prone to SCM, while others do not have any SCM events (shown by the wider distribution of fluorescence values observed, as well as by the representative images in Fig. S4b). This heterogeneity of Stop Codon Readthrough in single bacterial cells has been previously observed and suggested to play a role in facilitating adaptation to changing environments (Fan et al., 2017). We are currently investigating the causes of such heterogeneity among clonal populations and its phenotypic and evolutionary consequences.

On the other hand, the heterogeneity observed at low temperature challenges the quantification of the SCM. For example, at 18°C, the fluorescence distribution of the three biological replicas transformed with the reporter where a TGA is introduced at position 105 of the mScarlet has a median of fluorescence of 31%, 24%, and 65% relative to the median of fluorescence of the mScarlet (PC) (% values are shown next to the distributions). We used the relative median fluorescence throughout the manuscript to quantify SCM error rates as summary statistics, nevertheless, these numeric values do not represent the complexity of the distributions (they are not normal nor symmetric). In explanation, for the three biological replicas, a subpopulation of the cells has a median fluorescence of 100% relative to the PC and a subpopulation with less or no fluorescence at all. What makes a subpopulation grow more than the other remains unknown and is opening a new research line in our lab.

Despite this complex scenario, SCM events are consistent among biological replicas. SCM depends on external factors, such as temperature, and internal factors, such as the identity of the stop codon and the sequence context.

We added a paragraph to the results section highlighting the heterogeneity and its consequences (line 169): “The experiment at 25°C seems to be an outlier on the observed temperature trend because it presented fewer reporters with SCM than the one at 37°C (Fig 2A, 2B, and S2). However, most of the reporters that showed at 37°C SCM events did so with a low rate (below 1%, Fig S2).

To better address the temperature effect and to study the reproducibility among biological replicas, we focused on the reporters with TAA, TAG, and TGA at positions 105 and 155. We analysed three biological replicas of each reporter at 18°C, 25°C, 30°C, 37°C, and 42°C (Fig. S4A). This experiment confirmed higher SCM levels as the temperature decreased. Interestingly, we also observed higher heterogeneity among cells from a clonal population at low temperatures (shown by the wider distribution of fluorescence values observed, as well as by the representative images in Fig. S4B). At low temperatures, some cells are very prone to SCM, while others do not have any SCM events. It has been suggested that this heterogeneity may facilitate adaptation to changing environments²⁵. The heterogeneity observed at low temperature challenges the quantification of the SCM. We used the relative median fluorescence throughout the manuscript to quantify SCM error rates as a summary statistic. Nevertheless, these numeric values do not represent the distributions' complexity (they are neither normal nor symmetric). Despite this complex scenario, SCM events are consistent among biological replicas.”

We also included Fig. S4.

Fig. S4. As temperature decreases *E. coli* cultures present higher stop codon miscoding levels and higher degree of heterogeneity among cells from a clonal population. A) The distributions of the mean fluorescence per cells of *E. coli* cultures transformed with the reporters that inserted each of the stop codons at positions 105 and 155 and grown at 18°C, 25°C, 30°C, 37°C, and 42°C. Three biological replicas are shown per reporter. The vertical lines represent the median of fluorescence of the NC and the PC (wild-type mScarlet). B) Fluorescence and bright field image of an *E. coli* culture transformed with the reporter that introduced a TGA at position 105 of the mScarlet. While some cells are prone to SCM (marked with arrows), others do not show any SCM.

3- Also the results of the validation experiments shown in Figs 2E and F are not convincing. The misreading efficiency expressed as % of His-tag signal varies from 0 to 80%, whereas most of the fluorescence signals are low, below 20%, except for 3 outliers (94, 105 and 135) It is absolutely unclear how the authors manage to obtain a correlation coefficient of 0.7 based on these data. Then, they delete points that have a particularly large deviation using a 30% difference criterion (i.e., they exclude points that deviate by more than 30% in the two types of readout - why?). 30% is entirely an arbitrary value; at 15% they would have to exclude even more points and then the whole validation exercise becomes meaningless. Instead, they should have excluded all of the points where the signal is too close to 0 in both types of signals and then calculate the correlation coefficient. In my opinion, there is a very poor correlation between fluorescence and His-tag reported, which raises serious questions about the validity of the experimental approach.

This control experiment aimed to identify what we define as “dark reporters”: constructs with high SCM rates without showing fluorescence. This phenomenon is due to the misincorporation of amino acids that can affect the maturation of the fluorophore or the conformation/stability of mScarlet. Further, now also confirm that the high fluorescence signal observed in the reporters with TGA introduced in positions 94, 105, and 135 of the mScarlet are not a result of an enhanced mScarlet fluorescence variant. Since at TGA sites usually tryptophan is misincorporated by SCM, we studied the fluorescence of the variants where tryptophan is introduced in positions 94 (same sequence as wild-type mScarlet), 105, and 135 (Fig. S7A). While the Ala-105-Trp mutant had a comparable fluorescence to the wild-type mScarlet, Pro-135-Trp mutant displayed one order of magnitude reduced fluorescence, and therefore we reclassified this reporter as a “dark reporter”. We confirmed and validated by His-tag detection a SCM rate of 75% for the reporter that introduced a TGA at position 135 (see answer 4 to Reviewer #3 and the new Fig 3E). The reclassification of the reporter for amino acid 135 does not change any of the conclusions of the paper. We hypothesise that the higher fluorescence observed previously for this reporter relates to a different amino acid misincorporation.

We now removed the correlation analysis in Figure 2E because the correlation is indeed mainly driven by the few reporters with high SCM, therefore, we now think that the correlation analysis was not very informative, as suggested by Reviewer #1 and #2. Most reporters consistently show no SCM in the fluorescence and western blot assays. Instead, we performed a more qualitative analysis that allowed us to identify a total of five dark reporters out of 117 (Fig. S7B, Table S4). We considered a reporter dark when the discordance between His-tag and fluorescence signal is over 30% (5 reporter out of 117). From the set of non-dark reporters, 75 presented less than 0.1% of SCM rate (calculated as the fluorescence signal relative to the wild-type mScarlet), 33 presented between 0.1-10% of SCM rate and 4 presented more than 10% of SCM.

We changed the main text (line 236): “Most reporters consistently show no SCM in the fluorescence and western blot assays. We further checked that the highest SCM error rate detected (Trp-95-TGA, Ala-105-TGA, and Pro-135-TGA) were not an artefact resulting from an enhanced mScarlet fluorescence variant. Since at TGA sites usually tryptophan is misincorporated by SCM, we studied the fluorescence of the variants where tryptophan is introduced in positions 94 (same sequence as wild-type mScarlet), 105, and 135 (Fig S7A). While the Ala-105-Trp mutant had a comparable fluorescence to the wild-type mScarlet, Pro-135-Trp mutant displayed one order of magnitude reduced fluorescence, and therefore we reclassified this reporter as a “dark reporter” (Fig S7B and Table S4). We confirmed by His-tag detection a SCM rate of 75% for the reporter that introduced a TGA at position 135 and hypothesise that the higher fluorescence observed previously for this reporter relates to a different amino acid misincorporation.”

We now substitute Fig. 2E and 2F by Fig. S7 and added Table S4.

Fig. S7. **A)** While introducing a tryptophan in position 105 did not compromise the fluorescence of the mScarlet, introducing a tryptophan in position 135 decreased one order of magnitude the fluorescence of the mScarlet. **B)** Comparing the fluorescence signal with the his-tag signal allows the classification of the reporters between dark reporters (discordance between His-tag and fluorescence signal is over 30%, Table S4) and fluorescent reporters. Among the fluorescence reporters, 75 presented 0-0.1% SCM error rate, 33 presented 0.1-10% SCM rate and 4 more than 10% SCM error rate (SCM error rate was measured as the median of fluorescence relative to the wild-type).

Reporter name	% His-tag	% Fluorescence
Ala-105-taa	26.1138	0.0303
Ala-105-tag	16.6564	13.8490
Ala-105-tga	53.4456	81.2308
Ala-165-taa	0.0000	0.0769
Ala-165-tag	0.0000	0.3108
Ala-165-tga	13.8180	4.0235
Arg-150-taa	0.0000	0.0000
Arg-150-tag	0.0000	0.3755
Arg-150-tga	51.3889	4.1801
Asn-195-taa	0.0000	0.3064
Asn-195-tag	0.0000	0.0956
Asn-195-tga	0.0000	1.0808
Asp-155-taa	0.0000	0.1104
Asp-155-tag	0.0000	0.4369
Asp-155-tga	7.6391	18.0366
Asp-170-taa	0.0000	0.0000
Asp-170-tag	0.0000	0.0042
Asp-170-tga	0.0000	0.0094
Gln-110-taa	0.0000	0.2310
Gln-110-tag	0.0000	1.6252
Gln-110-tga	43.2341	0.0235
Glu-11-taa	22.3912	0.0024
Glu-11-tag	0.0000	0.0032
Glu-11-tga	0.0000	0.0650
Glu-115-taa	0.0000	0.0072
Glu-115-tga	0.0000	0.6391
Glu-145-taa	0.0000	0.0224
Glu-145-tag	0.0000	0.0610
Glu-145-tga	3.1135	5.5713
Glu-31-taa	0.0000	0.0134
Glu-31-tag	0.0000	0.0822
Glu-31-tga	7.1517	0.6052
Glu-90-taa	0.0000	0.3587
Glu-90-tag	0.0000	0.0568
Glu-90-tga	30.2749	3.0498
Glu-95-taa	0.0000	0.0024
Glu-95-tag	0.0000	0.0000
Glu-95-tga	0.0000	2.5026
Gly-160-taa	0.0000	0.0000
Gly-160-tag	0.0000	0.0000
Gly-160-tga	33.9755	0.0000
Gly-21-taa	0.0000	0.0000
Gly-21-tag	0.0000	0.0000
Gly-21-tga	0.0000	0.0000

Gly-36-taa	0.0000	0.0000
Gly-36-tag	0.0000	0.0000
Gly-36-tga	24.9932	0.0078
His-205-taa	0.0000	0.0099
His-205-tag	0.0000	0.0287
His-205-tga	0.0000	1.6275
His-26-taa	0.0000	0.0070
His-26-tag	0.0000	0.0635
His-26-tga	0.0000	1.6590
Ile-120-tag	0.0000	0.0056
Ile-120-tga	0.0000	0.0022
Ile-61-tag	0.0000	0.0029
Ile-80-taa	0.0000	0.0037
Ile-80-tag	0.0000	0.0148
Ile-80-tga	0.0000	0.9731
Leu-125-taa	0.0000	0.0047
Leu-125-tag	0.0000	0.0020
Leu-125-tga	0.0000	0.5865
Leu-175-taa	0.0000	0.0085
Leu-175-tag	0.0000	0.0004
Leu-175-tga	0.0000	0.6199
Leu-200-taa	0.0000	0.0000
Leu-200-tag	0.0000	0.0000
Leu-200-tga	0.0000	0.0000
Lys-140-taa	0.0000	0.0032
Lys-140-tag	0.0000	0.0593
Lys-140-tga	11.2558	2.0226
Lys-16-taa	0.0000	0.0883
Lys-16-tag	0.0000	0.2042
Lys-16-tga	0.0000	0.3331
Lys-185-taa	0.0000	0.0000
Lys-185-tag	0.0000	0.0050
Lys-185-tga	0.0000	0.0030
Lys-46-tag	0.0000	0.0320
Lys-46-tga	0.0000	0.7955
Lys-75-taa	0.0000	0.0077
Lys-75-tag	0.0000	0.0067
Lys-75-tga	11.2602	5.9544
Lys-85-taa	0.0000	0.0022
Lys-85-tag	0.0000	0.0198
Lys-85-tga	0.0000	0.0006
Met-13-taa	0.0000	0.0249
Met-13-tag	0.0000	0.0057
Met-13-tga	9.4340	0.0037
Met-19-taa	0.0000	0.0434
Met-19-tag	0.0000	0.0233
Met-19-tga	0.0000	0.0024
Met-190-taa	0.0000	0.0110
Met-190-tga	48.607	0.2116
Met-190-tga	48.607	0.2116
Phe-100-taa	0.0000	0.0044
Phe-100-tag	0.0000	0.0079
Phe-100-tga	0.0000	0.0037
Phe-130-taa	0.0000	0.0000
Phe-130-tag	0.0000	0.0056
Phe-130-tga	0.0000	5.2781
Pro-135-taa	0.0000	0.2827
Pro-135-tag	0.0000	5.1460
Pro-135-tga	82.5845	78.6557
Pro-56-taa	0.0000	0.0174
Pro-56-tag	0.0000	0.0154
Pro-56-tga	0.0000	2.7410
Thr-180-taa	0.0000	0.0006
Thr-180-tag	0.0000	0.006
Thr-180-tga	0.0000	0.0008
Thr-210-taa	0.0000	0.1463
Thr-210-tag	0.0000	0.2253
Thr-210-tga	0.0000	0.2590
Trp-94-taa	0.0000	0.0019
Trp-94-tag	0.0000	0.0000
Trp-94-tga	42.6087	57.5179

Table S4. Percentage of His-tag expression relative to the wild-type mScarlet and percentage of fluorescence signal measured as the median of fluorescence relative to the wild-type mScarlet.

4- Another interesting set of data is mass spec identification of the read-through products. There are very few, even at 18°C, and the SCM efficiency is very low (up to 1.35%; Fig. 5). This is consistent with the literature data, where the spontaneous readthrough efficiency has been

estimated in the range of 0.2 to 2%, but strongly contradicts the authors' notion that SCM can reach 80% at 18° (Fig. 2).

We would like to clarify that we performed two sets of mass spectrometry experiments.

The first one was the analysis of nine select reporters to detect the peptides that confirm the SCM event and reveal the amino acids misincorporated at the site of the stop codon (Table S1, Fig 3D). The second set of experiments was designed to test if SCM occurs in the *E. coli* proteome (Fig 5). Whole proteome shotgun LC/MS-MS revealed 95 peptides originating from SCM events. This set was reduced to 49 peptides because we selected peptides detected in 3 out of the 6 studied replicates. We further filtered the set and removed peptides that extended by only one amino acid beyond the stop codon, resulting in 17 peptides. We have not done any quantification, and do not report SCM rates (or efficiencies, as the reviewer calls it). We only reported what % of the genes carrying a type of STOP codon had SCM event detected. We detected SCM in 0.23% of *E. coli* proteins with TAA (7 out of 2648) and 0.86% (10 out of 1158) of *E. coli* proteins with TGA stop codons, and none for TAG (0 out of 284).

We further compared *E. coli* grown at 18 and 37°C and we detected more unique SCM peptides at low temperature. However, we have not measured the SCM rate, and therefore there is no contradiction with our previous results.

5- Minor point:

l. 280: the authors state that they did not observe alternative-frame peptides, which is expected, because they use a 0-frame C-terminal tag for purification. It would be important to state that these results do not exclude that alternative-frame peptides do exist, as it is easy to overlook this.

Yes, we agree. We clarified this point now in the manuscript (line 345) "Thus, our approach does not allow us to rule out the existence of frameshifts as a cause of SCM."

Reviewer #2 (Remarks to the Author):

Introduction:

In this manuscript, Romero and colleagues have employed a reporter assay to investigate stop codon readthrough at many positions spread throughout a protein coding sequence. They utilize this approach to measure the dependence of readthrough propensity on sequence position, stop codon identity, and several environmental conditions. Their analyses reveal some local sequence features correlated with extent of readthrough. They follow up with mass spectrometry and RNA-seq measurements to determine the relative contributions of transcriptional and translational errors to the observed miscoding in a subset of reporter constructs. Finally, the authors perform proteome-wide mass spectrometry measurements and compare the readthrough trends for about 30 E. coli proteins to their results from the reporter assay.

Overall, the major contribution of this work is the medium-throughput (by today's standard) reporter-based measurements of sequence position and sequence context effects on stop codon readthrough. The systematic nature of these measurements allows the authors to measure effects that would be difficult to uncover otherwise, including surprisingly high levels of miscoding (e.g., 80%) that can arise at specific sequence positions and under environmental conditions as simple as growth in minimal media. However, there are unfortunately several serious issues with this manuscript. These range from the authors' portrayal of the novelty of their approach and results, to the presentation of the data, to the analyses performed and conclusions drawn therefrom. Below, we outline several major critiques, followed by several more minor points.

Major points:

1- In the Introduction (e.g., paragraph beginning at line 49), the authors reference prior work of Ballesteros et al. (EMBO J 2001), Meyerovich et al. (PNAS 2010), and Zhang et al. (PNAS 2020) as a basis to outline the novelty of their contributions herein. However, the referencing of prior work is misleading and incomplete:

- Zhang et al. 2020 also measured read-through at all three stop codons as well as multiple frameshift-prone sequences. Meyerovich et al 2010 similarly looked at TAA and TGA stop codons in their studies in another bacterial species, B. subtilis. Therefore, the measurement of readthrough effects at multiple stop codons in a given context (line 52) cannot be portrayed as a point of novelty in the present work.*

Yes, the reviewer is right that these studies have measured the read-through effect at all three stop codons at different environmental stress. Nevertheless, they have not tested the effect of environmental stress on multiple frameshift-prone sequences. Accordingly, we have removed from the main text (line 51) where we highlight the novelty of measuring the read-through effects at multiple stop codons at different environmental stresses.

- Similarly, the authors cite Rosenberger and Foskett (Mol Gen Genet 1981) as the inspiration for their reporter assay (lines 83-84). This is surely a classic work, but the assay used is almost identical to that presented in Meyerovich et al. 2010. This more recent example should be referenced when the assay is introduced.*

We added as a reference to the Meyerovich et al study regarding the assay (line 83) : "To monitor inaccuracy in protein synthesis termination, inspired by the pioneering work of R.F. Rosenberger and G.Foskett²⁴ and the more recent study of Meyerovich et al²³, we designed a fluorescence reporter whereby stop codon miscoding can be visualized and quantified in *E. coli*."

- The authors also fail to acknowledge a body of prior work that has used ribosome profiling to explore stop codon readthrough proteome-wide. This includes Dunn et al. (eLife 2013) and Baggett et al. (PLoS Genet 2017). The latter is cited by the authors, but only in a*

cursory fashion in the Discussion, related to the mutated RF2 in E. coli strain K-12, and not related to the conclusions of this work on stop-codon readthrough proteome-wide in E. coli.

We included the reference to Dunn *et al.* in the introduction (line 42) “Further, slippery sequences upstream the stop codon that cause the ribosome to slip and thereby lead to stop codon readthrough are often functional^{13,19,20},” and in the discussion (line 485) “Ribosome profiling to explore stop codon readthrough proteome-wide in *Drosophila melanogaster* proposed that readthrough added plasticity to the proteome during evolution¹³.”

We included the reference to Baggett *et al.* in the result section (line 408) “and the previous ribosome profiling study on *E. coli* that observed enrichment for TGA and depletion of TAA in the SCM events detected³⁴.”

We further included the reference to Baggett *et al.* in the discussion (line 489) “Further, ribosome profiling on *E. coli* determined that at >50 genes, the ribosome signature after the stop codon is suggestive of translation past the stop codon⁴¹.”

2- In the paragraph beginning at line 132 and throughout the paper, the authors comment on the highest error rates occurring at TGA stop codons (a known phenomenon). However, the K-12 strain of E. coli used for all experiments presented here has a mutation in RF2, the release factor that recognizes TGA, known to have reduced hydrolysis activity (Uno et al., Biochimie 1996) and increased readthrough (Baggett et al., PLoS Genet 2017). This fact should be noted as soon as results with highest error rates at TGA codons are reported in the manuscript. This is especially important considering that the most striking examples measured and cited, e.g., an error rate of ~80% at position 135 at 18 oC, occur specifically at TGA stop codons.

We agree and include in the result section, as soon as the TGA > TAG > TAA SCM error trend is mentioned (line 144): “The highest error rate detected with TGA could be due to a mutation in RF2 present in the wild-type K-12 MG1655 *E. coli* strain (Baggett et al., 2017; Uno et al., 1996), that is known to be less efficient in recognizing stop codons.”

3- In Figures 2, 3, 4, and 5, there are numerous serious issues in the presentation of the data impeding understanding of the results and their significance. A few examples of this systemic issue: Throughout the manuscript, the figure captions are very minimal and are missing key details in how experiments were conducted, what is being plotted in each panel, or even which experiments / results the plots pertain to. The captions are simply not descriptive enough and would need to be re-written to address this issue.

We improved the figure captions and included more details now both describing the plots and analysis as well as including details on the methods.

- *Fig 2A: the variable plotted in the heat map is never defined. Furthermore, the title of the legend is “log 10”, but the axis is labeled in linear units (i.e., 100,10,1... vs. 2,1,0...).*

Thank you for pointing this out. The variable plotted in the heat map is the percentage of the median of the fluorescence distribution of each reporter relative to the wild-type mScarlet (positive control).

We have changed the label and the panels 2A and 2C and extended the caption of:

Fig. 2A (line 909): “**A**) In cells grown in rich media, the stop codon miscoding (SCM) levels vary considerably as a function of the temperature. At 18°C more reporters displayed SCM events and at a higher rate. The variable plotted is the percentage of the median of the fluorescence distribution of each reporter relative to the wild-type mScarlet (PC).”

Fig. 2B (line 912): “**C**) Nutrient scarcity promoted SCM. In cells grown in minimal media, SCM levels increased while lowering the growth temperature. The variable plotted is the percentage of the media of the fluorescence distribution of each reporter, relative to the wild-type mScarlet (PC).”

- *Figs 2B, 2D: It is fundamentally unclear what is being depicted in these panels.*

The variable plotted is the percentage of the median of the fluorescence distribution of each reporter relative to the wild-type (positive control). Each panel is divided into three subpanels: the upper one describes the reporters with TAA, the middle with TAG, and the lower with TGA.

We have changed the panels 2B and 2D and extended the captions of:

Fig. 2B (line 909): “**B**) When cells were grown in rich media, at all tested temperatures, TAA was the most accurate codon terminating the protein synthesis and TGA the least. The variable plotted is the percentage of the median of the fluorescence distribution of each reporter relative to the wild-type (PC). The upper subpanel describes the reporters with TAA, the middle with TAG, and the lower with TGA.”

Fig 2D (line 918): “**D**) When cells are grown in minimal media, at all tested temperatures, TAA is the most accurate codon terminating the protein synthesis and TGA the least. The variable plotted is the percentage of the median of the fluorescence distribution of each reporter relative to the wild-type (PC). The upper subpanel describes the reporters with TAA, the middle with TAG, and the lower with TGA.”

- *Fig 2E: the presentation of the data is confusing. What is the meaning of the numbers? What is the meaning of the vertical lines from these numbers to the two lines? What are the two (dashed vs solid) lines?*

We agree with the reviewer. We now substitute Fig 2E and 2F by Fig S7 where we show a more qualitative analysis of the His-tag and fluorescence relationship and added Table S4.

We now substitute Fig. 2E and 2F by Fig. S7 and added Table S4.

- *It is not clear what exactly is being shown in Fig 3D. Is this a purely qualitative table of findings? What are the underlying quantitative data from the MS experiments?*

We agree that the figure needs to be clarified. Accordingly, we have changed the figure and extended the caption. The MS experiments presented in Fig 3D. and Table S1. were done with the reporters after purifying them with His-tag affinity resin. Thus, we could not quantify the likelihood of SCM events. That said, the MS analyses did provide information of the relative abundance of the detected specific amino acids misincorporation (e.g. 99% Trp and 1% Cys at position 105 TGA, Table S1) . Fig 3D. shows the major misincorporation, i.e., the misincorporations detected with a relative abundance >10% by MS, for the reporters with TAA(purple), TAG (green) and TGA (blue). The information on the relative abundances of the amino acids misincorporations is shown in Table S1.

Fig 3D (line 932): “**D**) **Stop codon miscoding events primarily occur due to amino acid misincorporations identified by mass spectrometry.** TGA (blue) is almost always replaced by tryptophan. TAA(purple) is mainly replaced by alanine, glutamine, and tryptophan and TGA(green) is replaced by glutamine and tryptophan. Misincorporation of a few other amino acids at the stop codon was clearly a minor process (relative abundance <10%). These MS experiments were done with the reporters after purifying them with His-tag affinity resin. Thus, we could not quantify the likelihood of SCM events. However, the MS analyses did provide information on the relative abundances of specific amino acids misincorporations (Table S1). Stop codon miscoding, SCM.”

- *Figs 4C, 4D: these lack a caption entirely. The manuscript text also fails to clarify what is shown here. Therefore, the analysis and results represented here are entirely unclear.*

Thank you for pointing out the missing captions. In Fig 4C and D we represented the percentage of RNA mismatches of the middle nucleotide in the x-axis depending on the nucleotide context, i.e., depending on the nucleotides upstream and downstream. For example, when “AAG” is written in the x-axis, the % of RNA mismatches for the A nucleotide is plotted when it is preceded by A and followed by G. Each panel is divided into three subpanels: the upper one describes the nucleotides that encode an amino acid, the middle one that encode a premature stop codon, and the lower one that encode a canonical stop codon. This analysis intended to discern whether the higher probability of a mismatch in a given position found only at premature stop codons (Fig 4A and 4B) is influenced by the nucleotide context. The analysis shows that the adjacent nucleotides do not influence the probability of a mismatch in a given position. We can see that nucleotides with the same genetic contexts have different probabilities of mismatches depending on what they encode for. For example, the probability of a mismatch for G preceded by a T and followed by an A is significantly higher when it encodes for a premature TGA than when it encodes for a canonical TGA or when it encodes for an amino acid, meaning an out of frame TGA stop codon. Thus, this analysis shows that the identity of the adjacent nucleotide does not influence the probability of mismatches. It suggests that the probability of mismatch in a given nucleotide is higher when it encodes for a premature stop codon than when it encodes an out of frame stop codon or canonical stop codon.

We agree, and accordingly, we changed the panels 4A and 4D and changed the Fig 4 captions (line 946):

Figure 4. Low RNA polymerase accuracy at premature stop codons. A) “The likelihood of the RNA polymerase mismatching is higher at a premature stop codon. **Density plots of %mRNA mismatches (nucleotide substitutions) observed in RNA-Seq experiments at the four nucleotides (T, G, C, A) grouped by amino-acid encoding (left), premature stop codon (middle) or canonical stop codon (right) encoding nucleotides.** **B)** The identity of the neighbouring nucleotides does not influence the accuracy of the RNA polymerase. Thus, the RNA polymerase is more prone to incorporate a mismatch at a stop codon only when it is in frame. **The same analysis is shown as in panel A) while including only those mismatches that lead to an amino acid change (non-synonymous mismatches).** **C)** The identity of the adjacent nucleotides does not influence the probability of RNA mismatch at a given position. The probability of RNA mismatch in a given nucleotide is higher when it encodes for a premature stop codon than it is for canonical stop codon or out of frame stop codon sites. The plot shows the percentage of RNA mismatches of a given nucleotide (represented in bold on the x-axis) depending on the adjacent nucleotides. **D)** The identity of the adjacent nucleotides does not influence the probability of non-synonymous RNA mismatch at a given position. The same analyses are shown as in panel C) but only for RNA mismatches that lead to amino acid changes (non-synonymous mismatches). **E)** RNA polymerase errors at premature stop codons result in a broad range of misincorporated amino acids. **Bar plot showing the amino acid encoded by the mRNA sequence at the premature stop codon versus the number of read. Next to each bar is shown the % that the amino is encoded.”**

4- The authors briefly postulate that GC content downstream of the stop codon is correlated with readthrough frequency due to increased stability of secondary structures in this nearby region. This naturally raises the question of what kind of secondary structures might be relevant and why they might lead to more misreading of the stop codon. With only 43 sites in total, it would be reasonable to calculate predicted secondary structures in the vicinity of each tested premature stop codon (and potentially all sites in mScarlet). The MFE of such structures, and other features such as whether they are predicted to encapsulate the stop codon itself, might correlate with observed SCM. The authors should computationally predict secondary structures for each tested site and determine to what extent they might explain their findings. Indeed, the authors cite Chen et al. (NSMB 2013) on

the point that downstream stem-loops containing GC pairs decrease ribosomal accuracy. Presentation of the corresponding secondary structures for the reporter constructs used here would help to assess if this mechanism is likely driving the results in this case and strengthen the paper.

Thank you for this great suggestion. Accordingly, we performed *in silico* analysis on the effect of predicted RNA secondary structure and SCM efficiencies. We used RNAfold2(Lorenz et al., 2011) to calculate the optimal RNA secondary structures at the experimentally tested temperatures. To assess whether mRNA secondary structures might induce errors in protein synthesis termination, we first predicted the minimum free energies (MFE) of the possible secondary structures in a 100-nt window downstream and upstream of the inserted stop codon. Based on Chen et al.(Chen et al., 2013), we defined a 6-nt spacer downstream of the stop codon, which is typically occupied by the ribosome. No spacer was considered upstream of the stop codon. Secondly, we analysed the correlation between the SCM score and the minimum free energy of the predicted structures and found no significant correlation (Fig. S9 A and B). As expected, decreasing the temperature stabilised the RNA secondary structures and may explain the higher SCM errors at lower temperatures (Fig. S9 A and B). Yet, the higher stability of the secondary structures would not explain the presence of hotspots for SCM errors in the mScarlet sequence. Further, we classified the reporters according to their SCM score in three groups: no SCM errors, mid-tendency to SCM errors, and high-tendency to SCM errors (see Method section for further details). We observed no significant differences (t-test) between the stability of the secondary structures of each group (Fig. S9C).

We then focused on the local thermodynamic stability of the secondary structures surrounding the premature stop codon. In the context of ribosomal frameshifting, it has been argued that local thermodynamic stability has a greater effect than the structure's overall stability (Mouzakis et al., 2013). Thus, we studied the number of all base pairs, the number of G-C pairs, and the number of longest stretch of consecutive pairs in a set of windows (5, 10, 20 and 50-nt) upstream and downstream of the premature stop codon of the most stable RNA structure previously predicted (100-nt window upstream and 100-nt downstream of the premature stop codon plus the 6-nt spacer). We found no correlation of these values with the SCM score (Fig. S9 D and E).

Overall, the predicted mRNA secondary structures could not explain the presence of hotspots for SCM. Moreover, the huge impact of the 5-nt before and 1-nt after the stop codon (see answer 4 to reviewer #3) on the likelihood of SCM suggests that interactions within the ribosomal complex might play a bigger role in the modulation of SCM events than the secondary structure of the mRNA.

We changed the main text (line 295) "Overall, these analyses indicate that **the adjacent region to the premature stop codon (4-nt upstream and 1-nt downstream)** influences the SCM rate. While G increases the SCM propensity, T has the opposite effect. It could be that the higher stability of the secondary structures in nucleotide regions of the mRNA with high GC content hampers the fidelity of translation. **To test this hypothesis, we predicted the minimum free energies (MFE) of the possible secondary structures in a 100-nt window downstream and upstream of the inserted stop codon and analysed the correlation between the SCM score and the minimum free energy of the predicted structures and found no significant correlation (Fig. S9 A, B and C, see methods for more details).**

As expected, decreasing the temperature stabilised the RNA secondary structures that may explain the higher SCM errors at lower temperatures (Fig. S9 A, B and C). Yet, the higher stability of the secondary structures would not explain the presence of hotspots for SCM errors in the mScarlet sequence. Since in the context of ribosomal frameshifting, it has been argued that local thermodynamic stability has a greater effect than the structure's overall stability (Mouzakis et al., 2013), we then focused on the local thermodynamic stability of the secondary structures surrounding the premature stop codon and its correlation with the SCM scores (Fig. S9 D and E, see methods for more details). We found no correlation (Fig. S9 D and E).

Overall, the predicted mRNA secondary structures could not explain the presence of hotspots for SCM events”

We further changed the discussion section (line 472) “We hypothesise that the higher stability of the secondary structures of nucleotides at low temperatures could explain the effect of temperature on SCM. Strong mRNA secondary structures hamper the unfolding of the mRNA, potentially influencing the accuracy of protein synthesis⁴². The same arguments may support why the identity of the nucleotide at the adjacent regions affects the protein synthesis accuracy since stem-loops containing GC base pairs were shown to decrease ribosomal accuracy⁴². Although we found no significant correlation between predicted mRNA secondary structures and the likelihood of SCM (Fig S9), this lack of correlation may be due to the prediction’s limitations (Lorenz et al., 2011).”

We included the method section (line 687): “**mRNA secondary structure prediction**
We used RNAfold2(Lorenz et al., 2011) to calculate the optimal RNA secondary structure that has the minimum free energy at the experimentally tested temperatures. We predicted the minimum free energies of the possible secondary structures in a 100-nt window downstream and upstream of the inserted stop codon. Based on Chen et al.(Chen et al., 2013), we defined a 6-nt spacer downstream of the stop codon, which is typically occupied by the ribosome. No spacer was considered upstream of the stop codon (Fig S9A, 9B and 9C).
We then focused on the local thermodynamic stability of the secondary structures surrounding the premature stop codon. We studied the number of all base pairs, the number of G-C pairs, and the number of pairs within the longest stretch of consecutive pairs in a set of windows (5, 10, 20 and 50-nt) upstream (Fig S9D) and downstream (Fig S9E) of the premature stop codon of the most stable RNA structure previously predicted (100-nt window upstream and 100-nt downstream of the premature stop codon plus the 6-nt spacer).”

We included Fig. S9 in the supplementary material.

Figure S9. The impact of predicted mRNA secondary structure on SCM. **A)** The predicted minimum free energy (MFE) is shown for the most stable secondary structure elements in a 100-nt window upstream of the premature stop codon at different temperatures. Decreasing the temperature stabilised the predicted RNA secondary structures. There is no correlation between the stability of the predicted RNA secondary structures and the likelihood of SCM (SCM score). **B)** The predicted minimum free energy (MFE) is shown for the most stable secondary structure elements in a 100-nt window downstream of the premature stop codon at different temperatures. Decreasing the temperature stabilised the predicted RNA secondary structures. However, there is no correlation between the stability of the predicted RNA secondary structures and the likelihood of SCM. **C)** There are no significant differences in the stability (MFE) of the predicted RNA secondary structures of the reporters with no SCM errors, mid-tendency to SCM errors, and high-tendency to SCM errors (see Method section for further details). **D)** The local thermodynamic stability of the secondary structures (i.e., the number of all base pairs, G-C pairs, and the longest stretch of consecutive pairs) in a set of 5, 10, 20, and 50-nt windows upstream of the premature stop codon does not correlate with the likelihood of SCM events. **E)** The local thermodynamic stability of the secondary structures (i.e., the number of all base pairs, G-C pairs, and the longest stretch of consecutive pairs) in a set of 5, 10, 20, and 50-nt windows upstream of the premature stop codon does not correlate with the likelihood of SCM events.

5- here are some important issues with the analysis related to Figure 4 which limit how these results should be interpreted:

- *The apparently surprising finding that higher transcriptional errors are observed specifically at premature stop codons (PTCs) is actually strongly influenced by Rho-mediated premature transcription termination in the context of PTCs. This causes increased degradation of mRNAs in which the PTC is not corrected by an error, and an in-cell “selection” for the mRNAs containing such transcriptional errors. Therefore, the authors cannot conclude that transcriptional error rates are elevated at PTCs or invoke an unknown mechanism related to ribosome-RNAP coupling. Instead, the elevated observation of errors at the stop codons should be explained as a consequence of Rho-mediated transcription termination.*

This is an interesting hypothesis. We agree that the observed higher number of mismatches detected at premature and inframe stop codons might be influenced by the degradation of mRNA containing premature stop codons (Belasco, 2010; Nilsson et al., 1987). However, the fact that the mismatches identified are often synonymous, i.e., lead to another stop codon (Fig 4E), suggests that this is not the only explanation. For example, for the reporter in which a TGA was introduced in position 56, the mismatch towards another stop codon is found two orders of magnitude more frequent than the mismatch towards cysteine. However, taking into account single point mutations and assuming that all nucleotide mutations have the same probability (Fig. 4A and 4B), a mutation towards cysteine would be more likely to happen. We observed similar results for premature TAA and TAG, suggesting this pattern is not an exception but rather the rule (Fig 4E). Thus, the hypothesis of a higher RNA polymerase error rate at premature and inframe stop codons cannot be ruled out.

Investigating the mechanism/s underlying this phenomenon is a novel and ongoing research line of the lab that will require further experiments, and it is out of the scope of this work. As part of this investigation, we recently observed a similar scenario in eukaryotic cells. Very briefly, we studied the mRNA sequence of several fluorescence reporters for SCM events in *S. cerevisiae* (data will be shown upon request). We also observed more RNA mismatches at premature stop codons than at the rest of the sequence. Although, in this case, the nonsense-mediated mRNA decay could be invoked to explain the higher number of mismatches observed (via degradation of mRNA containing premature stop codons), we also observed enrichment in nucleotide mismatches leading to other stop codons at premature stop codon sites.

- *The authors report error rates at the RNA level on the order of 0.01% to 0.15%. However, it is unclear how (or whether) such low error rates can be detected by the PacBio sequencing technology employed, which has recently reported error rates on the order of 0.2% with (Wenger et al., Nature Biotech 2019). The authors should discuss their choice of RNA sequencing technology, the expected sequencing error rates, and clearly state how they relate to the measured transcriptional error rates reported.*

Thank you for pointing out that we should indeed discuss the methodology and highlight in the manuscript that high fidelity RNA-seq technology is crucial and necessary to detect rare events such as the error rate we measured at premature stop codon sites. Here, not only the PacBio sequencing technology but also the sample preparation (we used the sample preparation protocol developed for SARS-CoV2 sequencing (https://www.protocols.io/view/ncov-2019-sequencing-protocol-bp2l6n26rgqe/v1?version_warning=no), and the choice of the reverse transcriptase and polymerase will impact the detected nucleotide mismatches that the PacBio consensus calling pipeline cannot detect and filter out. We provide here more details about these 3 steps:

1. Reverse transcription reaction.
Specifically, we used the Superscript IV reverse transcriptase that is a modification of Superscript III which leads to even higher efficiency of the reverse transcriptase reaction

(= yield). Error rates were reported for Superscript III in the original publication to be $5,01 \times 10^{-5}$ (median error rate, see Figure 6 in Houlihan et. al)(Houlihan et al., 2020).

2. cDNA amplification

We used Q5 NEB High-fidelity polymerase, that is one of the highest fidelity polymerases: Q5 incorporates 1 error in 28.000.000 base pairs (in comparison Taq polymerase incorporates 1 error in 100.000 base pairs, see table below).

[redacted]

<https://international.neb.com/products/pcr-qpcr-and-amplification-technologies/q5-high-fidelity-dna-polymerases/q5-high-fidelity-dna-polymerases/how-is-fidelity-measured>

3. PacBio HiFi sequencing

During the PacBio HiFi library sequencing (using PacBio polymerase which is a Phi29 derivative) we obtained reads up to 100 kB. Since our cDNA fragments are short (~1 kB), most of them have been read during the circular sequencing steps multiple times (quantified as the number of passes). Specifically, only ~15% of all reads had less than 10 full passes and the maximum number of passes was 60 (see plot below).

We applied the standard PacBio SMRT link pipeline for circular consensus sequencing (ccs) calling. This pipeline by default considers only reads with an error rate of less than 1%. Additionally, based on (Wenger et al., 2019), 10 passes correspond to QV of 30 (99,9% base accuracy), 20 passes correspond to QV40 (99,99% base accuracy), that agrees with our empirical error rates observed (Fig. S15, mean=0.005% mismatch per base).

To summarize, we added to the method section (line 748) “In order to detect rare transcriptional error events, we optimized: i) the reverse transcription reaction; ii) the cDNA amplification reaction; iii) PacHiFi sequencing step. Such optimization enabled us to achieve >99.99% accuracy. Briefly: i) 300 ng of total RNA per sample was mixed with 50 ng of random hexamers and dNTPs and hybridized at 65°C for 5 minutes. cDNA was reversely transcribed making use of the Thermo Fisher Superscript IV transcriptase according to the manufacturer’s instructions (reporter median error frequency of ⁵²).

ii) Then, mScarlet was specifically amplified from the resulting cDNA with forward primer 5’ AGTTATTTTACCACTCCCTATCAGT 3’ and reverse primer 5’ AGTAGCGGTAAACGGCAGAC 3’ resulting in an amplified PCR fragment of 948 bps. The NEB Q5® High-Fidelity DNA polymerase was used according to the manufacturer’s instructions. This DNA polymerase is one of the highest fidelity polymerases incorporating, according to the manufacturer, 1 error in 28.000.000 base pairs.”

We further added (line 781): “During the PacBio HiFi library sequencing we obtained reads up to 100 kB. Since our cDNA fragments are short (~1 kB), most of them have been read during the circular sequencing steps multiple times (quantified as the number of passes). Specifically, only ~15% of all reads had less than 10 full passes and the maximum number of passes was 60. The standard PacBio SMRT link pipeline for circular consensus sequencing by default considers only reads with an error rate of less than 1%. Additionally, based on Wenger *et al*, 10 passes correspond to 99,9% base accuracy, and 20 passes correspond to 99,99% base accuracy, that agrees with our empirical error rates observed (Fig S15, mean=0.005% mismatch per base)”

Finally, we also added Fig. S15.

Figure S15. Empirical calculation of RNA nucleotide misincorporation error rate (percentage of mRNA nucleotide mismatch per base) for **A)** all type of RNA mismatches and **B)** non synonymous RNA mismatches.

6- The results of the bioinformatic analysis for additional downstream stop codons in Fig. 5B are clearly influenced by the defective RF2 variant found in the K-12 strain of *E. coli*, as noted above. This should be noted in the text and the authors should perform the same analysis with a B strain of *E. coli* (with an efficient RF2) to see if the frequency of additional stop codons after a “TGA” is decreased as would be expected.

Thank you, this is a great suggestion to compare the K-12 strain and B strain of *E. coli*. We performed the same analysis that we did for the wild-type K-12 MG1655 *E. coli* strain, and we looked for additional stop codons in 15-, 20- and 30-nt windows downstream of the canonical stop codon for the BL21 *E. coli* strain. We observed that for both *E. coli* strains, it is more likely to find an additional stop codon after TGA, the most error-prone stop codon, then after TGA or TAA.

We added to the main text (line 414) “However, this selection pressure could be influenced by the fact that the wild-type K-12 MG1665 *E. coli* strain used here carries a defective RF2 that is less

efficient in recognizing the TGA stop codon. We, therefore, repeated the analyses using the BL21 *E. coli* strain, which expresses a more efficient RF2 variant. Both *E. coli* strains showed the same results (Fig 5B and SX). Regardless of the efficiency of the RF2, it is more likely to find an additional stop codon after TGA, the most error-prone stop codon, then after TAG, and after TAA.” and we included in the Supplementary Figure S13.

Figure S13. The occurrence of an additional stop codon in a 15, 30, and 60-nt window downstream of the stop codon correlates with the protein synthesis accuracy of the stop codon in the *E. coli* genome of the BL21 strain. TGA, as the least accurate stop codon, has the highest frequency of an additional stop codon in the 3' regions of genes.

7- For Figure 5C and the associated text, it is initially strange to see that some proteins have more and similar read-through at 37 oC vs. 18 oC, in contrast to the reporter measurements. The authors should comment on this. A large contribution to this effect may be that there is (apparently) no normalization of each protein's expression levels, which likely vary with temperature, which confounds interpretations of the relative amounts of readthrough at different temperatures. The authors should note this as well, or normalize for this effect and explain how this is done.

Figure 5C does not contain any quantification regarding SCM levels. It only lists the genes with SCM peptides identified in the proteome wide MS/MS analysis at both temperatures. We see more unique peptides at 18°C. The reviewer is right, if we were to quantify the error rate, we should take into account expression level. This would require quantitative mass spectrometry measurements, that we have not done.

Minor points:

1- In the discussion of the results of Fig. 2E-F, the authors make a major point of achieving a near “ideal” slope of 1 for their % fluorescence (main SCM readout) vs. % His-tag signal plots by excluding 4 “dark reporters” in Fig 2E. However, this overemphasizes the linearity of the results. The vast majority of points cluster near the origin of the plot, so few points are contributing much to the linear trend being fit. Essentially only 3 points drive the “near-ideal” linear fit (near ~40-80, ~50-100 in x,y), and this requires the removal of 4 data points. Thus, it is inappropriate to emphasize the slope of this fit; instead, what the authors have clearly shown is a positive relationship between the two variables, but not necessarily a clean linear one, based on the data at hand. (Happily, this change does not fundamentally alter the authors’ results or their interpretation.)

We completely agree and we realised that we have not been clear with the data presentation and the interpretation. The purpose of the Western blot analysis was to rule out that we miss SCM events using our fluorescence reporter because of lack of fluorescence due to amino acid

substitutions (dark reporters) induced by stop codon readthrough. We were happy to see that Western blotting that can detect non-fluorescent mScarlet, identified only 4 dark reporters.

Additionally, we were happy to see the overall great agreement between the results obtained via fluorescent microscopy and Western blots: indeed most reporters show no SCM in both assays, and the ones with high SCM are consistently high in both. We agree that emphasising the linear fit is not important and driven by a few points only. We removed Figure 3F. Also see our answer to Reviewer #1 who also raised issues with the data presentation.

Overall, the Western blot as an orthogonal (semi-quantitative) assay validated our results, and all further experiments were done using fluorescence microscopy that is quantitative and gives us access to cell to cell heterogeneity (the distribution of fluorescence values per cell).

2- Meyerovich et al. (PNAS 2010) measured temperature-dependence of frameshifting to rescue GFP reporter fluorescence, and the conclusions reached were similar to those in the present work for stop codon miscoding (lower temperature was correlated with more frameshifting / readthrough). This suggests a common mechanism may be at play. The authors should discuss the similarity of these results and potential implications.

We now add a few sentences discussing the topic (line 211): “Global inspection of these results suggests that environmental stress conditions, such as non-optimal growth temperature or nutrient scarcity, promote stop codon miscoding and, thus, protein heterogeneity. **A similar temperature effect has been previously reported for ribosomal frameshift errors²³, suggesting an increase in different types of protein synthesis errors at low temperatures. A known *E. coli* general response to environmental stress conditions compromises the downregulation of genes related to transcription and ribosomal biogenesis³⁷. This general stress response may decrease protein synthesis accuracy, which could modulate protein heterogeneity.** Since sudden environmental changes can modulate protein heterogeneity in a rapid and short-lasting fashion, we hypothesize that *E. coli* **might make use of protein synthesis’ errors** to quickly adapt to sudden and short-lasting stress conditions.”

3- “Minimal media” is misspelled in Figure 2.

Thank you, we fixed it.

4- In discussing the extent to which transcriptional errors, as opposed to misreading at the translational level, might drive SCM (section starting at line 284), the authors should cite previous work such as Dunn et al. (eLife 2013) which previously showed by ribosome profiling that readthrough events translational-wide are primarily not driven by sequence changes at the mRNA level. They should also cite the literature on transcriptional error rates and translational error rates such as Li and Lynch, eLife 2020, and Meyerovich et al., PNAS 2010 which suggest a roughly order-of-magnitude difference in these frequencies.

We are now citing all these references (line 381): “Overall, the RNA-seq and mass spectrometry results suggest that mainly translation errors contribute to SCM events. **This is in agreement with previous studies that revealed higher translation than transcription error rates^{13,23,43}.** Transcription errors contribute less to SCM events, yet they diversify the resulting protein sequence independent of the stop codon identity.”

5- The similarity in the trend of SCM between low-temperature and low-nutrient conditions is noted but not discussed substantially. However, this suggests some underlying mechanism common to these two cases. The authors might discuss this.

We now add a few sentences discussing the topic (line 211), see answer to minor point 2.

6- SCM “hotspots” are temperature-dependent in nontrivial ways, e.g., several reasonably highly-miscoded sites in mScarlet at 37 oC are no longer miscoded at 25 oC, and then re-emerge as

potential “hotspots” at 18 oC (Fig. 2A). The authors should comment on this more complex temperature-dependence and its possible origins. Could this be evidence of some elevation of SCM at 37 oC compared to 25 oC, for example, in contrast to the overall trend the authors emphasize?

See answer to question 2 of Reviewer #1.

Reviewer #3 (Remarks to the Author):

This study by Romero Romero et al looks at changes in stop codon miscoding events across different environmental conditions. The manuscript does a good job of showing that temperature and nutrition can impact miscoding events of a premature codon. This work also shows that while both transcriptional errors and ribosomal readthrough can lead to miscoding events, ribosomal readthrough is the main contributor to miscoding. Furthermore, they find that there is a correlation between the nucleotides after the stop codon and miscoding events. Overall, while this manuscript has made a lot of interesting observations, I believe it could be improved if they understood more of the miscoding events they are seeing.

1- While it is stated that “stop codon miscoding may occur with a rate as high as 80%, ... , suggesting that evolution frequently samples stop codon miscoding events.” It is unclear how evolutionary relevant the high rate of miscoding they see for a premature stop codon is to endogenous stop codon miscoding. For one they see fewer miscoding events the closer they get to the end of the ORF. They never see >10% miscoding after amino acid position 155.

While the reviewer is right that we detected the highest rate of SCM (up to 80%) at two positions (105 and 135) located at the N-terminal half of the mScarlet, we found no correlation between SCM error rate and amino acid position ($R=0.15$, $p=0.362$, Fig. S8). On the other hand, at position 215 (the closest position to the end of the ORF tested), SCM may occur with a rate as high as 3%. Even lower SCM error rates have been reported to have evolutionary relevance (Romero Romero et al. 2022; Li 2001).

We now specify in the main text (line 292) “Finally, to study whether the distance to the C-terminal modulates the propensity of SCM events, we analysed the correlation between the SCM score and the amino acid position in mScarlet. We found no significant correlation (Fig. S8, $R=0.15$, $p=0.362$,” and we include the Fig. S8.

Figure S8. Stop codon miscoding events occur evenly along the mScarlet sequence. The SCM score describes an SCM event’s likelihood at a given position (see Method section for further details). There is no significant correlation between SCM score and mScarlet amino acid position. We excluded four positions with a discordance between His-tag and fluorescence signal over 30% (dark reporters).

2- Similarly, while they find significant transcriptional misincorporation for premature stop codons, there is an almost unmeasurable amount of transcriptional misincorporation at canonical stop codons. This shows that the cell treats premature stop codons very differently than canonical stop codons.

Yes, we agree, from the transcription perspective, our results showed that the cell treats premature stop codons differently than canonical ones, an intriguing result that needs further research (see comments to reviewer #2, point 5). However, from the translation perspective, we do not have any indication of differentiation. Since ribosomal errors are the main contributors to SCM errors, the

results derived from the library of reporters, even though they involved premature stop codons, could help us understand the termination errors of canonical stop codons. In fact, the proteome-wide detection of SCM events by mass spectrometry agreed with the conclusions derived from the study of the library of reporters. *I. e.*, for both studies: i) we detected that TGA is the most error-prone stop codon: ii) the identity of the base following the stop codon influences the likelihood of SCM, and, iii) there are more cases of SCM at 18°C than at 37°C temperature.

3- It seems the manuscript could be strengthened if there was a better understanding of why there is such a difference between a premature stop codon and the canonical stop codon. Is it related to distance? If you added a GFP on to the end of Scarlet does that increase the miscoding at the canonical Scarlet stop codon? If this was true, this could mean that stop codons in multi-gene operons could increase readthrough during environmental stress and the gene products could become fused? Could this increase protein function efficiency?

We would like to note that our results showed a difference between premature and canonical stop codons only at the transcriptional level. However, transcription errors have a minor contributor to the overall rate of SCM errors and we do not see any correlation between the overall rate of SCM errors and distance from the premature stop codon to the canonical one (see answer to question #1, Fig. S8).

That said, we found the reviewer's suggestion very interesting that environmental stress, inducing SCM events, might generate fused proteins in multi-gene operons. To address this question we analysed whether the distribution of the three stop codons within multi-gene operons differ from the distribution of stop codons of single genes. For this purpose, using RegulonDB database (Tierrafría et al., 2022), we classified the *E. coli* genes into three categories:

- 1) *Single-gene operon*, those in operons that contain only one gene.
- 2) *Final multi-gene operon*, those expressed last of the multi-gene operon.
- 3) *Within multi-gene operon*, those not expressed last of the multi-gene operon.

We found that genes within multi-gene operons are enriched in TGA, the most error-prone codon, while depleted in TAA, the most accurate one (Fig. S14A). We then focused on the identity of the nucleotide after the TGA stop codon. Genes within multi-operon are enriched in G and depleted in T after the TGA stop codon. Since we observed that the presence of T increases and G decreases the protein synthesis termination efficiency, it seems that SCM errors within multi-operon genes are promoted. Therefore the hypothesis that environmental stress conditions could induce the expression of fused protein complexes seems plausible. At least we do not find any indication of selection pressure against SCM events within multi-operon genes. On the contrary, we found that SCM may be promoted among these genes. Further, 56% of the genes detected by the proteome-wide mass spectrometry analysis that suffered SCM events are genes within multi-gene operons.

We added to the main text, discussion section (line 495) “Interestingly, we observed that genes within multi-genes operons are enriched in TGA, the most error-prone stop codon, while depleted in TAA, the most accurate one (Fig. S14A). Further, genes within multi-gene operons are enriched in G and depleted in T after the TGA stop codon (Fig. S14B). Since we observed that the presence of T increased and G decreased the protein synthesis termination accuracy, protein expression termination errors are predicted to be enhanced among genes within multi-gene operons. We hypothesise that SCM events in multi-gene operons might result in the expression of fused protein complexes, which may modulate their functionality. Further, 56% of the genes detected by the proteome-wide mass spectrometry analysis that suffered SCM events are genes within multi-gene operons (they represent 43% of the *E. coli* proteome), which supports this hypothesis.” and we include the Supplementary figure S14.

Figure S14. Stop codon miscoding events may be more likely among genes within multi-operons. Using the RegulonDB database (Tierrafría et al., 2022), we classified the *E. coli* genes into three categories: 1) *Single gene operon*, those in operons that contain only one gene. 2) *Final multi-gene operon*, those expressed last of the multi-gene operon. 3) *Within multi-gene operon*, those not expressed last of the multi-gene operon. **A)** Genes within multi-operon are enriched in TGA, the most error-prone codon, while depleted in TAA, the most accurate, compared to genes that belong to single-gene and final multi-gene operons. **B)** Genes within multi-operon are enriched in G and depleted in T at the first nucleotide position downstream of the TGA stop codon. We found that the presence of T increases and G decreases the protein synthesis termination efficiency (Fig 3B and 3C).

4- The paper could also be improved if they moved beyond correlation for the effect of the nucleotides downstream of the stop codon and directly showed that mutating a G to T or T to G decreases or increases readthrough from their reporters.

Thank you for this great suggestion. We followed the reviewer's idea and studied a set of mutants designed to elucidate the effect of the nucleotides up and downstream of the stop codons on SCM events. We focused on those reporters with the highest SCM score: those with a premature stop codon in positions 105, 135, and 155. Since we observed that the identity of the nucleotides in the 5-nt window upstream and 1-nt window downstream of the stop codon modulate the likelihood of SCM, we mutated those regions. Specifically, we mutated the nucleotide downstream of the stop codon to T, which according to our findings, is expected to decrease the likelihood of SCM events. We mutated the 5-nt upstream of the stop codon to ATTAT because an enrichment in T and A is expected, according to our previous findings, to decrease the likelihood of SCM events. The exact mutations performed are shown in Table S3. Since these mutants change the amino acid sequence in up to four consecutive amino acids, that will probably impact the functionality of the mScarlet (e.g., could lead to dark reporter), we measured the SCM events detecting the His-tag expression with Western blot. We observed that both the downstream mutation towards T and the upstream mutation towards ATTAT reduced the likelihood of SCM events for all three tested positions. Moreover, mutating the upstreams and downstream regions simultaneously have an additive effect.

We now include in the main text (line 280) "To experimentally confirm the previous findings, we designed a set of mutants and measured their likelihood of SCM. Since these mutants changed the mScarlet's primary structure in up to four amino acids, what will likely affect its functionality, we measure the SCM events detecting the His-tag expression with western blot. We focus on those reporters with the highest SCM score: those with a premature stop codon in positions 105, 135, and 155. For these reporters, we mutated the downstream region towards T and the downstream regions towards ATTAT (Table S3). We predicted these mutants to increase the protein synthesis termination accuracy. We transformed *E. coli* cells with these mutants and grew them at 18°C. We observed that both the downstream mutations towards T and the upstream mutation towards ATTAT reduce the likelihood of SCM events for all three tested positions. Further, mutating the upstream and downstream region simultaneously have an additive effect (Fig. 3E). Thus, we confirmed our prediction experimentally.

Finally, to study whether the distance to the C-terminal modulates the propensity of SCM events, we analyzed the correlation between the SCM score and the mScarlet amino acid position. We found no significant correlation (Fig S7, $R=0.15$, $p=0.362$)."

We also include a new panel in main text Fig. 3E and Table S3

Premature stop codon position	WT sequence	Mutating the 1-nt downstream of the stop codon	Mutating the 5-nt upstream of the stop codon	Mutating the 5-nt upstream and 1-nt downstream of the stop codon
105	GCGGCTGAG	GCGGCTGAT	ATTATTGAG	ATTATTGAT
135	ATGGCTGAG	ATGGCTGAT	ATTATTGAG	ATTATTGAT
155	CGGAATGAC	CGGAATGAT	ATTATTGAC	ATTATTGAT

Table S3. Summary of the mutants designed to experimentally test the effect of the nucleotides up- and downstream of the stop codon on SCM events. We mutated the nucleotide downstream of the stop codon to T and 5-nt upstream of the stop codon to ATTAT.

Fig 3E. The downstream mutation to T and the upstream mutation to ATTAT reduce the likelihood of SCM events for all three tested positions. To quantify the SCM likelihood, we measured the His-tag expression with western blot in *E. coli* cells transformed with these mutants and grew at 18°C. We calculated the percentage of His-tag expression compared with the internal positive control (PC, wild-type mScarlet).

5- There are a number of small fixes to the manuscript, especially in the Figures and Figure legends:

- Figure 2B and 2D subpanels should be labeled better.
- Line 346 - "after TGA, the most error-prone stop codon, then after TGA, and then after TAA" The second mention of TGA should presumably be TAG.
- Line 347 - "trend is maintained if we extend the search to 20-nt and 30-nt windows (Fig 5B)" but Fig 5B shows 15-nt, 30-nt, and 60-nt, not 15, 20, 30 as mentioned in the text.
- Figure 5B - It is unclear what the blue, green, and purple colors mean
- Figure 5C - The figure legend mentions 9 genes exclusive to 18C and 3 genes detected at 37C (I assume that should be exclusive to 37C), yet the figure only shows 5 exclusive to 18C and 2 to 37C.
- Line 824 - The figure legend for Figure 4 is missing for Fig 4C,D. There should also be more description in the figure one what is being presented in Fig 4C,D.

- Line 832 0 “TAG, as the last accurate stop codon” this should say TGA.

We thank the reviewer for pointing out these typos and missing information. We fixed the typos, changed Fig 5, and changed the caption of:

Fig. 2 (line 908) **Figure 2.** Non-optimal growth temperatures and nutrient scarcity promote stop codon miscoding. A) In cells grown in rich media, the stop codon miscoding (SCM) levels vary considerably as a function of the temperature. At 18°C more reporters displayed SCM events and at a higher rate. The variable plotted is the percentage of the median of the fluorescence distribution of each reporter, normalizing by the wild-type (PC). B) When cells were grown in rich media, at all tested temperatures, TAA was the most accurate codon terminating the protein synthesis and TGA the least. The variable plotted is the percentage of the median of the fluorescence distribution of each reporter relative to the wild-type (PC). Each panel is divided into three subpanels: the upper one describes the reporters with TAA, the middle with TAG, and the lower with TGA. C) Nutrient scarcity promoted SCM. In cells grown in minimum media, SCM levels increased while lowering the growth temperature. The variable plotted is the percentage of the media of the fluorescence distribution of each reporter, normalizing by the wild-type (PC). D) When cells are grown in minimum media, at all tested temperatures, TAA is the most accurate codon terminating the protein synthesis and TGA the least. The variable plotted is the percentage of the median of the fluorescence distribution of each reporter relative to the wild-type (PC). Each panel is divided into three subpanels: the upper one describes the reporters with taa, the middle with tag, and the lower with tga. E) C-terminal His-tag expression correlates with fluorescence signal. Only four out of 117 reporters underestimate the rate of SCM events likely due to loss of fluorescence (“dark reporters”). Each reporter is represented by a number (position of the mScarlet where the stop codon is introduced) and a colour code, purple for taa reporters, green for tag and blue for tga. F) Excluding the four dark reporters, there is a good agreement between the percentage of His-tag and fluorescence signal. Each reporter is represented by a number (position of the mScarlet where the stop codon is introduced) and a colour code, blue the four “dark reporters” and the rest in black.”

Fig. 4C and 4D (line 953). **C)** The identity of the adjacent nucleotides does not influence the probability of RNA mismatch at a given position. The probability of RNA mismatch in a given nucleotide is higher when it encodes for a premature stop codon that is in the ribosome’s open frame. The plot shows the percentage of RNA mismatched of a given nucleotide (represented in bold on the x-axis) depending on the adjacent nucleotides. **D)** The identity of the adjacent nucleotides does not influence the probability of synonymous RNA mismatch at a given position. The probability of synonymous RNA mismatch in a given nucleotide is higher when it encodes for a premature stop codon that is in the ribosome’s open frame. The plot shows the percentage of synonymous RNA mismatched of a given nucleotide (represented in bold on the x-axis) depending on the adjacent nucleotides.

References

- Baggett, N. E., Zhang, Y., & Gross, C. A. (2017). Global analysis of translation termination in *E. coli*. *PLoS Genetics*, 13(3), e1006676.
- Belasco, J. G. (2010). All things must pass: contrasts and commonalities in eukaryotic and bacterial mRNA decay. *Nature Reviews. Molecular Cell Biology*, 11(7), 467–478.
- Chen, C., Zhang, H., Broitman, S. L., Reiche, M., Farrell, I., Cooperman, B. S., & Goldman, Y. E. (2013). Dynamics of translation by single ribosomes through mRNA secondary structures. *Nature Structural & Molecular Biology*, 20(5), 582–588.
- Fan, Y., Evans, C. R., Barber, K. W., Banerjee, K., Weiss, K. J., Margolin, W., Igoshin, O. A., Rinehart, J., & Ling, J. (2017). Heterogeneity of Stop Codon Readthrough in Single Bacterial Cells and Implications for Population Fitness. *Molecular Cell*, 67(5), 826–836.e5.

- Houlihan, G., Arangundy-Franklin, S., Porebski, B. T., Subramanian, N., Taylor, A. I., & Holliger, P. (2020). Discovery and evolution of RNA and XNA reverse transcriptase function and fidelity. *Nature Chemistry*, 12(8), 683–690.
- Lorenz, R., Bernhart, S. H., Höner Zu Siederdisen, C., Tafer, H., Flamm, C., Stadler, P. F., & Hofacker, I. L. (2011). ViennaRNA Package 2.0. *Algorithms for Molecular Biology: AMB*, 6, 26.
- Mahmood, T., & Yang, P.-C. (2012). Western blot: technique, theory, and trouble shooting. *North American Journal of Medical Sciences*, 4(9), 429–434.
- Mouzakis, K. D., Lang, A. L., Vander Meulen, K. A., Easterday, P. D., & Butcher, S. E. (2013). HIV-1 frameshift efficiency is primarily determined by the stability of base pairs positioned at the mRNA entrance channel of the ribosome. *Nucleic Acids Research*, 41(3), 1901–1913.
- Nilsson, G., Belasco, J. G., Cohen, S. N., & von Gabain, A. (1987). Effect of premature termination of translation on mRNA stability depends on the site of ribosome release. *Proceedings of the National Academy of Sciences of the United States of America*, 84(14), 4890–4894.
- Romero Romero, M. L., Landerer, C., Poehls, J., & Toth-Petroczy, A. (2022). Phenotypic mutations contribute to protein diversity and shape protein evolution. *Protein Science: A Publication of the Protein Society*, 31(9), e4397.
- Tierrafría, V. H., Rioualen, C., Salgado, H., Lara, P., Gama-Castro, S., Lally, P., Gómez-Romero, L., Peña-Loredo, P., López-Almazo, A. G., Alarcón-Carranza, G., Betancourt-Figueroa, F., Alquicira-Hernández, S., Polanco-Morelos, J. E., García-Sotelo, J., Gaytan-Núñez, E., Méndez-Cruz, C.-F., Muñiz, L. J., Bonavides-Martínez, C., Moreno-Hagelsieb, G., ... Collado-Vides, J. (2022). RegulonDB 11.0: Comprehensive high-throughput datasets on transcriptional regulation in *Escherichia coli* K-12. *Microbial Genomics*, 8(5). <https://doi.org/10.1099/mgen.0.000833>
- Uno, M., Ito, K., & Nakamura, Y. (1996). Functional specificity of amino acid at position 246 in the tRNA mimicry domain of bacterial release factor 2. *Biochimie*, 78(11-12), 935–943.
- Wenger, A. M., Peluso, P., Rowell, W. J., Chang, P.-C., Hall, R. J., Concepcion, G. T., Ebler, J., Functammasan, A., Kolesnikov, A., Olson, N. D., Töpfer, A., Alonge, M., Mahmoud, M., Qian, Y., Chin, C.-S., Phillippy, A. M., Schatz, M. C., Myers, G., DePristo, M. A., ... Hunkapiller, M. W. (2019). Accurate circular consensus long-read sequencing improves variant detection and assembly of a human genome. *Nature Biotechnology*, 37(10), 1155–1162.

REVIEWER COMMENTS

Reviewer #1 (Remarks to the Author):

Upon revision, the authors addressed some, but not all of my concerns. Furthermore, new text additions exposed weaknesses of the experiments and the interpretation. As stated in my previous report, a large part of the reported analysis is not novel, i.e. all of the systematic analysis of the readthrough (RT) efficiency of UAG, UGA and UAA has been published before. The effect of stop codon context was also studied before, but these previous results are not sufficiently discussed in this paper. Although the authors have added some citations, their list of important previous publications is far from complete and they are not addressing most glaring discrepancies, e.g. the role of the +1 nucleotide. In this respect, the omission of the work of Leif Isaksson is particularly striking, as he made seminal contributions to the field and his observations have been broadly supported by many groups.

On the positive side, there are three interesting observations:

1. The temperature effect. The caveat here is that the authors did not address the issues that I criticized in point 2 of my previous report. The data in Fig 2A remain inconsistent and the comment of the authors that 25° measurement is an "outlier" is less than satisfactory. They should either re-analyze the whole data set or investigate the reason for this deviation. The data provided in Fig S4 are complex and the whole argument becomes hermetic at this point. The data are not convincing and not properly dealt with.
2. The low RNA polymerase accuracy at premature stop codons. While I did not make comments on this issue, I find the criticism of the two other referees quite consistent and strong and the authors' replies verbose and not satisfactory.
3. The proteome-wide detection of stop codon RT. Interesting, but preliminary data.

Further major comments:

Introduction:

1. Line 34. Transcriptional stop codon miscoding (SCM) converts a stop codon into a sense codon, then the stop codon simply ceases to function as stop signal. It cannot be miscoded (or, more accurately, misread). Generally, stop codon readthrough is a misreading, not miscoding event. These definitions should be changed for accuracy. Similarly, the authors should not state that they study "infidelity", this is a misnomer. They determine the error frequency of termination.
2. The pioneering work on translation termination by Leif Isaksson should be considered, as it has a direct relevance for the project.

Results:

3. My concerns regarding Figure 2 A,B and Fig S2 are still valid. These are reproducibility issues. In the new Fig S4, we observe SCM rates at positions 105TGA and 155TGA (25°C) quantified around 10%. In contrast, the same positions in Fig S2B (and Fig2A) display rates of 1% and 0% respectively. In which medium was the experiment in Figure S4 (new) executed? LB? Furthermore, it appears that the experiment in Fig 2A was performed with a single replicate. Please add statistical information (number of technical and biological replicates) in all Figs and Fig panels. In the response letter, the authors mention that the cultures were grown at the laboratory's ambient temperature, which they claim to be 25°C. I doubt this is controlled enough, RT is usually 22° or anything between 19 and 27° depending on the season. Using more controlled conditions is essential.
4. In response to the comment that increased RT on UGA codons can be caused by the strain-specific variant of RF2, the authors tested the strain with an alternative RF2 isoform, but apparently only in the analysis of the stop codon frequency after the 1st stop-codon (Fig S13). However, a much more pressing question is whether a more efficient RF2 isoform results in lower error frequencies, which should be checked by mass spectrometry. In contrast, while comparing the stop-codon frequency

downstream of the ORF codon (lines 417-422), the comparison is not convincing, as the evolutionary distance between E.coli K-12 and E. coli B is relatively small. Comparing E. coli K-12 with more distantly related organisms, such as Salmonella enterica (which also harbors a wild-type prfB allele), would likely yield more insightful results.

5. The authors used pASK as a vector for their reporter constructs, claiming that it is a low-copy plasmid. According to reference 48 (line 529), this plasmid contains a ColE1 origin of replication. It also does not have the rop gene which keeps the copy number low. Therefore, the pASK plasmid is actually a high-copy plasmid. Did the author exchanged the ColE1 origin of replication with a p15A origin of replication? If yes, this should be stated explicitly in Methods.

6. The various recoded mScarlet isoforms could have potentially different protein stabilities (half-life) in vivo. This should be mentioned in the text as a caveat for the reporter assay and tested for at least some candidates.

7. Further related to the unclear reproducibility of the data, I have serious concerns regarding the Western blot results: Such experiment executed as a single replicate should not be considered as quantitative. WB is a sensitive technique in for identification of protein products, but it often produces variable quantitative technique. Quantification by WB requires necessarily replicates. Furthermore, the authors report only some of the gel pictures that were used for generating Table S4 and Fig S7B. I would like to see larger gel pictures (including possible additional/unspecific bands), loading controls (or at least pictures of the gels before the transfer for assessing gel loading) and protein ladders. Displaying a protein ladder would also help the reader to interpret the two bands displayed in Fig S6. The appearance of multiple bands seems to be a common issue when over expressing His-tagged fluorescent proteins. However, the authors write that they used only the upper band for the quantifications (Method section). The presence of multiple bands makes the quantification less reliable. I am also not very convinced about the criterion that they used for considering a reporter "dark". It is a bit arbitrary. Also, "four reporter out of 117 were considered dark" (line 232-233): in Figure S7B there are 5 dark reporters.

8. Fig S9: Some information are missing. Which medium was used? Which constructs (positions) were used? Is it always the same dataset of Fig S8? Is it the same dataset for all figure panels? In the last line of the figure legend "upstream" should be downstream".

9. Fig S12: The quality of some spectra is not great and some important information are missing. Were the samples cultivated at 18 or 37 °C? The peptide sequence could be better annotate by indicating the inserted amino acid at the stop codon position, as well as the extent of coverage of the stop codon region by the peptide. Moreover, the precursor mass should be displayed in the extracted ions chromatograms.

Discussion:

10. The text of the discussion became rather confusing after revision. In part, it is just repetition of the results. The important discrepancy concerning the nature of the favoured +1 nucleotide is not addressed. The evolutionary arguments are all unwarranted, as there is no evolutionary data in the Results. The discussion on the roles of secondary structures is confusing, as the authors state them as a possible regulator of RT, whereas their own analysis shows no correlation. (There are also different opinions about how the mRNA secondary structure elements affect frameshifting. The authors cite a single old paper, while there is a large body of more recent literature which doubts the role of thermodynamic stability alone. This is again the example of limited use of the available literature). The authors also mention the biological roles of protein isoforms as studied in Drosophila, but the coverage of the literature is poor, so the point is not well made.

11. Discussion and reviewer 3 response: the observation that many of the error-prone stop codons are located in polycistronic mRNAs is interesting, however one should also consider the presence of additional stop codons in the intergenic regions and in the two adjacent genes are in frame.

Throughout the text

12. Most of the evolutionary discussion is not warranted. The authors suggest that a general increase of translation error rate could be beneficial for the cell under stress conditions (lines 216-218 and

Discussion). The majority of errors, whether arising from DNA mutations, transcription, or translation, tend to be either detrimental or neutral to the cell. It is more reasonable to claim that recoding events could have an adaptive function only in some specific cases/genes. In other words, each case should be considered individually. Furthermore, the authors use the high frequency of recoding as an argument in favour of adaptivity of SCM (lines 454-458). Evolution is a complex phenomenon and a high frequency of a genetic trait cannot be used alone to support adaptation. For this purpose, there are specific computational methods (e.g. dN/dS). One possible strategy could involve examining whether mRNA sequences encoding the C-terminal extensions of genes exhibiting high rates of readthrough demonstrate notably elevated or diminished dN/dS values.

Minor points:

1. Line 31: "termination is optimized": I suggest refraining from using this passive form. In my opinion, it imparts an 'intelligent design' (or even creationist) connotation. Genetic traits arise as outcomes of mutational pressure, genetic drift, and selection.
2. Line 106-107: what was the inducer concentration in this experiment (2.1% rate)?
3. Line 142: "inefficient RF2" should be "inefficient RF2 (or better prfB) allele"
4. Line 143: There are numerous wild-type E. coli strains. Refer to it as K12 MG1655 strain.
5. Line 203: "minimum media" should be "minimal media" (or better medium)
6. Line 285: "downstream" should be "upstream"
7. Line 288: how and why was the ATTAT sequence chosen?
8. Line 295-296: why "4-nt upstream". Should not it be 5-nt?
9. Line 555: "in LB-agar plates" should be "on LB-agar plates"
10. Lines 691-693: Why wasn't a spacer upstream of the stop codon taken into account for the structural predictions?

Reviewer #2 (Remarks to the Author):

We thank the authors for providing a revised manuscript that responds to each of our comments and concerns.

There is one additional point that we believe the authors should address in the manuscript: the influence of Rho-mediated premature transcription termination and subsequent RNA decay on the population of mRNAs and transcriptional errors that is observed. The influence of Rho should be discussed in the context of the transcriptional errors reported (Figure 4), and the authors should include as well the argument made in their rebuttal that the preponderance of transcriptional errors resulting in other stop codons (rather than sense codons) at PTCs clearly indicates that there are additional mechanisms at play. This will substantially help readers who may otherwise be skeptical based on the function of Rho-mediated transcription termination of PTC-carrying mRNAs in E. coli.

Otherwise, we believe the revisions address the points we raised, and that the resulting manuscript is substantially improved.

We provide just a few minor points for additional corrections where typos in the text and figure panels confuse the apparent intended meaning:

- Fig. 2C caption: Use of "media" where "median" was presumably intended.
- Fig. 3D caption: repeated reference is made to TAA and TAG stop codons commonly being miscoded as tryptophan; whereas based on the data in Fig. 3D and Table S1 it appears the authors meant to refer to tyrosine in these cases (TGA, on the other hand, indeed is commonly miscoded as tryptophan).
- Fig. 4B caption: Caption is unclear. Panel 4B does not seem to have any bearing on nucleotide

context as described in the caption.

- Line 211: Apparent stray "Although" at the start of this sentence – presumably a typo?

Reviewer #3 (Remarks to the Author):

Overall, I find the reviewers did a good job of answering my critiques and find the changes they made to the paper improve the manuscript. The key finding seems to be that temperature can affect the readthrough rate as measured by both fluorescence and western blots. They now directly show that the nucleotide sequence surrounding the stop codon influences readthrough at these positions.

I wonder if Figure S4 could be included in Figure 2, as it seems to be a better presentation of the data and is more easily interpretable by the reader.

I don't want to ask them to do more experiments at this phase, but in general I would be interested to know if the same effects are seen in log-phase at low temperature, as cells grown to saturation are nutrient limited, hence why their growth is saturated. Do cells lose this heterogeneity of read-through in log phase, or is there no read through at all even at low temperatures?

Response to reviewers' comments

Reviewer #1 (Remarks to the Author):

Upon revision, the authors addressed some, but not all of my concerns. Furthermore, new text additions exposed weaknesses of the experiments and the interpretation. As stated in my previous report, a large part of the reported analysis is not novel, i.e. all of the systematic analysis of the readthrough (RT) efficiency of UAG, UGA and UAA has been published before. The effect of stop codon context was also studied before, but these previous results are not sufficiently discussed in this paper. Although the authors have added some citations, their list of important previous publications is far from complete and they are not addressing most glaring discrepancies, e.g. the role of the +1 nucleotide. In this respect, the omission of the work of Leif Isaksson is particularly striking, as he made seminal contributions to the field and his observations have been broadly supported by many groups.

On the positive side, there are three interesting observations:

1. The temperature effect. The caveat here is that the authors did not address the issues that I criticized in point 2 of my previous report. The data in Fig 2A remain inconsistent and the comment of the authors that 25° measurement is an “outlier” is less than satisfactory. They should either re-analyze the whole data set or investigate the reason for this deviation. The data provided in Fig S4 are complex and the whole argument becomes hermetic at this point. The data are not convincing and not properly dealt with.

Figure 2A shows the study of SCM events when growing cells at different temperatures and in minimal and rich media. In minimal media, the temperature effect on SCM is clear. Lowering the growth temperature increases the rate of SCM for TAG and TGA stop codons, and for TAA, the trend is 18°C>25°C>37°C~42°C (Fig 2D). However, in rich medium, the temperature effect on SCM is less clear. Lowering the growth temperature also increases the rate of SCM for TAA and TAG stop codons with the trends 18°C>25°C>37°C~42°C and 18°C>25°C>37°C~42°C respectively (Fig 2D). Notably, TGA shows a different pattern (18°C > 25°C < 37°C > 42°C). Two scenarios are considered:

1. There is no temperature effect in rich media and the pattern observed is a result of variability among experiments.
2. There is indeed a temperature effect with TGA inconsistency at 25°C attributed to challenges in analysing fluorescence data due to (i) cell-to-cell heterogeneity, (ii) non-symmetrical and multimodal fluorescence distributions, and (ii) difficulty in differentiating low fluorescence values from the background fluoresce.

In order to discern between the two scenarios, we first performed an experiment with biological replicates (It is not possible to study biological replicates with the whole library due to time constraints) and with an extra temperature to better define the temperature trend (Fig S4). This experiment shows that:

1. There are high levels of SCM heterogeneity among biological replicates.
2. The complex fluorescence distribution pattern, challenging to summarise with a single statistical value (median), led to different median fluorescence values for relatively similar populations. For example, SCM error rates at position 105TGA (18°C) ranged from 31% to 65%, and in 155TGA (18°C), from 3.2% to 11% (Fig S4).

The experiment supported the second scenario, indicating a consistent temperature effect trend for TGA and other conditions and stop codons: 18°C > 25°C > 37°C > 42°C (Fig S4).

Additionally, we reanalysed the 25°C data set with a different threshold to differentiate between background fluorescence and fluorescence signal due to SCM events. Previously, we defined the threshold as the median plus two standard deviations of the fluorescence signal of the negative control (NC). Since we included two biological replicates of NC, we chose the highest threshold. Figure 1A shows the fluorescence distribution of the reporters with the TGA stop codon. At the right side of the distribution is shown the percentage of fluorescence signal compared with the positive control (PC) of the distribution with a median above the defined threshold.

For this new analysis, we defined the threshold as the median of fluorescence of the NC and we also chose the value of the highest threshold. Figure 1B below shows the fluorescence distribution of the reporters with the TGA stop codon. At the right side of the distribution is shown the percentage of fluorescence signal compared with the positive control (PC) of the distribution with a median above the defined threshold.

This analysis (Fig. 1) showed considerable variation in the number of reporters considered to display SCM, depending on the threshold definition. While this does not alter the SCM rate's magnitude, it influences the number of reporters considered to exhibit SCM.

Response Figure 1. Fluorescence distributions displayed by the *E. coli* cells, transformed with the TGA reporters and grown at 25°C in rich media. The propensity of SCM, calculated as the percent of the median fluorescence compared with the positive control (PC, wild-type mScarlet), is shown for the distributions with a median fluorescence higher than the defined threshold. A) The threshold is defined as the median plus two standard deviations of the fluorescence signal of the negative control (NC). B) The threshold is defined as the median fluorescence signal of the NC. Each fluorescence distribution is derived from one replicate and 280 to 11537 cells.

2. The low RNA polymerase accuracy at premature stop codons. While I did not make comments on this issue, I find the criticism of the two other referees quite consistent and strong and the authors' replies verbose and not satisfactory.

Without knowing specifically what the reviewer finds not satisfactory, we find it hard to reply. The two other referees found our answer satisfactory and even asked us to add it to the discussion of the paper.

3. The proteome-wide detection of stop codon RT. Interesting, but preliminary data.

We do not consider the proteome-wide mass spectrometry data preliminary. We have provided two biological replicates and three technical replicates, and used strict quality filters to identify the peptides that correspond to potential stop codon miscoding events. We are confident about the data and the interpretation. We agree however, that the data opens many other questions and potential research avenues for the future.

Further major comments:

Introduction:

1. Line 34. Transcriptional stop codon miscoding (SCM) converts a stop codon into a sense codon, then the stop codon simply ceases to function as stop signal. It cannot be miscoded (or, more accurately, misread). Generally, stop codon readthrough is a misreading, not miscoding event. These definitions should be changed for accuracy. Similarly, the authors should not state that they study "infidelity", this is a misnomer. They determine the error frequency of termination.

The purpose of introducing a new term in our manuscript, namely stop codon miscoding (SCM), was to have a term that includes both transcription and translation events and that it does not imply a mechanism. For example, readthrough suggests that the ribosome skips the stop codon when, in fact, it misincorporated an amino acid. We wanted to find a neutral word that does not relate to a mechanism. Both DNA and mRNA are *encoding* the genetic information. Therefore, we believe *miscoding* fits to describe nucleotide misincorporations/indels. Similarly, the ribosome and tRNA *decode* the mRNA. Therefore, we like the term miscoding for all events that could, in theory, lead to RT events, such as amino acid misincorporation/STOP codon skipping/frameshift by the ribosome.

Since no other referee raised issues regarding our nomenclature, we would like to stick to SCM, and we added clarifying sentences to the definition of the term.

In the introduction (line 37) we introduce: "Thus, SCM covers transcription and translation events that deviate from the genetic code without implying specific mechanisms."

We changed the wording and do not mention infidelity anymore. In line 42 we changed infidelity to errors.

2. The pioneering work on translation termination by Leif Isaksson should be considered, as it has a direct relevance for the project.

Thank you for pointing us to more literature to cite.

We extended the result section (line 269): "This is consistent with previous work reporting that the two amino acids at the C-terminal of the nascent peptide, and therefore the six nucleotides upstream of the stop codon, modulate termination inaccuracy³⁸⁻⁴⁰."

We extended the discussion (line 486) including the work of Leif Isaksson: "It has been reported that the amino acid identity at the C-terminal of the nascent peptide modulates termination inaccuracy⁴⁹⁻⁵¹. Proline and glycine in the -1 and -2 positions upstream of the stop codon

increases termination inaccuracy⁵⁰. Interestingly, codons encoding for proline and glycine are enriched in C and G. Thus, the observed correlation between GC content upstream of the stop codon and the fidelity of termination might be a consequence of the amino acid effect on the termination.”

Results:

3. My concerns regarding Figure 2 A,B and Fig S2 are still valid. These are reproducibility issues. In the new Fig S4, we observe SCM rates at positions 105TGA and 155TGA (25°C) quantified around 10%. In contrast, the same positions in Fig S2B (and Fig2A) display rates of 1% and 0% respectively. In which medium was the experiment in Figure S4 (new) executed? LB?

Yes, the experiment presented in Figure S4 was done with LB medium. We now include this information in the caption.

Regarding the differences among experiments conducted at 25°C, we acknowledge a high level of SCM cell-to-cell variability. In fact, in Fig S4, we observed SCM error rates at position 155TGA (25°C) ranged from 3.9% to 12%. These differences increase from 0-1% to 12% when comparing experiments conducted independently (i.e., on different days, with variations in media preparation, growth time, etc.). We attribute these differences to:

1- Cell heterogeneity among bacterial clonal populations, as evidenced by the multimodal fluorescence distributions (Fig S2B and S4A) and the diverse fluorescence signals within a single clonal population (Fig S4B). It is worth noting that this heterogeneity has been previously observed, and it has been suggested to play a role in evolution, facilitating adaptation to changing environments (Fan et al. 2017).

2- The heterogeneity observed challenges the quantification of the SCM. Although we have used the median fluorescence to quantify SCM error rates, this numeric value by itself does not represent the distributions' complexity (they are neither normal nor symmetric). Consequently, populations with relatively similar profiles lead to very different median fluorescence values. For example, SCM error rates at position 105TGA (18°C) ranged from 31% to 65%, and in 155TGA (18°C), from 3.2% to 11%.

3- Since the expression level varies almost two-orders of magnitude with the temperature (see the fluorescence distributions of the PC, mScarlet WT, in Fig S4A) to assess the temperature effect, we calculate the median fluorescence relative to the PC, being 100% the PC and 0% the NC. Thus, inherent variations in the fluorescence signal of PC and NC will also influence the value of the error rate that we use to characterise each reporter.

We suggest that the divergences of the experiment conducted at 25°C from the overall temperature trend are primarily attributed to the factors mentioned earlier. To prove so, we selected 6 reporters to study biological replicates and we introduced an extra temperature to study, 30°C, and, therefore, to better define the temperature trend. This experiment confirmed that, although the relation between the temperature and the SCM error rate is not linear, low temperatures increased the SCM error rate. Further, it confirmed the cell-to-cell heterogeneity and the challenge to summarise the fluorescence distributions with a single parameter (see also answer to question #1).

Furthermore, it appears that the experiment in Fig 2A was performed with a single replicate. Please add statistical information (number of technical and biological replicates) in all Figs and Fig panels.

Yes, the experiment presented in Fig 2A was performed with a single replicate because the time scale of the experiment does not allow for replicate. We will now add this information to the captions of Fig 1, 2, S2, S3 and S4.

In the response letter, the authors mention that the cultures were grown at the laboratory's ambient temperature, which they claim to be 25°C. I doubt this is controlled enough, RT is usually 22° or anything between 19 and 27° depending on the season. Using more controlled conditions is essential.

All experiments were done at controlled temperatures. In our institute we have rooms with constant controlled temperatures at 18, 25, 37 and 42°C. For the revision we further add an experiment at 30°C that was done using an incubator.

4. In response to the comment that increased RT on UGA codons can be caused by the strain-specific variant of RF2, the authors tested the strain with an alternative RF2 isoform, but apparently only in the analysis of the stop codon frequency after the 1st stop-codon (Fig S13). However, a much more pressing question is whether a more efficient RF2 isoform results in lower error frequencies, which should be checked by mass spectrometry. In contrast, while comparing the stop-codon frequency downstream of the ORF codon (lines 417-422), the comparison is not convincing, as the evolutionary distance between *E. coli* K-12 and *E. coli* B is relatively small. Comparing *E. coli* K-12 with more distantly related organisms, such as *Salmonella enterica* (which also harbors a wild-type *prfB* allele), would likely yield more insightful results.

We agree that it would be very interesting to test if the more efficient RF2 isoforms result in lower error frequencies. We would have to repeat our work using the *E. coli* B strain, which is not possible due to time constraints, and is in general outside of the scope of this work.

This experiment was done in response to reviewer #2 who suggested that the results of the bioinformatic analysis for additional downstream stop codons in Fig. 5B are influenced by the defective RF2 variant found in the K-12 strain of *E. coli*. We think that comparing strains with smaller evolutionary distances is the most relevant since the changes we see can be more likely attributed to differences in RF2. However, when comparing distant species, it becomes less certain to attribute potential differences to RF2 performance.

5. The authors used pASK as a vector for their reporter constructs, claiming that it is a low-copy plasmid. According to reference 48 (line 529), this plasmid contains a ColE1 origin of replication. It also does not have the *rop* gene which keeps the copy number low. Therefore, the pASK plasmid is actually a high-copy plasmid. Did the author exchanged the ColE1 origin of replication with a p15A origin of replication? If yes, this should be stated explicitly in Methods.

We used the plasmid #65020 purchased from Addgene (<https://www.addgene.org/65020/>). This plasmid included the following modifications to the vector pASK-IBA3 plus:

- 1- The ampicillin-resistance gene is replaced with a chloramphenicol-resistance gene.
- 2- The pBR322 replicon is replaced with a p15A replicon.

We added this to the Methods (line 524) and also provide the plasmid map file: "(Purchased from Addgene #65020, <https://www.addgene.org/65020/>. This plasmid is a modified version of the vector pASK-IBA3 plus, with the following changes: the substitution of the ampicillin-resistance gene with a chloramphenicol-resistance gene and the replacement of the pBR322 replicon with a p15A replicon). We used"

6. The various recoded mScarlet isoforms could have potentially different protein stabilities (half-life) in vivo. This should be mentioned in the text as a caveat for the reporter assay and tested for at least some candidates.

Indeed, the amino acid substitutions can lead to changes in protein stability, half-life and even fluorescent intensities, as we see by the dark reporters. We extended the discussion about these caveats in the text. However testing protein half-life experimentally is outside the scope of this work:

We added to the results section (line 223): “or decreases the stability”

7. Further related to the unclear reproducibility of the data, I have serious concerns regarding the Western blot results: Such experiment executed as a single replica should not be considered as quantitative. WB is a sensitive technique in for identification of protein products, but it often produces variable quantitative technique. Quantification by WB requires necessarily replica. Furthermore, the authors report only some of the gel pictures that were used for generating Table S4 and Fig S7B. I would like to see larger gel pictures (including possible additional/unspecific bands), loading controls (or at least pictures of the gels before the transfer for assessing gel loading) and protein ladders. Displaying a protein ladder would also help the reader to interpret the two bands displayed in Fig S6. The appearance of multiple bands seems to be a common issue when over expressing His-tagged fluorescent proteins. However, the authors write that they used only the upper band for the quantifications (Method section). The presence of multiple bands makes the quantification less reliable. I am also not very convinced about the criterion that they used for considering a reporter “dark”. It is a bit arbitrary. Also, “four reporter out of 117 were considered dark” (line 232-233): in Figure S7B there are 5 dark reporters.

We would like to stress that we do not consider the Western blot results quantitative. We agree with the reviewer that gel based quantification has many issues, especially when working with a single replicate. The purpose of these experiments was to detect SCM independently of the amino acid misincorporated at the stop codon because with the fluorescence reporter the SCM detection depends on the sequence of the mScarlet synthesised. That is, to detect whether the reporters are dark. Our purpose was not quantifying SCM errors. Indeed, the data derived from the His-tag detection are shown in Fig S6B in a qualitative fashion.

In Fig S5 we show the His-tag bands of all the reporters that presented visible His-tag signal together with the positive and negative control included in each of the gels. As the reviewer asked, we provide here the full gel images. For the area quantification of the His-tag signal we excluded the band at around 20 kDa because it does not correspond to the mScarlet (in in-gel fluorescence experiments, the mScarlet wild-type shows two bands between 26 and 34 KDa). When the SCM rate is low, the second band is below the detection limit. Thus, we decided to consider for the analysis only the band that we confidently can attribute to the mScarlet expression. We now repeat the analysis taking into accounts all bands revealed with the his-tag antibody. Table 1 below shows this analysis and the comparison with the previous one. The reporters that we considered dark are highlighted in bold. The dark reporters show also discrepancy between fluorescence signal and his-tag signal when considering all bands revealed by the his-tag antibody.

Gel1

Gel2

Gel3

Gel4

Gel5

Gel6

Gel7

Gel8

Gel9

Gel10

Gel11

Gel12

Gel13

Gel14

Response Figure 2. Detection of His-tag expression with His-tag antibodies. For each gel it is shown, in the first column the in-gel fluorescence signal, in the middle column, the total protein expression signal assayed using Fast Green FCF staining solution and, in the last column, the his-tag signal. All gels contain the signal of cells expressing the mScarlet wild-type as positive control (p) and cells expressing an empty vector as negative control (n) together with the protein marker leader. The name of the reporter loaded in each well is annotated in the button area of the gel.

Reporter name	% His-tag (upper band)	% His-tag (all bands)	% Fluorescence
Ala-105-taa	26.1138	13.6590	0.0303
Ala-105-tag	16.6564	9.2802	13.8490
Ala-105-tga	53.4456	32.1139	81.2308
Ala-165-taa	0.0000	0.0000	0.0769
Ala-165-tag	0.0000	0.0000	0.3108
Ala-165-tga	13.8180	1.9881	4.0235
Arg-150-taa	0.0000	0.0000	0.0000
Arg-150-tag	0.0000	0.0000	0.3755
Arg-150-tga	51.3889	13.8508	4.1801
Asn-195-taa	0.0000	0.0000	0.3064
Asn-195-tag	0.0000	0.0000	0.0956
Asn-195-tga	0.0000	0.0000	1.0808

Asp-155-taa	0.0000	0.0000	0.1104
Asp-155-tag	0.0000	0.0000	0.4369
Asp-155-tga	9.1593	5.2311	18.0366
Asp-170-taa	0.0000	0.0000	0.0000
Asp-170-tag	0.0000	0.0000	0.0042
Asp-170-tga	0.0000	0.0000	0.0094
Gln-110-taa	0.0000	0.0000	0.2310
Gln-110-tag	0.0000	0.0000	1.6252
Gln-110-tga	43.2341	15.4396	0.0235
Glu-111-taa	22.3912	11.1038	0.0024
Glu-111-tag	0.0000	0.0000	0.0032
Glu-111-tga	0.0000	0.0000	0.0650
Glu-115-taa	0.0000	0.0000	0.0072
Glu-115-tga	0.0000	0.0000	0.6391
Glu-145-taa	0.0000	0.0000	0.0224
Glu-145-tag	0.0000	0.0000	0.0610
Glu-145-tga	3.1135	1.6389	5.5713
Glu-31-taa	0.0000	0.0000	0.0134
Glu-31-tag	0.0000	0.0000	0.0822
Glu-31-tga	7.1517	4.6316	0.6052
Glu-90-taa	0.0000	0.0000	0.3587
Glu-90-tag	0.0000	0.0000	0.0568
Glu-90-tga	19.2487	14.4003	3.0498
Glu-95-taa	0.0000	0.0000	0.0024
Glu-95-tag	0.0000	0.0000	0.0000
Glu-95-tga	0.0000	0.0000	2.5026
Gly-160-taa	0.0000	0.0000	0.0000
Gly-160-tag	0.0000	0.0000	0.0000
Gly-160-tga	33.9755	21.5681	0.0000
Gly-21-taa	0.0000	0.0000	0.0000
Gly-21-tag	0.0000	0.0000	0.0000
Gly-21-tga	0.0000	0.0000	0.0000
Gly-36-taa	0.0000	0.0000	0.0000
Gly-36-tag	0.0000	0.0000	0.0000
Gly-36-tga	7.1570	8.9036	0.0078
His-205-taa	0.0000	0.0000	0.0099
His-205-tag	0.0000	0.0000	0.0287
His-205-tga	0.0000	0.0000	1.6275
His-26-taa	0.0000	0.0000	0.0070
His-26-tag	0.0000	0.0000	0.0635
His-26-tga	0.0000	0.0000	1.6590
Ile-120-tag	0.0000	0.0000	0.0056
Ile-120-tga	0.0000	0.0000	0.0022
Ile-61-tag	0.0000	0.0000	0.0029
Ile-80-taa	0.0000	0.0000	0.0037
Ile-80-tag	0.0000	0.0000	0.0148
Ile-80-tga	1.4096	1.0796	0.9731
Leu-125-taa	0.0000	0.0000	0.0047
Leu-125-tag	0.0000	0.0000	0.0020
Leu-125-tga	0.0000	0.0000	0.5865
Leu-175-taa	0.0000	0.0000	0.0085
Leu-175-tag	0.0000	0.0000	0.0004
Leu-175-tga	1.3221	0.2406	0.6199
Leu-200-taa	0.0000	1.4442	0.0000
Leu-200-tag	0.0000	0.0000	0.0000
Leu-200-tga	2.4623	0.0000	0.0000

Lys-140-taa	0.0000	0.0000	0.0032
Lys-140-tag	0.0000	0.0000	0.0593
Lys-140-tga	11.2558	8.6891	2.0226
Lys-16-taa	0.0000	0.0000	0.0883
Lys-16-tag	0.0000	0.0000	0.2042
Lys-16-tga	0.0000	0.0000	0.3331
Lys-185-taa	0.0000	0.0000	0.0000
Lys-185-tag	0.0000	0.0000	0.0050
Lys-185-tga	0.0000	0.0000	0.0030
Lys-46-tag	0.0000	0.0000	0.0320
Lys-46-tga	0.0000	0.0000	0.7955
Lys-75-taa	0.0000	0.0000	0.0077
Lys-75-tag	0.0000	0.0000	0.0067
Lys-75-tga	11.2602	3.1105	5.9544
Lys-85-taa	0.0000	0.0000	0.0022
Lys-85-tag	0.0000	0.0000	0.0198
Lys-85-tga	0.0000	0.0000	0.0006
Met-13-taa	0.0000	0.0000	0.0249
Met-13-tag	0.0000	0.0000	0.0057
Met-13-tga	9.4340	2.1921	0.0037
Met-19-taa	0.0000	0.0000	0.0434
Met-19-tag	0.0000	0.0000	0.0233
Met-19-tga	0.0000	0.0000	0.0024
Met-190-taa	0.0000	0.0000	0.0110
Met-190-tga	48.6070	0.0000	0.2116
Met-190-tga	74.7775	14.1293	0.2116
Phe-100-taa	0.0000	0.0000	0.0044
Phe-100-tag	0.0000	0.0000	0.0079
Phe-100-tga	0.0000	0.0000	0.0037
Phe-130-taa	0.0000	0.0000	0.0000
Phe-130-tag	0.0000	0.0000	0.0056
Phe-130-tga	0.0000	0.0000	5.2781
Pro-135-taa	0.0000	0.0000	0.2827
Pro-135-tag	2.6279	0.0000	5.1460
Pro-135-tga	82.5845	78.2258	78.6557
Pro-56-taa	0.0000	0.0000	0.0174
Pro-56-tag	0.0000	0.0000	0.0154
Pro-56-tga	0.0000	0.0000	2.7410
Thr-180-taa	0.0000	0.0000	0.0006
Thr-180-tag	0.0000	0.0000	0.0006
Thr-180-tga	0.0000	0.0000	0.0008
Thr-210-taa	0.0000	0.0000	0.1463
Thr-210-tag	0.0000	0.0000	0.2253
Thr-210-tga	0.0000	0.0000	0.2590
Trp-94-taa	0.0000	0.0000	0.0019
Trp-94-tag	0.0000	0.0000	0.0000
Trp-94-tga	73.3500	18.7394	57.5179

Table 1. Percentage of His-tag expression relative to the wild-type mScarlet considering only the upped band of the gels (first column) and all the bands (second column), and percentage of fluorescence signal measured as the median of fluorescence relative to the wild-type mScarlet. The reporters that we considered dark are highlighted in bold.

We changed the caption of figure S4 to “**Figure S4. Detection of stop codon miscoding analysing the His-tag expression with His-tag antibodies.**”

8. Fig S9: Some information are missing. Which medium was used? Which constructs (positions) were used? Is it always the same dataset of Fig S8? Is it the same dataset for all figure panels? In the last line of the figure legend “upstream” should be downstream”.

In **Fig. S7** and **S8**, for panels **A, B, D,** and **E**, we plotted the SCM score on the y-axis. The SCM score is an aggregated metric that summarises the results of the experiments done in minimal and rich medium and at 18, 25, 37 and 42°C (the dataset of **Fig. S2**). The SCM score was calculated as described in the method section entitled “*Stop codon miscoding likelihood score (SCM score)*”. Briefly:

Reporters with median fluorescence below a threshold (median plus two standard deviations of NC fluorescence) received a score of 0, while the highest signal reporter was assigned a score of 1.0. Reporters were ranked between 0 and 1.0 based on their median fluorescence. This ranking process was repeated for all stop codons under all the studied conditions. The average score across conditions and stop codons yielded a unique score per position and, therefore, per genome context.

We fixed the **Fig. S7** and **S8**.

9. Fig S12: The quality of some spectra is not great and some important information are missing. Were the samples cultivated at 18 or 37 °C? The peptide sequence could be better annotate by indicating the inserted amino acid at the stop codon position, as well as the extent of coverage of the stop codon region by the peptide. Moreover, the precursor mass should be displayed in the extracted ions chromatograms.

We agree that the identification of SCM peptides often relied on a few fragment ions. However, due to the concurrent isolation of many precursors, signals acquired using data-independent acquisition can usually not be expected to be comparable with data-dependent acquisition or targeted measurements. We consider the co-elution of several specific fragment ions, along with the matching precursor mass and a match between predicted and measured retention time to be sufficient evidence. Identical samples were grown at both 37 and 18°C and then measured. Information about which peptides were detected in which condition is given in Table S2. We thank the reviewer for the suggestion to add more information. We now highlight the amino acid at the Stop codon, which also serves to identify the regions before and after the Stop codon covered by the peptide. We also added the precursor m/z and charge as given in DIA-NN's identification table to exactly specify the detected precursor. Due to mass errors, the actual detected precursor m/z differs from sample to sample, but all mass errors (for all peptides included in the analysis, for all samples) were below 10 ppm.

Discussion:

10. The text of the discussion became rather confusing after revision. In part, it is just repetition of the results. The important discrepancy concerning the nature of the favoured +1 nucleotide is not addressed. The evolutionary arguments are all unwarranted, as there is no evolutionary data in the Results. The discussion on the roles of secondary structures is confusing, as the authors state them as a possible regulator of RT, whereas their own analysis shows no correlation. (There are also different opinions about how the mRNA secondary structure elements affect frameshifting. The authors cite a single old paper, while there is a large body of more recent literature which doubts the role of thermodynamic stability alone. This is again the example of limited use of the available literature). The authors also mention the biological roles of protein isoforms as studied in *Drosophila*, but the coverage of the literature is poor, so the point is not well made.

We addressed the discrepancy concerning the nature of the favoured +1 nucleotide in the results section (line 279): “This result agrees partially with a pioneer work that shows how T and C favour termination efficiency⁴³. However, it differs from previous observations in mammalian cells where the nucleotide at the +4 position increased readthrough in the following order C > U > A > G³¹.”

Regarding the evolutionary discussion, see answer to question #12.

We now expanded the discussion on the roles of secondary structures.

We added this paragraph to the discussion section (line xxx): “At least in the context of ribosomal frameshifting, the effect of mRNA structure on ribosome movement appears to depend not only on thermodynamic stability but also on the exact distance between structure and ribosome^{44,53}, the size of the structure⁵⁴ and the number of possible conformations^{55,56}. These more complex relationships may not be captured by prediction methods.”

We included pioneer work of Isaksson to the discussion section (line 494): “On the other hand, it has been reported that the amino acid identity at the C-terminal of the nascent peptide modulates termination accuracy^{38–40}. Proline and glycine in the -1 and -2 positions upstream of the stop codon increases termination inaccuracy³⁹. Interestingly, codons encoding for proline and glycine are enriched in C and G. Thus, the observed correlation between GC content upstream of the stop codon and the fidelity of termination might be a consequence of the amino acid effect on the termination.”

Limited literature exists describing the biological implications of SCM events. Most available references discussing functional aspects related to termination errors primarily focus on ribosomal readthrough at a mRNA slippery sequence near a stop codon. Our current focus diverges from these topics, as we concentrate specifically on a different error in the termination process.

11. Discussion and reviewer 3 response: the observation that many of the error-prone stop codons are located in polycistronic mRNAs is interesting, however one should also consider the presence of additional stop codons in the intergenic regions and in the two adjacent genes are in frame.

We thank the reviewer for this suggestion, and we followed it: Analogous to the previous analysis where we compared polycistronic and non-polycistronic genes, we compared the stop codons of ‘fusing genes’ with all other genes. Fusing genes are those where an SCM event could potentially fuse two genes. These genes are followed by another gene within the same operon, in the same frame, and without an in-frame Stop codon in between. We see that among the fusing genes, TGA is underrepresented and TAA is overrepresented (see Fig. 3 below). So in contrast to the implications of the previous analysis, this result shows that there might have been selection for stronger stop codons (as TAA is more reliable than TGA) for those genes where SCM would have a large effect. It remains true that TGA is overrepresented among the polycistronic genes, but this does not seem to be the case to allow the fusing of genes by SCM. It is interesting to think about other reasons why the distribution of stops within operons is different from that of single genes, but currently, we cannot speculate.

In the new version of the manuscript we removed the hypothesis that SCM events in multi gene operons might result in the expression of fused protein complexes.

Response Figure 3. Among genes susceptible to fusion through a SCM event, TGA, the most error prone stop codon, is underrepresented while TAA, the most accurate stop codon, overrepresented.

Throughout the text:

12. Most of the evolutionary discussion is not warranted. The authors suggest that a general increase of translation error rate could be beneficial for the cell under stress conditions (lines 216-218 and Discussion). The majority of errors, whether arising from DNA mutations, transcription, or translation, tend to be either detrimental or neutral to the cell. It is more reasonable to claim that recoding events could have an adaptive function only in some specific cases/genes. In other words, each case should be considered individually. Furthermore, the authors use the high frequency of recoding as an argument in favour of adaptivity of SCM (lines 454-458). Evolution is a complex phenomenon and a high frequency of a genetic trait cannot be used alone to support adaptation. For this purpose, there are specific computational methods (e.g. dN/dS). One possible strategy could involve examining whether mRNA sequences encoding the C-terminal extensions of genes exhibiting high rates of readthrough demonstrate notably elevated or diminished dN/dS values.

We agree that the adaptive potential of SCM events is a hypothesis sparked by the current work and previous works we cite and would need further analysis and experimental validation. It could be an extensive follow-up work. Here, we only mention this as an outlook in the discussion.

Minor points:

1. Line 31: "termination is optimized": I suggest refraining from using this passive form. In my opinion, it imparts an 'intelligent design' (or even creationist) connotation. Genetic traits arise as outcomes of mutational pressure, genetic drift, and selection.

We agree and changed it to termination is optimal.

2. Line 106-107: what was the inducer concentration in this experiment (2.1% rate)?

We used 400 mg/L AHT.

We included in results (line 107): "at 400 mg/L AHT concentration,"

3. Line 142: "inefficient RF2" should be "inefficient RF2 (or better prfB) allele"

We fixed it.

4. Line 143: There are numerous wild-type E. coli strains. Refer to it as K12 MG1655 strain.

Added.

5. Line 203: "minimum media" should be "minimal media" (or better medium)

We changed it.

6. Line 285: "downstream" should be "upstream"

We changed it.

7. Line 288: how and why was the ATTAT sequence chosen?

We focused on those reporters with no detectable SCM events in any of the conditions studied. From these reporters, we selected the one with the highest T content in the 5-nt window upstream of the stop codon. With these criteria we selected the reporters that introduced a stop codon in position 85 and the upstream region is ATTAT.

We added to the result section (line xxx): "because, according to our analysis, T reduces the likelihood of SCM events when placed after the stop codon. We mutated the upstream region towards ATTAT"

because T reduces the likelihood of SCM events when placed in a 5-nt window before the stop codon. This sequence is the richest in T content among reporters that did not exhibit SCM events in any of the studied conditions”

8. Line 295-296: why “4-nt upstream”. Should not it be 5-nt?

We changed it.

9. Line 555: “in LB-agar plates” should be “on LB-agar plates”

We changed it.

10. Lines 691-693: Why wasn't a spacer upstream of the stop codon taken into account for the structural predictions?

It was not trivial how to account for the presence of the ribosome. The helicase activity at the ribosome entry has been investigated and the mRNA is generally assumed to be linearised within the ribosome. But the picture is muddled by the fact that an mRNA molecule is usually translated by many ribosomes simultaneously, which should suppress most larger structures. However, it is known that these structures can have a strong effect on translation (e.g., frameshifting in viruses). So there is (to our knowledge) no clear picture of how exactly the ribosome interacts with mRNA structure. For downstream structures, we took cues from other work using spacers, but there was no such guidance for upstream structures. Instead of making too many assumptions, we chose to simply look for correlations between SCM and predicted structures in the vicinity of the Stop codon.

Reviewer #2 (Remarks to the Author):

We thank the authors for providing a revised manuscript that responds to each of our comments and concerns.

There is one additional point that we believe the authors should address in the manuscript: the influence of Rho-mediated premature transcription termination and subsequent RNA decay on the population of mRNAs and transcriptional errors that is observed. The influence of Rho should be discussed in the context of the transcriptional errors reported (Figure 4), and the authors should include as well the argument made in their rebuttal that the preponderance of transcriptional errors resulting in other stop codons (rather than sense codons) at PTCs clearly indicates that there are additional mechanisms at play. This will substantially help readers who may otherwise be skeptical based on the function of Rho-mediated transcription termination of PTC-carrying mRNAs in *E. coli*.

We agree with the reviewer and added to the result section (line 360) the following: “The observed increase of mismatches may be the result of the selective degradation of mRNA containing premature stop codons^{49,50}, e.g. by Rho-mediated transcription termination of premature stop codon-carrying mRNAs. However, the prevalence of synonymous mismatches leading to another stop codon (Fig. 4E) suggests a higher RNA polymerase error rate at these stop codons.”

Otherwise, we believe the revisions address the points we raised, and that the resulting manuscript is substantially improved.

We are pleased to see that the points were sufficiently addressed and we would like to thank again for the useful comments.

We provide just a few minor points for additional corrections where typos in the text and figure panels confuse the apparent intended meaning:

- Fig. 2C caption: Use of “media” where “median” was presumably intended.
- Fig. 3D caption: repeated reference is made to TAA and TAG stop codons commonly being miscoded as tryptophan; whereas based on the data in Fig. 3D and Table S1 it appears the authors meant to refer to tyrosine in these cases (TGA, on the other hand, indeed is commonly miscoded as tryptophan).
- Fig. 4B caption: Caption is unclear. Panel 4B does not seem to have any bearing on nucleotide context as described in the caption.
- Line 211: Apparent stray “Although” at the start of this sentence – presumably a typo?

Answer: Thank you for the thorough reading of our manuscript. We fixed the typos and edited Fig. 4 caption.

Reviewer #3 (Remarks to the Author):

Overall, I find the reviewers did a good job of answering my critiques and find the changes they made to the paper improve the manuscript. The key finding seems to be that temperature can affect the readthrough rate as measured by both fluorescence and western blots. They now directly show that the nucleotide sequence surrounding the stop codon influences readthrough at these positions.

I wonder if Figure S4 could be included in Figure 2, as it seems to be a better presentation of the data and is more easily interpretable by the reader.

This is a great suggestion. We now included Figure S4 in Figure 2.

I don't want to ask them to do more experiments at this phase, but in general I would be interested to know if the same effects are seen in log-phase at low temperature, as cells grown to saturation are nutrient limited, hence why their growth is saturated. Do cells lose this heterogeneity of read-through in log phase, or is there no read through at all even at low temperatures?

It is indeed an interesting question what the SCM rates are in log-phase vs. stationary phase. Unfortunately, we have not measured fluorescence of cells in log-phase, only in stationary phase. However, we expect an increase in the SCM rate in the stationary phase when the nutrient depletion effect is intensified.

That saying, Wentzel *et al.* studied the effect of the growth phase on stop codon ribosomal readthrough, ie., frameshift of the ribosome after a slippery sequence located right before a stop codon, and they showed that at stationary phase, readthrough is lower than during active growth (Wentzel, Stancek, and Isaksson 1998). However, Mordret *et al.*, found the opposite trend when studied amino acid misincorporation proteome wide, they observed an increase in the rate of mistranslation as cells approached stationary phase (Mordret et al. 2019).

References

- Fan, Yongqiang, Christopher R. Evans, Karl W. Barber, Kinshuk Banerjee, Kalyn J. Weiss, William Margolin, Oleg A. Igoshin, Jesse Rinehart, and Jiqiang Ling. 2017. "Heterogeneity of Stop Codon Readthrough in Single Bacterial Cells and Implications for Population Fitness." *Molecular Cell* 67 (5): 826-836.e5.
- Mordret, Ernest, Orna Dahan, Omer Asraf, Roni Rak, Avia Yehonadav, Georgina D. Barnabas, Jürgen Cox, Tamar Geiger, Ariel B. Lindner, and Yitzhak Pilpel. 2019. "Systematic Detection of Amino Acid Substitutions in Proteomes Reveals Mechanistic Basis of Ribosome Errors and Selection for Translation Fidelity." *Molecular Cell* 75 (3): 427-441.e5.
- Wentzel, A. M., M. Stancek, and L. A. Isaksson. 1998. "Growth Phase Dependent Stop Codon Readthrough and Shift of Translation Reading Frame in Escherichia Coli." *FEBS Letters* 421 (3): 237-42.

REVIEWER COMMENTS

Reviewer #1 (Remarks to the Author):

The authors have addressed some minor issues, but the major ones remain critical. These critical issues concern the data reproducibility, which is the key issue for an experimental paper. The author's attitude that this is "not possible due to time constraints" (from the authors' response) is not satisfactory, as the data must be reproducible. There are also many problems with the text. Taken alone, each of them is minor, but together they make an impression of sloppiness in dealing with the text and the data. The authors should take this criticism seriously, because the poor readability of the paper and using unusual terminology diminish the potential impact of the paper.

Specifically, the following major problems still require attention.

1. The reporter assay and sample-to-sample variation of measurements at different temperatures.

In response to my criticism, the authors state the following:

"Regarding the differences among experiments conducted at 25°C, we acknowledge a high level of SCM cell-to-cell variability. In fact, in Fig S4, we observed SCM error rates at position 155TGA (25°C) ranged from 3.9% to 12%. These differences increase from 0-1% to 12% when comparing experiments conducted independently (i.e., on different days, with variations in media preparation, growth time, etc.)."

But this is exactly what replicates and statistics are good for: To distinguish biological patterns from variability among replicates! This is an essential issue that must be solved before the paper can be published.

In order to assert that the frequency of RT increases at low temperatures and to define temperature trends, they need to provide replicates and estimate sample variability. In fact, the reporter assay exhibits considerable variability with non-normal distributions (which is fine! It happens often in in-vivo reporter assays). Extrapolating trends from a single replicate is not feasible when the assay inherently displays high variability.

Importantly, if we cannot rely on the dataset of Fig S2 (erroneously stated as S4 in the point-by-point reply), it follows that we cannot rely on the conclusions drawn from Fig S7 and S8.

I suggest to use only the positions that were tested in triplicates or quadruplicates (Positions 105 and 155 - New Fig 2C) to test their temperature dependency hypothesis (they could use a t-test or a non parametric test). The statistical test should encompass ALL replicates (4 in total), including those from Fig2C and FigS2/Fig1E/Fig2A,B.

In the response, they also say:

"Additionally, we reanalysed the 25°C data set with a different threshold to differentiate between background fluorescence and fluorescence signal due to SCM events. Previously, we defined the threshold as the median plus two standard deviations of the fluorescence signal of the negative control (NC). Since we included two biological replicates of NC, we chose the highest threshold. Figure 1A shows the fluorescence distribution of the reporters with the TGA stop codon. At the right side of the distribution is shown the percentage of fluorescence signal compared with the positive control (PC) of the distribution with a median above the defined threshold.

For this new analysis, we defined the threshold as the median of fluorescence of the NC and we also chose the value of the highest threshold. Figure 1B below shows the fluorescence distribution of the reporters with the TGA stop codon. At the right side of the distribution is shown the percentage of fluorescence signal compared with the positive control (PC) of the distribution with a median above

the defined threshold”.

This has not much to do with the reproducibility issue. The problem does not lie with the threshold they have chosen, this is not a valid argument. There are still significant differences in SCM rates between FigS2 and Fig2C at positions 105TGA and 155TGA (at 25°C), almost one order of magnitude. I would expect that normalising by the median of the positive control would, at least partially, diminish the variability among experiments. However, it does not appear to be effective.

There is a high variability in the number of considered cells:

FigS2 legend: “Each distribution is derived from one replicate and 21 to 37029 cells.”

FigS4 legend: “Each distribution is derived from one replicate and 20 to 10967 cells.”

The number of cells considered for each distribution are missing in the legend of Fig 1 and 2.

Which distributions were derived from only 20 cells? It seems like a relatively small number compared to 37029 and 10967.

Why are some replicates missing in the new Fig2C?

In FigS3, it would be helpful to include an explanation of the y-axis label (% of fluorescence) in the figure legend. I assume it represents fluorescence relative to the positive control, but this is not explained. Furthermore, in the legend (line 82), it should be error-pRone.

The problem of the temperature control should be explicitly addressed in the Methods section, for example by providing the information

“All experiments were done at controlled temperatures. In our institute we have rooms with constant controlled temperatures at 18, 25, 37 and 42°C”.

2. Data from Western blots and dark reporters

- The authors acknowledge that the WB data should be interpreted as qualitative. This should be explicitly stated in the main text.

- To document the quality of WB, I suggest to include the full gel pictures in the supplementary information. These pictures are way more informative than the current Fig S5.

- Considering that the additional bands might be degradation products, I find more appropriate the quantification using all bands. I suggest including the table from the response letter directly into the paper.

- There are still discrepancies in what the author claim and what it is represented in the current Fig S6B. They say that four reporter out of 117 were considered dark. However, in Figure S6B there are 5 dark reporters.

- How was the discrepancy between the WB data and the fluorescence data calculated? Which equation was used? I do not understand why some of the reporters where not classified as “dark” (e.g. Ala-105-taa, Glu-11-taa, Glu-31-tga, Gly-36-tga, Lys-140-tga, Met-13-tga)

- Furthermore, according to the data in the table (in authors’ response), Pro-135-tga should not be considered dark, but it is in Fig S6B.

- One of the two Met-190-tga samples in the response table should be actually Met-190-tag.

- Shouldn’t Met-190-tga be considered dark as reported in Fig S6B?

- The meaning of the colors should be explained in Fig S6 legend.

3. The mass spectrometry data

One of the reasons why I consider the proteome-wide MS data as 'preliminary' is well illustrated in the authors' response:

"We agree that the identification of SCM peptides often relied on a few fragment ions. However, due to the concurrent isolation of many precursors, signals acquired using data-independent acquisition can usually not be expected to be comparable with data-dependent acquisition or targeted measurements."

Another reason is that the masses of the peptides they discovered should undergo some form of validation, such as utilizing aqua peptides or prediction tools for spectra/retention time.

4. Terminology and text.

Most authors in the field take great care to precisely define what they study, namely stop codon readthrough and I don't see why this should be renamed. I am not satisfied with the authors' idea to lump together transcription and translation errors. This should be strictly avoided by using different terms and clearly explained in the text. Also, while the authors indicated that they changed the term "infidelity" to "error frequency", this was not done throughout; instead, the authors just used "inaccuracy", which is as bad and imprecise as "infidelity". This will not help for the impact and makes the reading quite hard.

- L. 31 "...is accurate to achieve high fidelity". The sentence is obviously redundant and has to be revised.
- L. 36. Again, readthrough is an error of misreading, not of miscoding. If, instead, a polymerase makes a mistake, it is not a readthrough error, and these two levels (transcription and translation) should be clearly distinguished.
- Throughout the text: Once the term SCM is introduced, it should be consistently used.
- L. 57 and throughout: replace "inaccuracy" by "error frequency" or "error rate".
- Strictly speaking, an mRNA codon is UGA, not TGA. This dilemma is not solved in the paper, which, again, makes it a difficult read.
- L. 59: "how frequent errors are": please correct grammar
- L. 68: "We thus propose": why "thus"? This is entirely unclear in this context.
- The authors claim higher readthrough error frequency than expected, but actually do not compare them with the previously data published data. The numbers from the previous publications should be explicitly cited.
- L. 186; "Despite this complex scenario, SCM events are consistent among biological replicates". This statement contradicts to the text in l. 166-173 and to the reply to reviewers.

Response to the reviewer's comments

Dear Reviewer,

Thank you very much for your consideration and comments.

Below, please find below our point-by-point response to the reviewers' comments.

Please find attached the revised manuscript, including the required additions, corrections, and clarifications.

Reviewer #1 (Remarks to the Author)

The authors have addressed some minor issues, but the major ones remain critical. These critical issues concern the data reproducibility, which is the key issue for an experimental paper. The author's attitude that this is "not possible due to time constraints" (from the authors' response) is not satisfactory, as the data must be reproducible. There are also many problems with the text. Taken alone, each of them is minor, but together they make an impression of sloppiness in dealing with the text and the data. The authors should take this criticism seriously, because the poor readability of the paper and using unusual terminology diminish the potential impact of the paper.

Specifically, the following major problems still require attention.

1. The reporter assay and sample-to-sample variation of measurements at different temperatures.

In response to my criticism, the authors state the following:

"Regarding the differences among experiments conducted at 25°C, we acknowledge a high level of SCM cell-to-cell variability. In fact, in Fig S4, we observed SCM error rates at position 155TGA (25°C) ranged from 3.9% to 12%. These differences increase from 0-1% to 12% when comparing experiments conducted independently (i.e., on different days, with variations in media preparation, growth time, etc.)."

But this is exactly what replicates and statistics are good for: To distinguish biological patterns from variability among replicates! This is an essential issue that must be solved before the paper can be published.

In order to assert that the frequency of RT increases at low temperatures and to define temperature trends, they need to provide replicates and estimate sample variability. In fact, the reporter assay exhibits considerable variability with non-normal distributions (which is fine! It happens often in in-vivo reporter assays). Extrapolating trends from a single replicate is not feasible when the assay inherently displays high variability.

Importantly, if we cannot rely on the dataset of Fig S2 (erroneously stated as S4 in the point-by-point reply), it follows that we cannot rely on the conclusions drawn from Fig S7 and S8.

I suggest to use only the positions that were tested in triplicates or quadruplicates (Positions 105 and 155 - New Fig 2C) to test their temperature dependency hypothesis (they could use a t-test or a non parametric test). The statistical test should encompass ALL replicates (4 in total), including those from Fig2C and FigS2/Fig1E/Fig2A,B.

We agree that it is crucial to show that the temperature effect that we repeatedly observed is significant. To strengthen our conclusions, we provide statistical evidence for both the cell-to-cell variability and the temperature effect.

First, we compared the 3 replicas shown in Fig 2C using the Friedman test, a non-parametric analysis suitable for non-normally distributed data. To meet the requirements of the Friedman test, which necessitates equal-sized samples, we assessed the maximum number of cells available across replicas. We excluded replicas with fewer than 200 cells from the analysis. In cases where only 2 replicas were present, the Friedman test is inappropriate, so we used the sign test. The obtained statistics confirmed the dissimilarity between the replicas (Table S1), as previously acknowledged in our last response and in pertinent studies (Fan et al. 2017).

The key question arises: Is the observed variability between experiments conducted at different temperatures attributable to a temperature effect or cell-to-cell variability? In order to address this question, we performed the Wilcoxon test, a non-parametric method suitable for non-normally distributed data, as an alternative to the t-test. We first pooled the replicas together as a dataset, sampling each replica equally to avoid that only one replica biases the results. Likewise, for the Friedman test, the Wilcoxon test also requires equal-sized replicates. We assessed the maximum number of cells available across replicas and excluded those with fewer than 200 cells. The Wilcoxon test confirmed that when a stop codon was inserted at positions 105 and 155, in addition to observing significant variability within the dataset, there is also evidence supporting the existence of a temperature effect (Fig S3, Table S2). Specifically, our data reveals a significant increase in the SCM rate as the temperature decreases, particularly evident at 18°C. Notably, this effect becomes more pronounced with larger magnitudes of the SCM rate. It is also important to note that the temperature effect may be less apparent than the variability observed between replicates when the SCM rate is low. Nonetheless, our findings provide statistical evidence of a temperature influence on SCM events when a stop codon is inserted at positions 105 and 155: 18°C>25°C>30°C~37°C>42°C.

To assess the generality of the temperature effect, we applied the Wilcoxon test to all the studied reporters under all experimental conditions. We assessed the median of fluorescence relative to the WT, excluding those reporters that showed no SCM at any of the studied temperatures. The Wilcoxon test shows statistical evidence of a non-linear temperature-driven effect on SCM rate: 18°C>25°C~37°C>42°C (Fig S4, Table S4)

Position	Stop codon	T (°C)	Friedman	Degree of	p-value
105	TAA	18	832.49	2	< 2.2e-16
105	TAG	18	52.338	2	4.315e-12
105	TGA	18	2277.7	2	< 2.2e-16
105	TAA	25	0.31694	1	0.5734
105	TAG	25	42.497	2	5.914e-10
105	TGA	25	72.741	2	< 2.2e-16
105	TAA	30	21.835	2	1.813e-05
105	TAG	30	70.029	2	6.215e-16
105	TGA	30	72.824	1	< 2.2e-16
105	TAA	37	18.638	1	1.58e-05
105	TAG	37	31.358	2	1.551e-07
105	TGA	37	25.535	2	2.851e-06
105	TAA	42	1927.5	2	< 2.2e-16
105	TAG	42	113.33	2	< 2.2e-16
105	TGA	42	436.02	2	< 2.2e-16
155	TAA	18	1311.9	2	< 2.2e-16
155	TAG	18	441.7	2	< 2.2e-16

155	TGA	18	943.45	1	< 2.2e-16
155	TAA	25	7.4871	2	0.02367
155	TAG	25	41.11	2	1.183e-09
155	TGA	25	236.3	1	< 2.2e-16
155	TAA	30	-	0	-
155	TAG	30	2.4378	1	0.1184
155	TGA	30	16.124	1	5.932e-05
155	TAA	37	51.265	2	7.38e-12
155	TAG	37	30.643	2	2.218e-07
155	TGA	37	-	0	-
155	TAA	42	295.2	2	< 2.2e-16
155	TAG	42	53.465	2	2.456e-12
155	TGA	42	199.69	2	< 2.2e-16

Table S1: Friedman (when comparing 3 replicas) and sign (when comparing 2 replicas) tests confirm a high level of cell-to-cell variability, indicating significant differences among biological replicates. Replicas with fewer than 200 cells were excluded from the analysis. Empty cells represent instances where only one replica displayed more than 200 cells, and the test could not be performed.

Fig S4. Wilcoxon test shows statistical evidence of a temperature-driven effect on SCM: 18°C>25°C>30°C~37°C>42°C when a stop codon is inserted at positions 105 and 155. Cell counts were maximized across replicas, excluding those with <200 cells. The test evaluated if one cell distribution significantly exceeded another. Temperature decrease correlates with increased SCM rates, especially notable at higher SCM rates (i.e., at 18°C and with TGA stop codon).

Ala-105-TGA	n1	n2	statistic	p	p.adj	p.adj.signif
18>25	1188	1188	612277	1.05E-106	1.05E-105	****
18>30	1188	1188	682781	3.06E-171	3.06E-170	****
18>37	1188	1188	691100	7.14E-180	7.14E-179	****
18>42	1188	1188	657809	1.28E-146	1.28E-145	****

25>30	1188	1188	541192	3.19E-57	3.19E-56	****
25>37	1188	1188	553799	7.4E-65	7.4E-64	****
25>42	1188	1188	485879	1.57E-29	1.57E-28	****
30>37	1188	1188	361295	0.245	1	ns
30>42	1188	1188	328111	0.983	1	ns
37>42	1188	1188	312479	1	1	ns
Asp-155-TGA	n1	n2	statistic	p	p.adj	p.adj.signif
18>25	731	731	153135	0.000349	0.003	**
18>30	731	731	188044	1.03E-21	1.03E-20	****
18>37	731	731	205217	3.32E-36	3.32E-35	****
18>42	731	731	238965	4.68E-76	4.68E-75	****
25>30	731	731	209597	1.59E-40	1.59E-39	****
25>37	731	731	231594	4.61E-66	4.61E-65	****
25>42	731	731	248095	1.96E-89	1.96E-88	****
30>37	731	731	186012	2.94E-20	2.94E-19	****
30>42	731	731	229364	3.51E-63	3.51E-62	****
37>42	731	731	204646	1.16E-35	1.16E-34	****
Ala-105-TAG	n1	n2	statistic	p	p.adj	p.adj.signif
18>25	3150	3150	4762079	0	0	****
18>30	3150	3150	4946163	0	0	****
18>37	3150	3150	4949600	0	0	****
18>42	3150	3150	4947129	0	0	****
25>30	3150	3150	2658833	0.000255	0.003	**
25>37	3150	3150	2374101	0.982	1	ns
25>42	3150	3150	3624003	2.9E-111	2.9E-110	****
30>37	3150	3150	1904772.5	1	1	ns
30>42	3150	3150	3433016	7.42E-78	7.42E-77	****
37>42	3150	3150	3633638	4.14E-113	4.14E-112	****
Asp-155-TAG	n1	n2	statistic	p	p.adj	p.adj.signif
18>25	434	434	70856	7.24E-20	7.24E-19	****
18>30	434	434	77709	9.09E-32	9.09E-31	****
18>37	434	434	59317	1.78E-06	1.78E-05	****
18>42	434	434	80741	5.61E-38	5.61E-37	****
25>30	434	434	55492	0.000756	0.008	**
25>37	434	434	37662	1	1	ns
25>42	434	434	76355	3.5E-29	3.5E-28	****
30>37	434	434	32807	1	1	ns
30>42	434	434	75328	2.68E-27	2.68E-26	****
37>42	434	434	77743	7.8E-32	7.8E-31	****
Ala-105-TAA	n1	n2	statistic	p	p.adj	p.adj.signif
18>25	729	729	177737	1.95E-15	1.95E-14	****
18>30	729	729	220320	1.93E-53	1.93E-52	****
18>37	729	729	177688	2.09E-15	2.09E-14	****
18>42	729	729	172009	3.67E-12	3.67E-11	****
25>30	729	729	162378	1.25E-07	1.25E-06	****

25>37	729	729	122808	0.964	1	ns
25>42	729	729	133324	0.48	1	ns
30>37	729	729	65377	1	1	ns
30>42	729	729	92722	1	1	ns
37>42	729	729	122043	0.973	1	ns
Asp-155-TAA	n1	n2	statistic	p	p.adj	p.adj.signif
18>25	1273	1273	649876	8.99E-78	8.99E-77	****
18>30	1273	1273	596193	3.42E-48	3.42E-47	****
18>37	1273	1273	433201	0.017	0.172	ns
18>42	1273	1273	670695	3.4E-91	3.4E-90	****
25>30	1273	1273	359172	1	1	ns
25>37	1273	1273	259891	1	1	ns
25>42	1273	1273	625587	1.72E-63	1.72E-62	****
30>37	1273	1273	339173	1	1	ns
30>42	1273	1273	651642	7.21E-79	7.21E-78	****
37>42	1273	1273	658220	5.07E-83	5.07E-82	****

Table S2: Wilcoxon test shows statistical evidence of a temperature-driven effect on SCM, analyzing the highest cell counts across replicas and excluding those with fewer than 200 cells. The test assessed whether one cell distribution was significantly greater than another. Temperature decrease leads to higher SCM rates in a non-linear fashion, especially notable at 18°C. This effect is more pronounced with higher SCM rates and may be less apparent at low SCM rates.

Fig S5. Wilcoxon test shows statistical evidence of a non-linear temperature-driven effect on SCM: 18°C>25°C~37°C>42°C. We assessed the median of fluorescence relative to the wt for all the reporters, excluding those with no SCM at any of the studied temperatures.

T (°C)	n1	n2	statistic	p	p.adj	p.adj.signif
18>25	102	102	3731	4.5E-11	2.7E-10	****
18>37	102	102	4002	1.69E-12	1.01E-11	****

18>42	102	102	4098	1.18E-14	7.08E-14	****
25>37	102	102	800.5	0.227	1	ns
25>42	102	102	938	3.84E-05	0.00023	***
37>42	102	102	577	6.46E-05	0.000388	***

Table S3. Wilcoxon test shows statistical evidence of a non-linear temperature-driven effect on SCM: 18°C>25°C~37°C>42°C. We assessed the median of fluorescence relative to the wt for all the reporters, excluding those with no SCM at any of the studied temperatures.

We included these analyses in the manuscript:

- Result section (line 175): “These experiments, along with their statistical analysis (see Methods section for further details), revealed two findings: i) significant cell-to-cell heterogeneity within a clonal population (Fig. 2C, 2D and Table S1), and ii) higher levels of SCM with decreasing temperature, particularly evident at 18°C, when a stop codon was inserted at positions 105 and 155 (Fig. S4 and Table S2). Previous studies have suggested that such heterogeneity may facilitate adaptation to changing environments (Fan et al. 2017). The observed heterogeneity poses challenges for quantifying SCM. Throughout this work, we utilized relative median fluorescence as a summary statistic to quantify SCM error rates. Nevertheless, these numeric values do not fully represent the distributions’ complexity, as they are neither normal nor symmetric.”

- Result section (line 195): “To assess the generality of the temperature effect, we statistically analyzed all reporters under all experimental conditions (see Methods section). We assessed the median of the fluorescence relative to the wild-type, excluding those reporters that showed no SCM at the studied temperatures. The analyses revealed evidence of a non-linear temperature-driven effect on SCM: 18°C>25°C~37°C>42°C (Fig. S5, Table S3).”

- Method section (line 620): “**Statistical Analysis**

Friedman and sign test

We employed the Friedman and sign tests to evaluate the statistical differences among the biological replicas depicted in Fig 2C. These tests are non-parametric and suitable for non-normally distributed data. The Friedman test is ideal for comparing more than two samples, while the sign test is suitable for comparing two samples. To satisfy the requirements of these tests, which demand equal-sized samples, we determined the maximum number of cells available across replicas. Replicas with fewer than 200 cells were excluded from the analysis (Table S1).

Wilcoxon test

To determine whether one distribution significantly exceeded another, we performed the Wilcoxon test. This non-parametric method is suitable for analyzing non-normally distributed data, serving as an alternative to the t-test. Similar to the Friedman and sign tests, the Wilcoxon test necessitates equal-sized replicates. We evaluated the maximum number of cells available across replicas and excluded those with fewer than 200 cells. Then, we pooled the replicas together as a dataset, ensuring equal representation from each replica to prevent bias in the results.

We further included Tables S1, S2, and S3, and Fig. S4 and S5.

In the response, they also say:

“Additionally, we reanalysed the 25°C data set with a different threshold to differentiate between background fluorescence and fluorescence signal due to SCM events. Previously, we defined the threshold as the median plus two standard deviations of the fluorescence signal of the negative control (NC). Since we included two biological replicates of NC, we chose the highest threshold. Figure 1A shows the fluorescence distribution of the reporters with the TGA stop codon. At the right side of the

distribution is shown the percentage of fluorescence signal compared with the positive control (PC) of the distribution with a median above the defined threshold.

For this new analysis, we defined the threshold as the median of fluorescence of the NC and we also chose the value of the highest threshold. Figure 1B below shows the fluorescence distribution of the reporters with the TGA stop codon. At the right side of the distribution is shown the percentage of fluorescence signal compared with the positive control (PC) of the distribution with a median above the defined threshold”.

This has not much to do with the reproducibility issue. The problem does not lie with the threshold they have chosen, this is not a valid argument. There are still significant differences in SCM rates between FigS2 and Fig2C at positions 105TGA and 155TGA (at 25°C), almost one order of magnitude. I would expect that normalising by the median of the positive control would, at least partially, diminish the variability among experiments. However, it does not appear to be effective.

The challenge here lies in distinguishing signal from noise within our data. Depending on the threshold defined, the fluorescence signal may be interpreted as either noise or SCM signal.

For instance, in Figures 2A and 2B, most reporters with TGA exhibit SCM rates within the 0-1% range at 37°C, while the majority show 0% SCM rate at 25°C. However, altering the threshold, such as using the median of the NC, alters the scenario. The table below illustrates the reporters that initially displayed a 0% SCM rate with the 2-zscore of NC as the threshold but now exhibit higher SCM rates.

That being said, the Friedman and sign statistical analyses (as discussed in answer #1) support the reviewer’s observation of significant differences between replicates, indicating cell-to-cell heterogeneity, as previously noted in the manuscript. However, the Wilcoxon statistical analysis (as discussed in answer #1) concludes that, despite this high variability among replicates, there is still a temperature influence on SCM events.

reporter	SCM rate (%)
Lys-140-tga	0.754442
Glu-31-tga	0.610429
Lys-46-tga	0.589922
Lys-16-tga	0.319777
Arg-150-tga	0.206733
Phe-130-tga	0.184244
Leu-175-tga	0.176778
Ala-165-tga	0.129128
Glu-6-tga	0.072813
Glu-90-tga	0.057605
Pro-56-tga	0.056038
Trp-94-tga	0.027558

There is a high variability in the number of considered cells:

FigS2 legend: “Each distribution is derived from one replicate and 21 to 37029 cells.”

FigS4 legend: “Each distribution is derived from one replicate and 20 to 10967 cells.”

The number of cells considered for each distribution are missing in the legend of Fig 1 and 2.

Which distributions were derived from only 20 cells? It seems like a relatively small number compared to 37029 and 10967.

We visually confirmed uniform optical density (OD) across all wells of the plate under a given experimental condition. Nonetheless, variations in OD were observed among different experimental

conditions, attributable to differences in growth rates influenced by temperature and media composition (refer to Table S8 for OD values). To standardize the cell densities for microscopy, we implemented specific dilution steps during plate preparation, diluting samples at ratios of 1:2500 for 37°C and 25°C, 1:1000 for 42°C, and 1:500 for 18°C.

Despite these adjustments, the variability observed in cell numbers within the images predominantly stems from the inherent challenges of maintaining a high-throughput setup. Specifically, the precise handling of minute volumes using a robotic system for coating the 384-well plate poses difficulties in achieving uniform cell distribution. Then, the automated imaging acquisition process, essential for maintaining efficiency, made the manual selection of well areas with a more consistent distribution of cells impractical, thus contributing significantly to the observed variability.

We extended the Microscopy Screenings and Sample Preparation section (line 601): **“We implemented different dilution steps in response to variations in growth rates (Table S8 and S9): 1:2500 for 37°C and 25°C, 1:1000 at 42°C, and 1:500 at 18°C for LB media, and 1:1250 for 37°C and 25°C, 1:500 at 42°C, and 1:100 18°C at M9 media.”**

We added Table S8 with OD values and Table S9 with the number of cells used to derive the fluorescence distributions

Temperature (°C)	media	Mean OD	SD OD
18	LB	0.67	0.21
25	LB	2.37	0.30
30	LB	2.16	0.51
37	LB	3.20	0.25
42	LB	1.99	0.71
18	M9	0.10	0.03
25	M9	0.35	0.32
30	M9	0.30	0.10
37	M9	0.56	0.04
42	M9	0.29	0.02

Table S8. OD values of mScarlet wild-type *E. coli* cultures grown in a 384-well plate without shaking under various temperature and media conditions, offering insights into growth dynamics in response to environmental factors. The data represent three biological replicates.

Position	Stop codon	Number of cells	Replica	Temperature (°C)
105	TAA	13055	1	18
105	TAA	15626	2	18
105	TAA	11252	3	18
105	TAG	15948	1	18
105	TAG	12041	2	18
105	TAG	9376	3	18
105	TGA	11737	1	18
105	TGA	11576	2	18
105	TGA	12435	3	18
105	TAA	5	1	25

105	TAA	1139	2	25
105	TAA	1425	3	25
105	TAG	1050	1	25
105	TAG	1955	2	25
105	TAG	1614	3	25
105	TGA	3221	1	25
105	TGA	2212	2	25
105	TGA	2374	3	25
105	TAA	4545	1	30
105	TAA	243	2	30
105	TAA	3122	3	30
105	TAG	3956	1	30
105	TAG	1454	2	30
105	TAG	5003	3	30
105	TGA	5006	1	30
105	TGA	4246	2	30
105	TGA	2567	3	30
105	TAA	22	1	37
105	TAA	2185	2	37
105	TAA	1289	3	37
105	TAG	4764	1	37
105	TAG	2228	2	37
105	TAG	5475	3	37
105	TGA	396	1	37
105	TGA	2402	2	37
105	TGA	2623	3	37
105	TAA	15509	1	42
105	TAA	7924	2	42
105	TAA	3763	3	42
105	TAG	7855	1	42
105	TAG	9704	2	42
105	TAG	1603	3	42
105	TGA	3413	1	42
105	TGA	8074	2	42

105	TGA	6330	3	42
155	TAA	1244	1	18
155	TAA	8036	2	18
155	TAA	10336	3	18
155	TAG	8776	1	18
155	TAG	2250	2	18
155	TAG	13268	3	18
155	TGA	4543	1	18
155	TGA	3572	2	18
155	TGA	4103	3	18
155	TAA	1244	1	25
155	TAA	8036	2	25
155	TAA	10336	3	25
155	TAG	3313	1	25
155	TAG	2250	2	25
155	TAG	4783	3	25
155	TGA	55	1	25
155	TGA	4168	2	25
155	TGA	2722	3	25
155	TAA	47	1	30
155	TAA	32	2	30
155	TAA	1273	3	30
155	TAG	217	1	30
155	TAG	1247	2	30
155	TAG	0	3	30
155	TGA	9	1	30
155	TGA	548	2	30
155	TGA	600	3	30
155	TAA	2850	1	37
155	TAA	4622	2	37
155	TAA	2581	3	37
155	TAG	12964	1	37
155	TAG	6192	2	37
155	TAG	1810	3	37

155	TGA	6	1	37
155	TGA	0	2	37
155	TGA	731	3	37
155	TAA	3765	1	42
155	TAA	4627	2	42
155	TAA	20948	3	42
155	TAG	12978	1	42
155	TAG	5313	2	42
155	TAG	3496	3	42
155	TGA	1633	1	42
155	TGA	1296	2	42
155	TGA	10744	3	42

Table S9. Number of cells imaged and used to derive the fluorescence distributions of Fig. 2C.

Why are some replicates missing in the new Fig2C?

Some replicates are missing because the acquired images had no or very few cells. We visually confirmed that all wells of the plate were grown. However, as previously explained, the challenges of working in a high-throughput fashion—such as coating a 394-well plate, handling minute volumes with robotic pipetting, and automating microscopy acquisition—make it difficult to avoid occasional images with very few or no cells.

In FigS3, it would be helpful to include an explanation of the y-axis label (% of fluorescence) in the figure legend. I assume it represents fluorescence relative to the positive control, but this is not explained. Furthermore, in the legend (line 82), it should be error-pRone.

We appreciate the reviewer's thorough examination and have revised the figure caption: **"Figure S3. Non-optimal growth temperatures and nutrient scarcity promote stop codon miscoding (SCM). A) More SCM events occur at lower temperatures, with TGA reporters displaying a higher incidence compared to TAG and TAA reporters. We considered those reporters with a median fluorescence higher than the threshold defined as the median fluorescence plus two standard deviations of the NC. B) Box plots summarising fluorescence distributions of *E. coli* cells expressing nine selected reporters grown under various conditions, highlighting: i) increased errors in minimum media (M9) compared to rich media (LB), ii) TAA as the most accurate stop codon and TGA as the least accurate, and iii) higher SCM events at low temperatures and in minimum media. The Y-axis represents the median fluorescence relative to the positive control (PC). C) mScarlet thermostability assay. mScarlet remained functional, i.e., fluorescent, until 70°C (mean and standard deviation of three replicates are shown)."**

The problem of the temperature control should be explicitly addressed in the Methods section, for example by providing the information

"All experiments were done at controlled temperatures. In our institute we have rooms with constant controlled temperatures at 18, 25, 37 and 42°C".

We extended the method section (line 598): “The cells were grown at different temperatures (18, 25, 37, and 42 °C) **at constantly controlled temperatures** until reaching saturation (24-48h) under light protection.”

2. Data from Western blots and dark reporters

- The authors acknowledge that the WB data should be interpreted as qualitative. This should be explicitly stated in the main text.

We added to the result section (line 249): “**Notably, we have considered the Western blot His-tag detection assay as qualitative analysis. Consequently, for downstream analyses, we relied on fluorescence measurements to quantify SCM events.**”

- To document the quality of WB, I suggest to include the full gel pictures in the supplementary information. These pictures are way more informative than the current Fig S5.

We modified the now Fig S7 caption: “**Figure S7. Detection of stop codon miscoding by analyzing the His-tag expression with His-tag antibodies in bands corresponding to the expected size for full-length mScarlet.**”

We further added a supplementary figure with the full gel pictures, Fig S8.

- Considering that the additional bands might be degradation products, I find more appropriate the quantification using all bands. I suggest including the table from the response letter directly into the paper.

We acknowledge that the additional bands smaller than the expected size of mScarlet wild-type may represent degradation products. Nonetheless, we focused solely on bands around 20 kDa because in cases where the SCM rate is low, the second band falls below the detection limit. As a result, our approach may underestimate His-tag detection for low SCM error rates.

The aim of these experiments was to detect SCM regardless of the amino acid inserted at the stop codon. This is because with the fluorescence reporter, detecting SCM depends on the sequence of the synthesized mScarlet protein. In other words, we aimed to identify dark reporters.

We initially identified five dark reporters: Gln-110-tga, Arg-150-tga, Gly-160-tga, Met-190-tag, and Met-190-tga, proposing the exclusion of positions 110, 150, 160, and 190 for subsequent analysis (Figure A below). We now repeated the downstream analysis without excluding any position (Figure B below). Given the robustness of the results across these variations, we propose not to exclude any reporters. Moreover, as noted by the reviewer, the definition of a dark reporter is arbitrary.

We now modified the section entitled **C-terminal His-tag detection as an orthologous method to quantify stop codon miscoding events.**

We also removed the previous Figs. S6B and changed Fig. 3A, 3B and 3C.

Comparison of the influence of nucleotide sequences surrounding premature stop codons on the probability of stop codon miscoding. A) Analysis excluding reporters introducing stop codons at positions 110, 150, 160, and 190. **B)** Analysis including all reporters. Given the robustness of the results across these variations, combined with the arbitrary nature of reporter exclusion, we propose not to exclude any reporters

- There are still discrepancies in what the author claim and what it is represented in the current Fig S6B. They say that four reporter out of 117 were considered dark. However, in Figure S6B there are 5 dark reporters.

Initially, four positions (110, 150, 160, and 190) were identified as dark based on the His-tag detection criteria (% median fluorescence - %His-tag signal > 30%). However, upon further analysis, an additional position, 135, was reclassified as dark following the study of the mScarlet mutant Pro-135-Trp, as detailed in line 248 of the main text and in Fig. S6B.

That said, considering the arbitrary nature of defining dark reporters and the complexity it adds to the analysis, we have decided to exclude this particular analysis from the manuscript.

- How was the discrepancy between the WB data and the fluorescence data calculated? Which equation was used? I do not understand why some of the reporters were not classified as "dark" (e.g. Ala-105-*taa*, Glu-11-*taa*, Glu-31-*tga*, Gly-36-*tga*, Lys-140-*tga*, Met-13-*tga*)

We classified a reporter as dark when the % median fluorescence - %His-tag signal was greater than 30%. However, in the revised version, we excluded this definition. This decision was made due to the arbitrary nature of defining dark reporters, the complexity it added to the analysis, and the robustness of the results from the downstream analysis of all reporters, including those previously classified as dark.

- Furthermore, according to the data in the table (in authors' response), Pro-135-tga should not be considered dark, but it is in Fig S6B.

Pro-135-tga was reclassified as dark following the study of the mScarlet mutant Pro-135-Trp, as detailed in line 233 of the main text and in Fig. S6B. However, in the revised version, we excluded the classification of dark reporters.

- One of the two Met-190-tga samples in the response table should be actually Met-190-tag.

We appreciate the reviewer's attention to detail and have corrected the typo as suggested.

- Shouldn't Met-190-tga be considered dark as reported in Fig S6B?

Yes, with our previous criteria both Met-190-tag and Met-190-tga should both be considered dark.

- The meaning of the colors should be explained in Fig S6 legend.

We removed Fig S6B from the revised version.

3. The mass spectrometry data

One of the reasons why I consider the proteome-wide MS data as 'preliminary' is well illustrated in the authors' response:

"We agree that the identification of SCM peptides often relied on a few fragment ions. However, due to the concurrent isolation of many precursors, signals acquired using data-independent acquisition can usually not be expected to be comparable with data-dependent acquisition or targeted measurements."

Another reason is that the masses of the peptides they discovered should undergo some form of validation, such as utilizing aqua peptides or prediction tools for spectra/retention time.

We appreciate the suggestion for additional validation in our proteome-wide SCM peptide identifications. The original identifications were done using the DIA-NN software, which already predicts fragmentation spectra and retention times and compares them to the data as part of its scoring algorithm. In addition, we manually inspected the MS2 signals of each peptide and excluded all that showed poor agreement with predicted spectra (Fig. S14).

To increase confidence in the identifications, we further utilized, as the reviewer suggested, the ProSIT software (<https://www.proteomicsdb.org/prosit/>)(Gessulat et al. 2019) to predict retention times for all identified peptides (canonical and SCM). To assess the reliability of these predictions, we compared them to the indexed retention times calculated by DIA-NN based on observed retention times (Fig. S18). The correlation is generally very good. However, there are some outliers, peptides that eluted much earlier than predicted by ProSIT, which we suspect to be false positives. One of the putative SCM peptides, CGHSQQK, was among these outliers. Consequently, we have excluded this peptide from the list of SCM identifications and from all tables and figures.

We extended the method section (line 913): "In addition, indexed retention times predicted by ProSIT were compared to measured (indexed) retention times, and outliers eluting much earlier or later than predicted were removed from the analyses (Fig S18). Peptides meeting all the filters described were further considered for analyses of differential abundance, sequence context, and stop codon usage."

We included Fig S18.

Fig. S18. Agreement between indexed retention times predicted by Prosit and indexed RT calculated from measured RT. For all precursors reported by DIA-NN, indexed retention times were predicted with Prosit. The plot shows the iRT predicted by Prosit vs. the iRT calculated by DIA-NN, as given in its main output table. Each dot represents a precursor; SCM precursors are shown in blue. The outlier SCM precursor in the bottom right part, eluting much earlier than predicted, was removed from the analysis.

4. Terminology and text.

Most authors in the field take great care to precisely define what they study, namely stop codon readthrough and I don't see why this should be renamed.

The reason to use a new terminology is as follows:

Our output is a discrepancy between the genomic sequence and the protein sequence.

This discrepancy can be due to:

- 1) Stop codon readthrough, an event where the ribosome incorporates in the protein sequence an amino acid, carried by the corresponding aa-tRNA, instead of a release factor that mediates release of the polypeptide sequence.
- 2) Transcriptional error, where an RNA-polymerase-DNA dependent incorporates in the nascent mRNA a nucleotide that does not correspond to the DNA sequence it is reading.

We believe that the most precise way to define what we study is to specifically talk about “stop codon readthrough” for the first case and “transcriptional error” for the second case. However, to describe the general discrepancy between the genomic sequence and the protein sequence, we feel that using the term “stop codon readthrough” is incorrect, as it fails to describe the second instance. For this reason, we decided to coin a new term that can generalize this occurrence. Our intention is not to have more impact, but to avoid misunderstandings in the interpretation of our data.

I am not satisfied with the authors' idea to lump together transcription and translation errors. This should be strictly avoided by using different terms and clearly explained in the text.

We think that using the term “Stop codon readthrough” to describe an event that might not involve a ribosomal readthrough (such as in the case of a transcriptional error) is misleading. We have now clearly defined what we mean with “SCM” in lines 33-38 and describe the difference between transcription and translation errors, justifying the need to coin a new umbrella term.

Line 33: “However, “stop codon miscoding” (SCM) may occur either by a transcription error, when the RNA polymerase misincorporates a nucleotide and eliminates the stop codon, or by a translation error in which the ribosome misincorporates a tRNA at the stop codon (also called stop codon readthrough and nonsense suppression^{5,6}). Thus, SCM covers transcription and translation events that deviate from the genetic code without identifying a specific mechanism.”

Also, while the authors indicated that they changed the term “infidelity” to “error frequency”, this was not done throughout; instead, the authors just used “inaccuracy”, which is as bad and imprecise as “infidelity”. This will not help for the impact and makes the reading quite hard.

- L. 31 “...is accurate to achieve high fidelity”. The sentence is obviously redundant and has to be revised.

We changed the introduction (line 31) “Protein synthesis termination can achieve high fidelity, yet it is not perfect.”

- L. 36. Again, readthrough is an error of misreading, not of miscoding. If, instead, a polymerase makes a mistake, it is not a readthrough error, and these two levels (transcription and translation) should be clearly distinguished.

We agree that transcription and translation origin of errors should be studied and distinguished. Since in this manuscript we did both and we know both mechanisms contribute, we prefer using stop codon miscoding that we define here as an umbrella term to describe both RT and transcription error, when we do not know the origin. When we do know that errors come from transcription error, we refer to it as “likelihood of the RNA polymerase mismatching”, and show plots of “%mRNA mismatches” e.g. in Fig. 4 and Fig S13.

- Throughout the text: Once the term SCM is introduced, it should be consistently used.

We have added the term SCM throughout the text (line 48, 50, 56, 70, 87, 91, 103, 112, 127, 141, 144, 171, 175, 179, 180, 188, 189, 191, 195, 197, 217, 220, 245, 247, 249, 275, 318, 393, 452, 463, 502 and 525)

- L. 57 and throughout: replace “inaccuracy” by “error frequency” or “error rate”.

We have changed the term inaccuracy throughout the text (line 49, 56, 83, 115, 190, 191, 214, 265, 275, 452, and 492)

- Strictly speaking, an mRNA codon is UGA, not TGA. This dilemma is not solved in the paper, which, again, makes it a difficult read.

We acknowledge the distinction between mRNA (U) and DNA (T) codons. However, we have opted for consistency by using the DNA codon notation throughout the paper for clarity and readability.

- L. 59: “how frequent errors are”: please correct grammar

We corrected it.

- L. 68: "We thus propose": why "thus"? This is entirely unclear in this context.

The term "thus" is used to show that the proposal of rules is a direct outcome of the confirmation of the dependence of protein termination accuracy on stop codon identity and genetic context.

- The authors claim higher readthrough error frequency than expected, but actually do not compare them with the previously data published data. The numbers from the previous publications should be explicitly cited.

We acknowledge the reviewer's concern and have revised the manuscript accordingly. The statement "We showed that SCM is more frequent than previously thought" is supported by the discussion in lines 459-465 and 472-474, where specific error rates from previous publications are cited. We have now included explicit citations in the summary statement to provide clarity and context.

- L. 186; "Despite this complex scenario, SCM events are consistent among biological replicates". This statement contradicts to the text in l. 166-173 and to the reply to reviewers.

We now removed the statement: "Despite this complex scenario, SCM events are consistent among biological replicates". We changed this section and added the result of the statistical analyses that the reviewer suggested (see answer to question #1).

References

- Fan, Yongqiang, Christopher R. Evans, Karl W. Barber, Kinshuk Banerjee, Kalyn J. Weiss, William Margolin, Oleg A. Igoshin, Jesse Rinehart, and Jiqiang Ling. 2017. "Heterogeneity of Stop Codon Readthrough in Single Bacterial Cells and Implications for Population Fitness." *Molecular Cell* 67 (5): 826-836.e5.
- Gessulat, Siegfried, Tobias Schmidt, Daniel Paul Zolg, Patroklos Samaras, Karsten Schnatbaum, Johannes Zerweck, Tobias Knaute, et al. 2019. "Prosit: Proteome-Wide Prediction of Peptide Tandem Mass Spectra by Deep Learning." *Nature Methods* 16 (6): 509–18.

REVIEWERS' COMMENTS

Reviewer #1 (Remarks to the Author):

Overall, the authors have improved the manuscript by removing some problematic results/statements. However, I still see some issues, particularly concerning reproducibility, which remain inadequately addressed. Additionally, the problem of terminology persists.

- Reproducibility and statistical analysis

It seems the authors have made some adjustments in the requested direction, which I appreciate. However, I still see some issues. Understanding what they did is sometimes challenging, considering also the errors in figure references within the text:

For the statistical analysis at positions 105 and 155 (new TableS1-S2 and new FigS4), it seems they used exclusively the replicates from Fig2C, excluding those from Figure S2 despite our explicit request for inclusion ('ALL replicates - 4 in total'). This is important, particularly considering that the main discrepancies were observed between these two datasets.

Moreover, they decline providing additional replicates in those cases where they are not 3.

Concerning the cumulative statistics for all positions (new TableS3 and new FigS5), it is unclear which data were used (M9 or LB medium, FigS2 data alone, or FigS2 combined with Fig2C data)

Statistical assessment of the FigS2 (together with the data of Fig 2C) data is important, as it also underlies the conclusions drawn in the former FigS7 and S8 ('Stop codon miscoding events occur evenly along mScarlet sequence' and 'The impact of predicted mRNA secondary structure on SCM').

- Number of cells used for the reconstruction of the fluorescence distributions

In certain instances, the fluorescence distributions in Fig2C were estimated based on a very low number of cells. This appears to be a source of variability among replicates (e.g., Position 105 TAA replicate 1 or Position 155 TGA replicate 1)

- Mass spectrometry data

I appreciate the introduction in the manuscript of some form of validation. However, I still believe these are mostly preliminary (yet valid) data. A targeted analysis would probably allow to obtain a reliable fragmentation spectrum and establish the existence of these peptides. Adding a comment in the manuscript indicating the preliminary nature of the MS data would be fair.

-Terminology

I don't understand why the authors insist on their incorrect and uncommon terminology. The precise problem lies in the creation of a common term to describe two very distinct biological phenomena. Given the large body of previous work the authors would be well advised to keep the common terminology to ensure the impact of their data in the field.

Response to Reviewer's comments

Reviewer #1 (Remarks to the Author):

Overall, the authors have improved the manuscript by removing some problematic results/statements. However, I still see some issues, particularly concerning reproducibility, which remain inadequately addressed. Additionally, the problem of terminology persists.

- Reproducibility and statistical analysis

It seems the authors have made some adjustments in the requested direction, which I appreciate. However, I still see some issues. Understanding what they did is sometimes challenging, considering also the errors in figure references within the text:

For the statistical analysis at positions 105 and 155 (new TableS1-S2 and new FigS4), it seems they used exclusively the replicates from Fig2C, excluding those from Figure S2 despite our explicit request for inclusion ('ALL replicates - 4 in total'). This is important, particularly considering that the main discrepancies were observed between these two datasets.

We acknowledge the observation that we exclusively used replicates from Fig 2C, omitting those from Figure S2 despite your explicit request. Our decision to focus on the replicates from Fig 2C was based on the consistent demonstration of significant differences among biological replicates in this dataset as indicated by the Friedman test (Table S1). Therefore, adding another replicate from Figure S2, which would introduce additional variability among replicates, would not alter the result and seems unnecessary.

Furthermore, the dataset from Fig 2C includes replicates at 18°C, 25°C, 30°C, 37°C, and 42°C, while the dataset from Figure S2 does not include experiments at 30°C. Considering that the Wilcoxon test (Table S2 and Fig S4) analyzes whether the replicates grown at lower temperatures exhibit significantly higher stop codon miscoding rates than those at higher temperatures. We consider it crucial to include as many temperature datapoint as possible and therefore exclude the dataset from Figure S2.

Nevertheless, we also provide statistical evidence of a temperature trend analyzing the dataset of Figure S4 (Table S3 and Fig S5).

Moreover, they decline providing additional replicates in those cases where they are not 3.

The dataset of Figure S2 includes three biological replicates. However, it is not uncommon in high throughput studies to encounter instances where a small number of replicates may not exhibit optimal growth conditions. In our study, a few replicates (two out of 90) did not grow properly. This can be attributed to the challenges associated with growing small volumes of *E. coli* cultures (200 µL) required for high-throughput studies. In these two cases, we analyzed the available two replicates.

Concerning the cumulative statistics for all positions (new TableS3 and new FigS5), it is unclear which data were used (M9 or LB medium, FigS2 data alone, or FigS2 combined with Fig2C data)

Table S3 and FigS5 refers to the dataset of FigS2 where cells were grown at 18°C, 25°C, 30°C, 37°C, and 42°C in LB and M9 media.

We now include "We assessed the median of fluorescence relative to the wild-type for all the reporters studied in LB and M9 media (dataset from Fig S2), excluding those with no SCM at any of the studied temperatures." in the caption of Figure S5 and Table S3.

Statistical assessment of the FigS2 (together with the data of Fig 2C) data is important, as it also underlies the conclusions drawn in the former FigS7 and S8 ('Stop codon miscoding events occur evenly along mScarlet sequence' and 'The impact of predicted mRNA secondary structure on SCM').

Please refer to the previous response regarding reproducibility and statistical analysis.

- Number of cells used for the reconstruction of the fluorescence distributions
In certain instances, the fluorescence distributions in Fig2C were estimated based on a very low number of cells. This appears to be a source of variability among replicates (e.g., Position 105 TAA replicate 1 or Position 155 TGA replicate 1)

Yes, the number of cells vary across samples and conditions for the reasons we described in our previous response, and it certainly contributes to variability, yet, the main conclusions hold as the statistical analyses that the reviewer suggested previously confirmed.

- Mass spectrometry data

I appreciate the introduction in the manuscript of some form of validation. However, I still believe these are mostly preliminary (yet valid) data. A targeted analysis would probably allow to obtain a reliable fragmentation spectrum and establish the existence of these peptides. Adding a comment in the manuscript indicating the preliminary nature of the MS data would be fair.

We edited the discussion accordingly (lines 502): "Further validation through quantitative mass spectrometry will be invaluable in confirming the identity of these 16 peptides. Exploring their role in generating protein diversity is an interesting open research question."

-Terminology

I don't understand why the authors insist on their incorrect and uncommon terminology. The precise problem lies in the creation of a common term to describe two very distinct biological phenomena. Given the large body of previous work the authors would be well advised to keep the common terminology to ensure the impact of their data in the field.

We acknowledge the concern regarding the terminology used in our work. However, the field lacks a term for describing the complex biological phenomena we are addressing. We define here a new term that does not imply a mechanism and an origin of the errors, and encompasses both transcription and translation errors. We are not aware of an already existing term that is defined accordingly. We deliberately do not want to use stop codon readthrough (RT) because it implies that the ribosome "reads through", while our data clearly shows that also RNA polymerase errors contribute to the elimination of stop codons.

We feel that the introduction of a new term, such as "stop codon miscoding" (SCM) is necessary to pinpoint the importance of **both transcription and translation origin of errors and better reflects our data**. While RT is extensively used in the literature, it is unfortunately not used consistently. Sometimes it refers to both transcription and translation errors, sometimes it refers specifically to frameshift mediated stop codon readthrough (see for example Freitag et al., <https://www.nature.com/articles/nature11051>). Therefore, we prefer keeping the term we coined, SCM, defined in the current manuscript.